# Posterior Sampling for Competitive RL: Function Approximation and Partial Observation

**Shuang Qiu**[*]
HKUST
masqiu@ust.hk

**Ziyu Dai**[*]
New York University
zd2365@cims.nyu.edu

**Han Zhong**
Peking University
hanzhong@stu.pku.edu.cn

**Zhaoran Wang**
Northwestern University
zhaoranwang@gmail.com

**Zhuoran Yang**
Yale University
zhuoran.yang@yale.edu

**Tong Zhang**
HKUST
tongzhang@ust.hk

## Abstract

This paper investigates posterior sampling algorithms for competitive reinforcement learning (RL) in the context of general function approximations. Focusing on zero-sum Markov games (MGs) under two critical settings, namely self-play and adversarial learning, we first propose the self-play and adversarial generalized eluder coefficient (GEC) as complexity measures for function approximation, capturing the exploration-exploitation trade-off in MGs. Based on self-play GEC, we propose a model-based self-play posterior sampling method to control both players to learn Nash equilibrium, which can successfully handle the partial observability of states. Furthermore, we identify a set of partially observable MG models fitting MG learning with the adversarial policies of the opponent. Incorporating the adversarial GEC, we propose a model-based posterior sampling method for learning adversarial MG with potential partial observability. We further provide low regret bounds for proposed algorithms that can scale sublinearly with the proposed GEC and the number of episodes $T$. To the best of our knowledge, we for the first time develop generic model-based posterior sampling algorithms for competitive RL that can be applied to a majority of tractable zero-sum MG classes in both fully observable and partially observable MGs with self-play and adversarial learning.

## 1 Introduction

Multi-agent reinforcement learning (MARL) tackles sequential decision-making problems where multiple players simultaneously interact with the shared environment, affecting each other's behavior in a coupled manner. Under a competitive reinforcement learning (RL) setting, the goal of each player is to maximize (*resp.* minimize) her own cumulative gains (*resp.* losses) in the presence of other agents. Recent years have been tremendous practical successes of MARL in a variety of application domains, such as autonomous driving [56], Go [58], StarCraft [65], Dota2 [8] and Poker [10]. These successes are attributed to advanced MARL algorithms that can coordinate multiple players by exploiting potentially partial observations of the latent states and employ powerful function approximators (neural networks in particular), which empower us to tackle practical problems with large state spaces.

Apart from the empirical success, there is a growing body of literature on establishing theoretical guarantees for Markov games (MGs) [57] – a standard framework for describing the dynamics of competitive RL. In particular, [71, 16, 33, 28, 77] extend the works in single-agent reinforcement

---

[*]Equal Contribution.

37th Conference on Neural Information Processing Systems (NeurIPS 2023).

learning (RL) with function approximation [29, 60, 67, 34, 5, 11, 19, 31, 21] by developing sample-efficient algorithms that are capable to solve two-player zero-sum MGs with function approximation. In addition, as opposed to the aforementioned literature on MGs assuming the state of players is fully observable, the recent work [41] analyze Markov games under partial observability [41], i.e., the complete information about underlying states is lacking. However, most of the existing works are built upon the principle of "optimism in the face of uncertainty" (OFU) [38] for exploration. Furthermore, from a practical perspective, achieving optimism often requires explicit construction of bonus functions, which are often designed in a model-specific fashion and computationally challenging to implement.

Another promising strand of exploration techniques is based on posterior sampling, which is shown by previous works on bandits [12] and RL [48] to perform better than OFU-based algorithms. Meanwhile, posterior sampling methods, unlike OFU-based algorithms [31, 21] that need to solve complex optimization problems to achieve optimism, can be efficiently implemented by ensemble approximations [48, 44, 17, 46] and stochastic gradient Langevin dynamics (SGLD) [70]. Despite the superiority of posterior sampling, its theoretical understanding in MARL remains limited. The only exception is [73], which proposes a model-free posterior sampling algorithm for zero-sum MGs with general function approximation. However, [73] cannot capture some common tractable competitive RL models with a model-based nature, such as linear mixture MGs [16] and low witness rank MGs [28]. Moreover, their result is restricted to the fully observable MGs without handling the partial observability of the players' states. Therefore, we raise the following question:

*Can we design provably sample-efficient posterior sampling algorithms for competitive RL with even partial observations under general function approximation?*

Concretely, the above question poses three major challenges. First, despite the success of the OFU principle in partially observable Markov games (POMGs), it remains elusive how to incorporate the partial observations into the posterior sampling framework under a MARL setting with provably efficient exploration. Second, it is also unclear whether there is a generic function approximation condition that can cover more known classes in both full and partial observable MARL and is meanwhile compatible with the posterior sampling framework. Third, with the partial observation and function approximation, it is challenging to explore how we can solve MGs under the setups of self-play, where all players can be coordinated together, and adversarial learning, where the opponents' policies are adversarial and uncontrollable by the learner. Our work takes an initial step towards tackling such challenges by concentrating on the typical competitive RL scenario, the two-player zero-sum MG, and proposing statistically efficient posterior sampling algorithms under function approximation that can solve both self-play and adversarial MGs with full and partial observations.

**Contributions.** Our contributions are four-fold: **(1)** We first propose the two generalized eluder coefficient (GEC) as the complexity measure for the competitive RL with function approximation, namely self-play GEC and adversarial GEC, that captures the exploration-exploitation tradeoff in many existing MGs, including linear MGs, linear mixture MGs, weakly revealing POMGs, decodable POMGs. The proposed measures also generalize the recently developed GEC condition [83] from single-agent RL to MARL, suitably adjusting the exploration policy particularly for the adversarial setting. **(2)** Incorporating the proposed self-play GEC for general function approximation, we propose a model-based posterior sampling algorithm with self-play to learn the Nash equilibrium (NE), which successfully handles the partial observability of states along with a full observable setup by carefully designed likelihood functions. **(3)** We identify POMG models aligned with the form of the adversarial GEC, which fit MG learning with adversarially-varying policies of the opponent. We further propose a model-based posterior sampling algorithm for adversarial learning with general function approximation. **(4)** We prove regret bounds for our proposed algorithms that scale sublinearly with the number of episodes $T$, the corresponding GEC $d_{\mathrm{GEC}}$, and a quantity measuring the coverage of the optimal model by the initial model sampling distribution. To the best of our knowledge, we present the first model-based posterior sampling approaches to sample-efficiently learn MGs with function approximation, handling partial observability in both self-play and adversarial settings.

**Related Works.** There is a large body of literature studying MGs, especially zero-sum MGs. In the self-play setting, many papers have focused on solving approximate NE in tabular zero-sum MGs [6, 7, 71, 51, 43], zero-sum MGs with linear function approximation [71, 16], zero-sum MGs with low-rank structures [50, 79, 47], and zero-sum MGs with general function approximation [33, 28, 73]. On the other hand, there are also several recent papers focusing on the adversarial setting

[71, 61, 33, 28] aim to learn the Nash value under the setting of unrevealed opponent's policies, where the adversarial policies of the opponent are unobservable. In addition, another line of adversarial MGs concentrates on a revealed policy setting, where the opponent's full policy can be observed, leading to efficiently learning a sublinear regret comparing against the best policy in hindsight. Particularly, [42] and [77] develop efficient algorithms in tabular and function approximation settings, respectively. Our approach focuses on the unrevealed policy setting, which is considered to be a more practical setup. There are also works studying MGs from various aspects [59, 32, 45, 84, 35, 20, 52, 9, 82, 18, 72, 74], such as multi-player general-sum MGs, reward-free MGs, MGs with delayed feedback, and offline MGs, which are beyond the scope of our work. Most of the aforementioned works follow the OFU principle and differ from our posterior sampling methods. The recent work [73] proposes a model-free posterior sampling algorithm for two-player zero-sum MGs but is limited to the self-play setting with fully observable states. Moreover, their work requires a strong Bellman completeness assumption that is restrictive compared to only requiring realizability in our work, mainly due to the monotonicity, meaning that adding a new function to the function class may violate it. Many model-based models like linear mixture MGs [16] and low witness rank MGs [28] are not Bellman-complete, so they cannot be captured by [73]. Without the completeness assumption, our model-based posterior sampling approaches can solve a rich class of tractable MGs, including linear mixture MGs, low witness rank MGs [28], and even POMGs, tackling both self-play and adversarial learning settings.

Our work is related to a line of research on posterior sampling methods in RL. For single-agent RL, most existing works such as [54] analyze the Bayesian regret bound. There are also some works [4, 53, 76] focusing on the frequentist (worst-case) regret bound. Our work is more closely related to the recently developed feel-good Thompson sampling technique proposed by [81] for the frequentist regret bound, and its extension to single-agent RL [19, 2, 3] and two-player zero-sum MGs [73].

Our work is also closely related to the line of research on function approximation in RL. Such a line of works proposes algorithms for efficient policy learning under diverse function approximation classes, spanning from linear Markov decision processes (MDPs) [34], linear mixture MDPs [5, 85] to nonlinear and general function classes, including, for instance, generalized linear MDPs [69], kernel and neural function classes [75], bounded eluder dimension [49, 68], Bellman rank [29], witness rank [60], bellman eluder dimension [31], bilinear [21], decision-estimation coefficient [24, 14], decoupling coefficient [19], admissible Bellman characterization [15], and GEC [83] classes.

The research on partial observability in RL [27] is closely related to our work. The works [36, 30] show that learning history-dependent policies generally can cause an exponential sample complexity. Thus, many recent works focus on analyzing tractable subclasses of partially observable Markov decision processes (POMDPs), which includes weakly revealing POMDPs [30, 39], observable POMDPs [26, 25], decodable POMDPs [22, 23], low-rank POMDPs[66], regular PSR [78], PO-bilinear class [63], latent MDP with sufficient tests [37], B-stable PSR [13], well-conditioned PSR [40], POMDPs with deterministic transition kernels [30, 62], and GEC [83]. Nevertheless, in contrast to our work which focuses on the two-player competitive setting with partial observation, these papers merely consider the single-agent setting. The recent research [41] further generalizes weakly revealing POMDPs to its multi-agent counterpart, weakly revealing POMGs, in a general-sum multi-player setting based on the OFU principle. But when specialized to the two-player case, our work proposes a general function class that can subsume the class of weakly revealing POMGs as a special case. It would be intriguing to generalize our framework to the general-sum settings in the future.

**Notations.** We denote by $\mathrm{KL}(P||Q) = \mathbb{E}_{x \sim P}[\log(\mathrm{d}P(x)/\mathrm{d}Q(x))]$ the KL divergence and $D_{\mathrm{He}}^2(P, Q) = 1/2 \cdot \mathbb{E}_{x \sim P}(\sqrt{\mathrm{d}Q(x)/\mathrm{d}P(x)} - 1)^2$ the Hellinger distance. We denote by $\Delta_{\mathcal{X}}$ the set of all distributions over $\mathcal{X}$ and $\mathrm{Unif}(\mathcal{X})$ the uniform distribution over $\mathcal{X}$. We let $x \wedge y$ be $\min\{x, y\}$.

## 2  Problem Setup

We introduce the basic concept of the two-player zero-sum Markov game (MG), function approximation, and the new complexity conditions for function approximation. Concretely, we study two typical classes of MGs, i.e., fully observable MGs and partially observable MGs, as defined below.

**Fully Observable Markov Game.** We consider an episodic two-player zero-sum fully observable Markov game (FOMG[2]) specified by a tuple $(\mathcal{S}, \mathcal{A}, \mathcal{B}, \mathbb{P}, r, H)$, where $\mathcal{S}$ is the state space, $\mathcal{A}$ and

---

[2]FOMG is typically referred to as MG by most literature. We adopt FOMG to differentiate it from POMG.

$\mathcal{B}$ are the action spaces of Players 1 and 2 respectively, $H$ is the length of the episode. We denote by $\mathbb{P} := \{\mathbb{P}_h\}_{h=1}^H$ the transition kernel with $\mathbb{P}_h(s'|s,a,b)$ specifying the probability (density) of transitioning from state $s$ to state $s'$ given Players 1 and 2's actions $a \in \mathcal{A}$ and $b \in \mathcal{B}$ at step $h$. We denote the reward function as $r = \{r_h\}_{h=1}^H$ with $r_h : \mathcal{S} \times \mathcal{A} \times \mathcal{B} \mapsto [0,1]$ being the reward received by players at step $h$. We define $\pi = \{\pi_h\}_{h=1}^H$ and $\nu = \{\nu_h\}_{h=1}^H$ as *Markovian* policies for Players 1 and 2, i.e., $\pi_h(a|s)$ and $\nu_h(b|s)$ are the probability of taking action $a$ and $b$ conditioned on the current state $s$ at step $h$. Without loss of generality, we assume the initial state $s_1$ is fixed for each episode. We consider a realistic setting where the transition kernel $\mathbb{P}$ is *unknown* and thereby needs to be approximated using the collected data.

**Partially Observable Markov Game.** This paper further studies an episodic zero-sum partially observable Markov game (POMG), which is distinct from the FOMG setup in that the state $s$ is not directly observable. In particular, a POMG is represented by a tuple $(\mathcal{S}, \mathcal{A}, \mathcal{B}, \mathcal{O}, \mathbb{P}, \mathbb{O}, \mu, r, H)$, where $\mathcal{S}$, $\mathcal{A}$, $\mathcal{B}$, $H$, and $\mathbb{P}$ are similarly the state and action spaces, the episode length, and the transition kernel. Here $\mu_1(\cdot)$ denotes the initial state distribution. We denote by $\mathbb{O} := \{\mathbb{O}_h\}_{h=1}^H$ the emission kernel so that $\mathbb{O}_h(o|s)$ is the probability of having a partial observation $o \in \mathcal{O}$ at state $s$ with $\mathcal{O}$ being the observation space. Since we only have an observation $o$ of a state, the reward function is defined as $r := \{r_h\}_{h=1}^H$ with $r_h(o,a,b) \in [0,1]$ depending on actions $a, b$ and the observation $o$, and the policies for players are defined as $\pi = \{\pi_h\}_{h=1}^H$ and $\nu = \{\nu_h\}_{h=1}^H$, where $\pi_h(a_h|\tau_{h-1}, o_h)$ and $\nu_h(b_h|\tau_{h-1}, o_h)$ is viewed as the probability of taking actions $a_h$ and $b_h$ depending on all histories $(\tau_{h-1}, o_h)$. Here we let $\tau_h := (o_1, a_1, b_1 \ldots, o_h, a_h, b_h)$. Then, in contrast to FOMGs, the policies in POMGs are *history-dependent*, defined on all prior observations and actions rather than the current state $s$. We define $\mathbf{P}_h^{\pi,\nu}(\tau_h) := \int_{\mathcal{S}^h} \mu_1(s_1)$ $\prod_{h'=1}^{h-1} [\mathbb{O}_{h'}(o_{h'}|s_{h'}) \pi_{h'}(b_{h'}|\tau_{h'-1}, o_{h'}) \nu_{h'}(b_{h'}|\tau_{h'-1}, o_{h'}) \mathbb{P}_{h'}(s_{h'+1}|s_{h'}, a_{h'}, b_{h'})] \mathbb{O}_h(o_h|s_h) \mathrm{d}s_{1:h}$, which is the joint distribution of $\tau_h$ under the policy pair $(\pi, \nu)$. Removing policies in $\mathbf{P}_h^{\pi,\nu}$, we define the function $\mathbf{P}_h(\tau_h) := \int_{\mathcal{S}^h} \mu_1(s_1) \prod_{h'=1}^{h-1} [\mathbb{O}_{h'}(o_{h'}|s_{h'}) \mathbb{P}_{h'}(s_{h'+1}|s_{h'}, a_{h'}, b_{h'})] \mathbb{O}_h(o_h|s_h) \mathrm{d}s_{1:h}$. We assume that the parameters $\theta := (\mu_1, \mathbb{P}, \mathbb{O})$ are *unknown* and thus $\mathbf{P}_h$ is *unknown* as well, which should be approximated in algorithms via online interactions.

**Online Interaction with the Environment.** In POMGs, at step $h$ of episode $t$ of the interaction, players take actions $a_h^t \sim \pi_h^t(\cdot|\tau_{h-1}^t, o_h^t)$ and $b_h^t \sim \nu_h^t(\cdot|\tau_{h-1}^t, b_{h-1}^t, o_h^t)$ depending on their action and observation histories, receiving a reward $r_h(o_h^t, a_h^t, b_h^t)$ and transitions from the latent state $s_h^t$ to $s_{h+1}^t \sim \mathbb{P}_h(\cdot | s_h^t, a_h^t, b_h^t)$ with an observation $o_{h+1}^t \sim \mathbb{O}_h(\cdot|s_h^t)$ generated. When the underlying state $s_h^t$ is observable and the policies become Markovian, we have actions $a_h^t \sim \pi_h^t(s_h^t)$ and $b_h^t \sim \nu_h^t(s_h^t)$ and the reward $r_h(s_h^t, a_h^t, b_h^t)$. Then, it reduces to the interaction process under the FOMG setting.

**Value Function, Best Response, and Nash Equilibrium.** To characterize the learning objective and the performance of the algorithms, we define the value function as the expected cumulative rewards under the policy pair $(\pi, \nu)$ starting from the initial step $h = 1$. For FOMGs, we define the value function as $V^{\pi,\nu} := \mathbb{E}[\sum_{h=1}^H r_h(s_h, a_h, b_h) \mid s_1, \pi, \nu, \mathbb{P}]$, where the expectation is taken over all the randomness induced by $\pi$, $\nu$, and $\mathbb{P}$. For POMG, we define the value function as $V^{\pi,\nu} := \mathbb{E}[\sum_{h=1}^H r_h(o_h, a_h, b_h) | \pi, \nu, \theta]$, with the expectation taken for $\pi$, $\nu$, and $\theta$.

Our work studies the competitive setting of RL, where Player 1 (*max-player*) aims to maximize the value function $V^{\pi,\nu}$ while Player 2 (*min-player*) aims to minimize it. With the defined value function, given a policy pair $(\pi, \nu)$, we define their *best responses* respectively as $\mathtt{br}(\pi) \in \arg\min_\nu V^{\pi,\nu}$ and $\mathtt{br}(\nu) \in \arg\max_\pi V^{\pi,\nu}$. Then, we say a policy pair $(\pi^*, \nu^*)$ is a *Nash equilibrium* (NE) if

$$V^{\pi^*,\nu^*} = \max_\pi \min_\nu V^{\pi,\nu} = \min_\nu \max_\pi V^{\pi,\nu}.$$

Thus, it always holds that $\pi^* = \mathtt{br}(\nu^*)$ and $\nu^* = \mathtt{br}(\pi^*)$. For abbreviation, we denote $V^* = V^{\pi^*,\nu^*}$, $V^{\pi,*} = \min_\nu V^{\pi,\nu}$, and $V^{*,\nu} = \max_\pi V^{\pi,\nu}$, which implies $V^* = V^{\pi^*,*} = V^{*,\nu^*}$ for NE $(\pi^*, \nu^*)$. Moreover, we define the policy pair $(\pi, \nu)$ as an $\varepsilon$-approximate NE if it satisfies $V^{*,\nu} - V^{\pi,*} \le \varepsilon$.

**Function Approximation.** Since the environment is unknown to players, the model-based RL setting requires us to learn the true model of the environment, $f^*$, via (general) function approximation. We use the functions $f$ lying in a general model function class $\mathcal{F}$ to approximate the environment. We make a standard realizability assumption on the relationship between the model class and the true model.

**Assumption 1** (Realizability). *For a model class $\mathcal{F}$, the true model $f^*$ satisfies $f^* \in \mathcal{F}$.*

In our work, the true model $f^*$ represents the transition kernel $\mathbb{P}$ for the FOMG and $\theta$ for the POMG. For any $f \in \mathcal{F}$, we let $\mathbb{P}_f$ and $\theta_f = (\mu_f, \mathbb{P}_f, \mathbb{O}_f)$ be the models under the approximation function $f$ and $V_f^{\pi,\nu}$ the value function associated with $f$. For POMGs, we denote $\mathbf{P}_{f,h}^{\pi,\nu}$ and $\mathbf{P}_{f,h}$ as $\mathbf{P}_h^{\pi,\nu}$ and $\mathbf{P}_h$ under the model $f$.

**MGs with Self-Play and Adversarial Learning.** Our work investigates two important MG setups for competitive RL, which are the self-play setting and the adversarial setting. In the self-play setting, the learner can control *both* players together to execute the proposed algorithms to learn an approximate NE. Therefore, our objective is to design sample-efficient algorithms to generate a sequence of policy pairs $\{(\pi^t, \nu^t)\}_{t=1}^T$ in $T$ episodes such that the following regret can be minimized,

$$\mathrm{Reg}^{\mathrm{sp}}(T) := \sum_{t=1}^T \left[ V_{f^*}^{*,\nu^t} - V_{f^*}^{\pi^t,*} \right].$$

In the adversarial setting, we can no longer coordinate both players, and only *single* player is controllable. Under such a circumstance, the opponent plays arbitrary and even adversarial policies. Wlog, suppose that the main player is the max-player with the policies $\{\pi^t\}_{t=1}^T$ generated by a carefully designed algorithm and the opponent is min-player with arbitrary policies $\{\nu^t\}_{t=1}^T$. The objective of the algorithm is to learn policies $\{\pi^t\}_{t=1}^T$ to maximize the overall cumulative rewards in the presence of an adversary. To measure the performance of algorithms, we define the following regret for the adversarial setting by comparing the learned value against the Nash value, i.e.,

$$\mathrm{Reg}^{\mathrm{adv}}(T) = \sum_{t=1}^T \left[ V_{f^*}^* - V_{f^*}^{\pi^t,\nu^t} \right].$$

## 3 Model-Based Posterior Sampling for the Self-Play Setting

We propose algorithms aiming to generate a sequence of policy pairs $\{(\pi^t, \nu^t)\}_{t=1}^T$ by controlling the learning process of both players such that the regret $\mathrm{Reg}^{\mathrm{sp}}(T)$ is small. Such a regret can be decomposed into two parts, namely $\sum_{t=1}^T [V_{f^*}^* - V_{f^*}^{\pi^t,*}]$ and $\sum_{t=1}^T [V_{f^*}^{*,\nu^t} - V_{f^*}^*]$, which inspires our to design algorithms for learning $\{\pi^t\}_{t=1}^T$ and $\{\nu^t\}_{t=1}^T$ separately by targeting at minimizing these two parts respectively. Due to the symmetric structure of such a game learning problem, we propose the algorithm to learn $\{\pi^t\}_{t=1}^T$ as summarized in Algorithm 1. The algorithm for learning $\{\nu^t\}_{t=1}^T$ can be proposed in a symmetric way in Algorithm 3, which is deferred to Appendix A. Our proposed algorithm features an integration of the model-based posterior sampling and the exploiter-guided self-play in a multi-agent learning scenario. In Algorithm 1, Player 1 is the main player, while Player 2 is called the exploiter, who assists the learning of the main player by exploiting her weakness.

**Posterior Sampling for the Main Player.** The posterior sampling constructs a posterior distribution $p^t(\cdot | Z^{t-1})$ over the function class $\mathcal{F}$ each round based on collected data and a pre-specified prior distribution $p^0(\cdot)$, where $Z^{t-1}$ denotes the random history up to the end of the $(t-1)$-th episode. For ease of notation, hereafter, we omit $Z^{t-1}$ in the posterior distribution. Most recent literature shows that adding an optimism term in the posterior distribution can lead to sample-efficient RL algorithms. Thereby, we define the distribution $p^t(\cdot)$ over the function class $\mathcal{F}$ for the main player as in Line 3 of Algorithm 1, which is proportional to $p^0(f) \exp[\gamma V_f^* + \sum_{\tau=1}^{t-1} \sum_{h=1}^H L_h^\tau(f)]$. Here, $\gamma V_f^*$ serves as the optimism term, and $L_h^\tau(f)$ is the likelihood function built upon the pre-collected data. Such a construction of $p^t(\cdot)$ indicates that we will assign a higher probability (density) to a function $f$, which results in higher values of the combination of the optimism term and the likelihood function. We sample a model $\overline{f}^t$ from the distribution $p^t(\cdot)$ over the model class and learn the policy $\pi^t$ for the main player such that $(\pi^t, \overline{\nu}^t)$ is the NE of the value function under the model $\overline{f}^t$ in Line 4, where $\overline{\nu}^t$ is a dummy policy and only used in our theoretical analysis.

**Posterior Sampling for the Exploiter.** The exploiter aims to track the best response of $\pi^t$ to assist learning a low regret. The best response of $\pi^t$ generated by the exploiter is nevertheless based on a value function under a different model than $\overline{f}^t$. Specifically, for the exploiter, we define the posterior sampling distribution $q^t(\cdot)$ using an optimism term $-\gamma V_f^{\pi^t,*}$ and the summation of likelihood functions, i.e., $\sum_{\tau=1}^{t-1} \sum_{h=1}^H L_h^\tau(f)$, along with a prior distribution $q^0(\cdot)$, in Line 5 of Algorithm 1. The negative term $-\gamma V_f^{\pi^t,*}$ favors a model with a low value and is thus optimistic from the exploiter's perspective but pessimistic for the main player. We then sample a model $\underline{f}^t$ from $q^t(\cdot)$ and compute the best response of $\pi^t$, denoted as $\underline{\nu}^t$, under the model $\underline{f}^t$ as in Line 7.

---

**Algorithm 1** Model-Based Posterior Sampling for Self-Play (Max-Player)

---

1: **Input:** Model class $\mathcal{F}$, prior distributions $p^0$ and $q^0$, $\gamma_1$, and $\gamma_2$.
2: **for** $t = 1, \ldots, T$ **do**
3:      Draw a model $\overline{f}^t \sim p^t(\cdot)$ with defining $p^t(f) \propto p^0(f) \exp[\gamma_1 V_f^* + \sum_{\tau=1}^{t-1} \sum_{h=1}^{H} L_h^\tau(f)]$
4:      Compute $\pi^t$ by letting $(\pi^t, \overline{\nu}^t)$ be the NE of $V_{\overline{f}^t}^{\pi,\nu}$
5:      Draw a model $\underline{f}^t \sim q^t(\cdot)$ with defining $q^t(f) \propto q^0(f) \exp[-\gamma_2 V_f^{\pi^t,*} + \sum_{\tau=1}^{t-1} \sum_{h=1}^{H} L_h^\tau(f)]$
6:      Compute $\underline{\nu}^t$ by letting $\underline{\nu}^t$ be the best response of $\pi^t$ w.r.t. $V_{\underline{f}^t}^{\pi,\nu}$
7:      Collect data $\mathcal{D}^t$ by executing the joint exploration policy $\sigma^t$
8:      Define the likelihood functions $\{L_h^t(f)\}_{h=1}^H$ using the collected data $\mathcal{D}^t$
9: **end for**
10: **Return:** $(\pi^1, \ldots, \pi^T)$.

---

**Data Sampling and Likelihood Function.** With the learned $\pi^t$ and $\underline{\nu}^t$, we define a joint exploration policy $\sigma^t$ in Line 7 of Algorithm 1, by executing which we can collect a dataset $\mathcal{D}^t$. We are able to further construct the likelihood functions $\{L_h^t(f)\}_{h=1}^H$ in Line 8 using $\mathcal{D}^t$. Different game settings require specifying diverse exploration policies $\sigma^t$ and likelihood functions $\{L_h^t(f)\}_{h=1}^H$. Particularly, for the game classes mainly discussed in this work, we set $\sigma^t = (\pi^t, \underline{\nu}^t)$ for both FOMGs and POMGs. In FOMGs, we let $\mathcal{D}^t = \{(s_h^t, a_h^t, b_h^t, s_{h+1}^t)\}_{h=1}^H$, where for each $h \in [H]$, the data point $(s_h^t, a_h^t, b_h^t, s_{h+1}^t)$ is collected by executing $\sigma^t$ to the $h$-th step of the game. The corresponding likelihood function is defined using the transition kernel as

$$L_h^t(f) = \eta \log \mathbb{P}_{f,h}(s_{h+1}^t \,|\, s_h^t, a_h^t, b_h^t). \tag{1}$$

Furthermore, under the POMG setting, we let the dataset be $\mathcal{D}^t = \{\tau_h^t\}_{h=1}^H$, where the data point $\tau_h^t = (o_1^t, a_1^t, b_1^t \ldots, o_h^t, a_h^t, b_h^t)$ is collected by executing $\sigma^t$ to the $h$-th step of the game for each $h \in [H]$. We further define the associated likelihood function as

$$L_h^t(f) = \eta \log \mathbf{P}_{f,h}(\tau_h^t). \tag{2}$$

Such a construction of the likelihood function in a log-likelihood form can result in learning a model $f$ well approximating the true model $f^*$ measured via the Hellinger distance.

### 3.1 Regret Analysis for the Self-play Setting

Our regret analysis is based on a novel structural complexity condition for multi-agent RL and a quantity to measure how the well the prior distributions cover the optimal model $f^*$. We first define the following condition for the self-play setting.

**Definition 1** (Self-Play GEC). *For any sequences of functions $f^t, g^t \in \mathcal{F}$, suppose that a pair of policies $(\pi^t, \nu^t)$ satisfies: (a) $\pi^t = \mathrm{argmax}_\pi \min_\nu V_{f^t}^{\pi,\nu}$ and $\nu^t = \mathrm{argmin}_\nu V_{g^t}^{\pi^t,\nu}$, or (b) $\nu^t = \mathrm{argmin}_\nu \max_\pi V_{f^t}^{\pi,\nu}$ and $\pi^t = \mathrm{argmax}_\pi V_{g^t}^{\pi,\nu^t}$. Denoting the joint exploration policy as $\sigma^t$ depending on $f^t$ and $g^t$, for any $\rho \in \{f, g\}$ and $(\pi^t, \nu^t)$ following (a) and (b), the self-play GEC $d_{\mathrm{GEC}}$ is defined as the minimal constant $d$ satisfying*

$$\left| \sum_{t=1}^T \left( V_{\rho^t}^{\pi^t, \nu^t} - V_{f^*}^{\pi^t, \nu^t} \right) \right| \leq \left[ d \sum_{h=1}^H \sum_{t=1}^T \left( \sum_{\tau=1}^{t-1} \mathbb{E}_{(\sigma^\tau, h)} \ell(\rho^t, \xi_h^\tau) \right) \right]^{\frac{1}{2}} + 2H(dHT)^{\frac{1}{2}} + \epsilon HT.$$

Our definition of self-play GEC is inspired by [83] for the single-agent RL. Then, it shares an analogous meaning to the single-agent GEC. Here $(\sigma^\tau, h)$ implies running the joint exploration policy $\sigma^\tau$ to step $h$ to collect a data point $\xi_h^\tau$. The LHS of the inequality is viewed as the prediction error and the RHS is the training error defined on a loss function $\ell$ plus a burn-in error $2H(dHT)^{\frac{1}{2}} + \epsilon HT$ that is non-dominating when $\epsilon$ is small. The loss function $\ell$ and $\epsilon$ can be problem-specific. We determine $\ell(f, \xi_h)$ for FOMGs with $\xi_h = (s_h, a_h)$ and POMGs with $\xi_h = \tau_h$ respectively as

$$\text{FOMG: } D_{\mathrm{He}}^2(\mathbb{P}_{f,h}(\cdot|\xi_h), \mathbb{P}_{f^*,h}(\cdot|\xi_h)), \quad \text{POMG: } 1/2 \cdot \left( \sqrt{\mathbf{P}_{f,h}(\xi_h)/\mathbf{P}_{f^*,h}(\xi_h)} - 1 \right)^2, \tag{3}$$

such that $\mathbb{E}_{(\sigma,h)}[\ell(f,\xi_h)] = D^2_{\mathrm{He}}(\mathbf{P}^\sigma_{f^*,h}, \mathbf{P}^\sigma_{f,h})$ for POMGs. The intuition for GEC is that if hypotheses have a small training error on a well-explored dataset, then the out-of-sample prediction error is also small, which characterizes the hardness of environment exploration.

Since the posterior sampling steps in our algorithms depend on the initial distributions $p^0$ and $q^0$, we define the following quantity to measure how well the prior distributions $p^0$ and $q^0$ cover the optimal model $f^* \in \mathcal{F}$, which is also a multi-agent generalization of its single-agent version [2, 83].

**Definition 2** (Prior around True Model). *Given $\beta > 0$ and any distribution $p^0 \in \Delta_\mathcal{F}$, we define*

$$\omega(\beta, p^0) = \inf_{\varepsilon > 0}\{\beta\varepsilon - \ln p^0[\mathcal{F}(\varepsilon)]\},$$

*where $\mathcal{F}(\varepsilon) := \{f \in \mathcal{F} : \sup_{h,s,a,b} \mathrm{KL}^{\frac{1}{2}}(\mathbb{P}_{f^*,h}(\cdot \mid s,a,b)\|\mathbb{P}_{f,h}(\cdot \mid s,a,b)) \leq \varepsilon\}$ for FOMGs and $\mathcal{F}(\varepsilon) := \{f \in \mathcal{F} : \sup_{\pi,\nu} \mathrm{KL}^{\frac{1}{2}}(\mathbf{P}^{\pi,\nu}_{f^*,H}\|\mathbf{P}^{\pi,\nu}_{f,H}) \leq \varepsilon\}$ for POMGs.*

When the model class $\mathcal{F}$ is a finite space, if let $p^0 = \mathrm{Unif}(\mathcal{F})$, we simply know that $\omega(\beta, p^0) \leq \log|\mathcal{F}|$ where $|\mathcal{F}|$ is the cardinality of $\mathcal{F}$. Furthermore, for an infinite function class $\mathcal{F}$, the term $\log|\mathcal{F}|$ can be substituted by a quantity having logarithmic dependence on the covering number of the function class $\mathcal{F}$. With the multi-agent GEC condition and the definition of $\omega$, we have the following regret bound for both FOMGs and POMGs.

**Proposition 1.** *Letting $\eta = 1/2$, $\gamma_1 = 2\sqrt{\omega(4HT, p^0)T/d_{\mathrm{GEC}}}$, $\gamma_2 = 2\sqrt{\omega(4HT, q^0)T/d_{\mathrm{GEC}}}$, $\epsilon = 1/\sqrt{HT}$ in Definition 1, when $T \geq \max\{4H^2\omega(4HT, p^0)/d_{\mathrm{GEC}}, 4H^2\omega(4HT, q^0)/d_{\mathrm{GEC}}, d_{\mathrm{GEC}}/H\}$, under both FOMG and POMG settings, Algorithm 1 admits the following regret bound,*

$$\mathbb{E}[\mathrm{Reg}^{\mathrm{sp}}_1(T)] := \mathbb{E}[\textstyle\sum_{t=1}^T (V^*_{f^*} - V^{\pi^t,*}_{f^*})] \leq 6\sqrt{d_{\mathrm{GEC}}HT \cdot [\omega(4HT, p^0) + \omega(4HT, q^0)]}.$$

This proposition gives the upper bound $\mathbb{E}[\mathrm{Reg}^{\mathrm{sp}}_1(T)]$ following the updating rules in Algorithm 1 when the max-player is the main player. As Algorithm 3 is symmetric to Algorithm 1, we obtain the following regret bound of $\mathbb{E}[\mathrm{Reg}^{\mathrm{sp}}_2(T)]$ for Algorithm 3 when the min-player is the main player.

**Proposition 2.** *Under the same parameter settings as Proposition 1, Algorithm 3 admits the following regret bound,*

$$\mathbb{E}[\mathrm{Reg}^{\mathrm{sp}}_2(T)] := \mathbb{E}[\textstyle\sum_{t=1}^T (V^{*,\nu^t}_{f^*} - V^*_{f^*})] \leq 6\sqrt{d_{\mathrm{GEC}}HT \cdot [\omega(4HT, p^0) + \omega(4HT, q^0)]}.$$

Combining the results of Propositions 1 and 2, due to $\mathrm{Reg}^{\mathrm{sp}}(T) = \mathrm{Reg}^{\mathrm{sp}}_1(T) + \mathrm{Reg}^{\mathrm{sp}}_2(T)$, we obtain the following overall regret when running Algorithms 1 and 3 together.

**Theorem 1.** *Under the settings of Propositions 1 and 2, executing both Algorithms 1 and 3 leads to*

$$\mathbb{E}[\mathrm{Reg}^{\mathrm{sp}}(T)] \leq 12\sqrt{d_{\mathrm{GEC}}HT \cdot [\omega(4HT, p^0) + \omega(4HT, q^0)]}.$$

The above results indicate that the proposed posterior sampling self-play algorithms (Algorithms 1 and 3) separately admit a sublinear dependence on GEC $d_{\mathrm{GEC}}$, the number of learning episodes $T$, as well as $\omega(4HT, p^0)$ and $\omega(4HT, q^0)$ for both FOMG and POMG settings. They lead to the same overall regret bound combining Propositions 1 and 2. In particular, when $\mathcal{F}$ is finite with $p^0 = q^0 = \mathrm{Unif}(\mathcal{F})$, Algorithms 1 and 3 admit regrets of $O(\sqrt{d_{\mathrm{GEC}}HT \cdot \log|\mathcal{F}|})$. The quantity $\omega$ can be associated with the log-covering number if $\mathcal{F}$ is infinite. Please see Appendix C for analysis.

## 4 Posterior Sampling for the Adversarial Setting

Without loss of generality, we assume that the max-player is the main agent and the min-player is the opponent. Under this setting, the goal of the main player is to maximize her cumulative rewards as much as possible, comparing against the value under the NE, i.e., $V^*_{f^*}$. We develop a novel algorithm for this setting as summarized in Algorithm 2. In our algorithm, the opponent's policy is assumed to be arbitrary and is also *not revealed* to the main player. The only information about the opponent is the current state or the partial observation of her state as well as the actions taken.

We adopt the optimistic posterior sampling approach for the main player with defining an optimism term as $\gamma V^*_f$ motivated by the above learning target, and the likelihood function $L^t_h(f)$ with $L^t_h(f) := \eta \log \mathbb{P}_{f,h}(s^t_{h+1} \mid s^t_h, a^t_h, b^t_h)$ in (1) for FOMGs and $L^t_h(f) = \eta \log \mathbf{P}_{f,h}(\tau^t_h)$ in (2) for POMGs

---

**Algorithm 2** Model-Based Posterior Sampling with Adversarial Opponent

---

1: **Input:** Model class $\mathcal{F}$, prior distributions $p^0$, and $\gamma$.
2: **for** $t = 1, \ldots, T$ **do**
3:     Draw a model $f^t \sim p^t(\cdot)$ with defining $p^t(f) \propto p^0(f) \exp[\gamma V_f^* + \sum_{\tau=1}^{t-1} \sum_{h=1}^{H} L_h^\tau(f)]$
4:     Compute $\pi^t$ by letting $(\pi^t, \overline{\nu}^t)$ be NE of $V_{f^t}^{\pi,\nu}$
5:     Opponent picks an arbitrary policy $\nu^t$
6:     Collect a trajectory $\mathcal{D}^t$ by executing the joint exploration policy $\sigma^t$
7:     Define the likelihood functions $\{L_h^t(f)\}_{h=1}^H$ using the collected data $\mathcal{D}^t$
8: **end for**
9: **Return:** $(\pi^1, \ldots, \pi^T)$.

---

respectively. The policy $\pi^t$ learned by the main player is from computing the NE of the value function under the current model $f^t$ sampled from the posterior distribution $p^t$. In addition, the joint exploration policy is set to be $\sigma^t = (\pi^t, \nu^t)$ where $\nu^t$ is the potentially adversarial policy played by the opponent. Thus, we can collect the data defined as $\mathcal{D}^t = \{(s_h^t, a_h^t, b_h^t, s_{h+1}^t)\}_{h=1}^H$ and $\mathcal{D}^t = \{\tau_h^t\}_{h=1}^H$ with $\tau_h^t = (o_1^t, a_1^t, b_1^t \ldots, o_h^t, a_h^t, b_h^t)$ for FOMGs and POMGs respectively, collected by executing $\sigma^t$ to the $h$-th step of the game for each $h \in [H]$.

**Remark 1.** *In Algorithm 2, we define the joint exploration policy $\sigma^t = (\pi^t, \nu^t)$, which is the key to the success of the algorithm design under the adversarial setting, especially for POMGs. Under the single-agent setting, the prior work [83] sets the exploration policy for a range of partially observable models subsumed by the PSR model as $\pi_{1:h-1}^t \circ_h \mathrm{Unif}(\mathcal{A})$, i.e., running $\pi^t$ for steps $1$ to $h-1$ and then sampling the data at step $h$ by enforcing a uniform policy. Such an exploration scheme fails to work when facing an uncontrollable opponent who does not play a uniform policy at step $h$. Theoretically, we prove that employing policies $(\pi_{1:h}^t, \nu_{1:h}^t)$ for exploration without the uniform policy, the self-play and adversarial GEC conditions in Definitions 1 and 3 are still satisfied for a class of POMGs including weakly revealing and decodable POMGs. This eventually leads to a unified adversarial learning algorithm for both FOMGs and POMGs.*

### 4.1 Regret Analysis for the Adversarial Setting

Before demonstrating our regret analysis, we first define a multi-agent GEC fitting the adversarial learning scenario. Considering that the opponent's policy is uncontrollable during the learning, we let $\{\nu^t\}_{t=1}^T$ be arbitrary, which is clearly distinguished from self-play GEC defined in Definition 1.

**Definition 3** (Adversarial GEC)**.** *For any sequence of functions $\{f^t\}_{t=1}^T$ with $f^t \in \mathcal{F}$ and any sequence of the opponent's policies $\{\nu^t\}_{t=1}^T$, suppose that the main player's policies $\{\mu^t\}_{t=1}^T$ are generated via $\mu^t = \arg\max_\pi \min_\nu V_{f^t}^{\pi,\nu}$. Denoting the joint exploration policy as $\{\sigma^t\}_{t=1}^T$ depending on $\{f^t\}_{t=1}^T$, the adversarial GEC $d_{\mathrm{GEC}}$ is defined as the minimal constant $d$ satisfying*

$$\sum_{t=1}^T \left( V_{f^t}^{\pi^t,\nu^t} - V_{f^*}^{\pi^t,\nu^t} \right) \le \left[ d \sum_{h=1}^H \sum_{t=1}^T \left( \sum_{\tau=1}^{t-1} \mathbb{E}_{(\sigma^\tau, h)} \ell(f^t, \xi_h^\tau) \right) \right]^{\frac{1}{2}} + 2H(dHT)^{\frac{1}{2}} + \epsilon HT.$$

Our regret analysis for Algorithm 2 also depends on the quantity $\omega(\beta, p^0)$ that characterizes the coverage of the prior distribution $p^0$ on the true model $f^*$. Then, we have the following regret bound.

**Theorem 2.** *Letting $\eta = \frac{1}{2}$, $\gamma = 2\sqrt{\omega(4HT, p^0)T/d_{\mathrm{GEC}}}$, $\epsilon = 1/\sqrt{HT}$ in Definition 3, when $T \ge \max\{4H^2\omega(4HT, p^0)/d_{\mathrm{GEC}}, d_{\mathrm{GEC}}/H\}$, under both FOMG and POMG settings, Algorithm 2 admits the following regret bound,*

$$\mathbb{E}[\mathrm{Reg}^{\mathrm{adv}}(T)] \le 4\sqrt{d_{\mathrm{GEC}} HT \cdot \omega(4HT, p^0)}.$$

The above result indicates that we can achieve a meaningful regret bound by a posterior sampling algorithm with general function approximation, even when the opponent's policy is adversarial and her full policies $\nu^t$ are not revealed. This regret has a sublinear dependence on $d_{\mathrm{GEC}}$, the number of episodes $T$, as well as $\omega(4HT, p^0)$. Similarly, when $\mathcal{F}$ is finite, Algorithm 2 admits a regret of $O(\sqrt{d_{\mathrm{GEC}} HT \cdot \log |\mathcal{F}|})$. The term $\log |\mathcal{F}|$ can be the log-covering number of $\mathcal{F}$ if it is infinite.

# 5 Theoretical Analysis

This section presents several examples of tractable MG classes captured by the self-play and adversarial GEC, the discussion of the quantity $\omega(\beta, p^0)$, the proof sketches of main theorems, and the discussion of limitations.

**Examples.** We call the class with a low $d_{\mathrm{GEC}}$ the *low self-play GEC class* and *low adversarial GEC class*. Next, we analyze the relation between the proposed classes and the following MG classes. We also propose a new decodable POMG class generalized from the single-agent POMDP. We note that except for the linear MG, the other classes cannot be analyzed by the recent posterior sampling work [73]. We defer detailed definitions and proofs to Appendix B.

- **Linear MG.** This FOMG class admits a linear structure of the reward and transition by feature vectors $\boldsymbol{\phi}(s, a, b) \in \mathbb{R}^d$ as $r_h(s, a, b) = \mathbf{w}_h^\top \boldsymbol{\phi}(s, a, b)$ and $\mathbb{P}_h(s'|s, a, b) = \boldsymbol{\theta}_h(s')^\top \boldsymbol{\phi}(s, a, b)$ [71]. We then prove that "*linear MG $\subset$ low self-play/adversarial GEC*" with $d_{\mathrm{GEC}} = \widetilde{O}(H^3 d)$.
- **Linear Mixture MG.** This FOMG class admits a different type of the linear structure for the transition [16] as $\mathbb{P}_h(s'|s, a, b) = \boldsymbol{\theta}_h^\top \boldsymbol{\phi}(s, a, b, s')$ with $\boldsymbol{\phi}(s, a, b, s') \in \mathbb{R}^d$. We prove that "*linear mixture MG $\subset$ low self-play/adversarial GEC*" with $d_{\mathrm{GEC}} = \widetilde{O}(H^3 d)$.
- **Low Self-Play Witness Rank.** [28] defines this FOMG class for self-play by supposing that an inner product of specific vectors in $\mathbb{R}^d$ defined on current models can lower bound witnessed model misfit and upper bound the Bellman error with a coefficient $\kappa_{\mathrm{wit}}$, which generalizes linear/linear mixture MGs. We can prove "*low self-play witness rank $\subset$ low self-play GEC*" with $d_{\mathrm{GEC}} = \widetilde{O}(H^3 d / \kappa_{\mathrm{wit}}^2)$.
- **$\alpha$-Weakly Revealing POMG.** This POMG class assumes $\min_h \sigma_S(\mathbb{O}_h) \geq \alpha$ where $\mathbb{O}_h \in \mathbb{R}^{|\mathcal{O}| \times |\mathcal{S}|}$ is the matrix by $\mathbb{O}_h(\cdot|\cdot)$ and $\sigma_S$ is the $S$-th singular value [41]. We prove that "*$\alpha$-weakly revealing POMG $\subset$ low self-play/adversarial GEC*" with $d_{\mathrm{GEC}} = \widetilde{O}(H^3 |\mathcal{O}|^3 |\mathcal{A}|^2 |\mathcal{B}|^2 |\mathcal{S}|^2 / \alpha^2)$.
- **Decodable POMG.** We propose decodable POMGs by generalizing decodable POMDPs [23, 22], assuming that an unknown decoder $\phi_h$ recovers states from observations, i.e., $\phi_h(o) = s$. We can prove "*decodable POMG $\subset$ low self-play/adversarial GEC*" with $d_{\mathrm{GEC}} = \widetilde{O}(H^3 |\mathcal{O}|^3 |\mathcal{A}|^2 |\mathcal{B}|^2)$.

**Discussion of** $\omega(\beta, p^0)$**.** We briefly discuss the upper bound of the quantity $\omega(\beta, p^0)$ for FOMGs and POMGs. We refer readers to Appendix C for more detailed proofs. For FOMGs, according to Lemma 2 of [2], when $\mathcal{F}$ is finite, $p^0 = \mathrm{Unif}(\mathcal{F})$, then $\omega(\beta, p^0) \leq \log |\mathcal{F}|$ by its definition. When $\mathcal{F}$ is infinite, it shows that under mild conditions, there exists a prior $p^0$ over $\mathcal{F}$, $B \geq \log(6B^2/\epsilon)$, and $\nu = \epsilon/(6 \log(6B^2/\epsilon))$ such that $\omega(\beta, p^0) \leq \beta\epsilon + \log(\mathcal{N}(\frac{\epsilon}{6 \log(B/\nu)}))$, where $\mathcal{N}(\epsilon)$ is the $\epsilon$-covering number w.r.t. the distance $d(f, f') := \sup_{s,a,b,h} |D_{\mathrm{He}}^2(\mathbb{P}_{f,h}(\cdot \,|\, s, a, b), \mathbb{P}_{f^*,h}(\cdot \,|\, s, a, b)) - D_{\mathrm{He}}^2(\mathbb{P}_{f',h}(\cdot \,|\, s, a, b), \mathbb{P}_{f^*,h}(\cdot \,|\, s, a, b))|$. Since we have $|D_{\mathrm{He}}^2(P, R) - D_{\mathrm{He}}^2(Q, R)| \leq \frac{\sqrt{2}}{2} \|P - Q\|_1$ for any distributions $P, Q$, and $R$, the covering number w.r.t. the distance $d$ can connect to the more common covering number w.r.t. the $\ell_1$ distance. Thus, the upper bound of $\omega(\beta, p^0)$ can be calculated for different cases. Additionally, for POMGs, inspired by [2], our work proves that under similar conditions, $\omega(\beta, p^0)$ with finite and infinite $\mathcal{F}$ admit the same bounds as those for FOMGs. The difference is that the covering number is w.r.t. the distance $d(f, f') = \sup_{\pi,\nu} |D_{\mathrm{He}}^2(\mathbf{P}_{f,H}^{\pi,\nu}, \mathbf{P}_{f^*,H}^{\pi,\nu}) - D_{\mathrm{He}}^2(\mathbf{P}_{f',H}^{\pi,\nu}, \mathbf{P}_{f^*,H}^{\pi,\nu})|$, which further connects to the $\ell_1$ distance defined as $d_1(f, f') := \sup_{\pi,\nu} \|\mathbf{P}_{f,H}^{\pi,\nu} - \mathbf{P}_{f',H}^{\pi,\nu}\|_1$. Such a covering number under $\ell_1$ distance is further analyzed in [78]. Our work gives the first detailed proof for the upper bound of $\omega(\beta, p^0)$ under the partially observable setting, which is thus of independent interest.

Next, we outline our proof sketches. Detailed proofs are deferred to Appendices D, E, and F.

**Proof Sketch of Theorem 1.** To prove Theorem 1, we only need to combine the result in Propositions 1 and 2 via $\mathbb{E}[\mathrm{Reg}^{\mathrm{sp}}(T)] = \mathbb{E}[\mathrm{Reg}_1^{\mathrm{sp}}(T) + \mathrm{Reg}_2^{\mathrm{sp}}(T)]$. We thus first give a proof sketch for Proposition 1. We decompose $\mathrm{Reg}_1^{\mathrm{sp}}(T) = \mathrm{Term(i)} + \mathrm{Term(ii)}$ where

$$\mathrm{Term(i)} = \sum_{t=1}^T \left[ V_{f^*}^* - V_{f^*}^{\pi^t, \nu^t} \right], \quad \mathrm{Term(ii)} = \sum_{t=1}^T \left[ V_{f^*}^{\pi^t, \nu^t} - V_{f^*}^{\pi^t, *} \right].$$

Intuitively, $\mathbb{E}[\mathrm{Term(i)}]$ is the main player's regret incurred Line 4 of Algorithm 1 and $\mathbb{E}[\mathrm{Term(ii)}]$ is the exploiter's regret incurred by Line 6. We further show

$$\mathrm{Term(i)} \leq \sum_{t=1}^T \left[ -\Delta V_{\overline{f}^t}^* + V_{\overline{f}^t}^{\pi^t, \nu^t} - V_{f^*}^{\pi^t, \nu^t} \right], \quad \mathrm{Term(ii)} = \sum_{t=1}^T \left[ V_{f^*}^{\pi^t, \nu^t} - V_{\underline{f}^t}^{\pi^t, \nu^t} + \Delta V_{\underline{f}^t}^{\pi^t, *} \right],$$

where $\Delta V^*_{\overline{f}^t} := V^{\pi^t,\overline{\nu}^t}_{\overline{f}^t} - V^*_{f^*}$ and $\Delta V^{\pi^t,*}_{\underline{f}^t} = V^{\pi^t,\underline{\nu}^t}_{\underline{f}^t} - V^{\pi^t,*}_{f^*}$ are associated with the optimism terms in posterior distributions. The inequality above for Term(i) is due to Line 4 such that $V^{\pi^t,\overline{\nu}^t}_{\overline{f}^t} = \min_\nu V^{\pi^t,\nu}_{\overline{f}^t} \leq V^{\pi^t,\underline{\nu}^t}_{\overline{f}^t}$. By Definition 1 for self-play GEC, we obtain that $\sum_{t=1}^T \left( V^{\pi^t,\underline{\nu}^t}_{\overline{f}^t,1} - V^{\pi^t,\underline{\nu}^t}_{f^*} \right)$ and $\sum_{t=1}^T \left( V^{\pi^t,\underline{\nu}^t}_{f^*} - V^{\pi^t,\underline{\nu}^t}_{\underline{f}^t,1} \right)$ can be bounded by

$$\left[ d_{\mathrm{GEC}} \sum_{h=1}^H \sum_{t=1}^T \left( \sum_{\iota=1}^{t-1} \mathbb{E}_{(\sigma^\iota_{\exp},h)} \ell(\rho^t,\xi^\iota_h) \right) \right]^{1/2} + 2H(d_{\mathrm{GEC}} HT)^{\frac{1}{2}} + \epsilon HT,$$

where $\rho^t$ is chosen as $\overline{f}^t$ or $\underline{f}^t$ respectively. By Lemma 11 and Lemma 12, we prove that for both FOMGs and POMGs, the accumulation of the losses $\ell(\overline{f}^t,\xi^\iota_h)$ in (3) connects to the likelihood function $L^t_h$ defined in (1) and (2). Thus, we obtain $\mathbb{E}[\text{Term(i)}] \leq \sum_{t=1}^T \mathbb{E}_{Z^{t-1}} \mathbb{E}_{\overline{f}^t \sim p^t} \{ -\gamma_1 \Delta V^*_{\overline{f}^t} - \sum_{h=1}^H \sum_{\iota=1}^{t-1} [L^t_h(\overline{f}^t) - L^t_h(f^*)] + \log \frac{p^t(\overline{f}^t)}{p^0(\overline{f}^t)} \} + 2H(d_{\mathrm{GEC}} HT)^{\frac{1}{2}} + \epsilon HT$ and $\mathbb{E}[\text{Term(ii)}]$ has a similar bound based on $q^t$, where $Z^{t-1}$ is the randomness history. By Lemma 10, the posterior distributions $p^t$ and $q^t$ following Lines 3 and 5 of Algorithm 1 can minimize the above upper bounds for $\mathbb{E}[\text{Term(i)}]$ and $\mathbb{E}[\text{Term(ii)}]$. Therefore, we can relax $p^t$ and $q^t$ to be distributions defined around the true model $f^*$ to enlarge above bounds. When $T$ is sufficiently large and $\eta = 1/2$, we have

$$\mathbb{E}[\text{Term(i)}] \leq \omega(HT, p^0)T/\gamma_1 + \gamma_1 d_{\mathrm{GEC}} H/4 + 2H(d_{\mathrm{GEC}} HT)^{\frac{1}{2}} + \epsilon HT,$$
$$\mathbb{E}[\text{Term(ii)}] \leq \omega(HT, q^0)T/\gamma_2 + \gamma_2 d_{\mathrm{GEC}} H/4 + 2H(d_{\mathrm{GEC}} HT)^{\frac{1}{2}} + \epsilon HT.$$

Choosing proper values for $\epsilon, \gamma_1$, and $\gamma_2$, we obtain the bound for $\mathbb{E}[\text{Reg}^{\mathrm{sp}}_1(T)]$ in Theorem 1 via $\text{Reg}^{\mathrm{sp}}_1(T) = \text{Term(i)} + \text{Term(ii)}$. In addition, we can prove the bound of $\mathbb{E}[\text{Reg}^{\mathrm{sp}}_2(T)]$ in a symmetric manner. Finally, combining $\mathbb{E}[\text{Reg}^{\mathrm{sp}}_1(T)]$ and $\mathbb{E}[\text{Reg}^{\mathrm{sp}}_2(T)]$ gives the result in Theorem 1.

**Proof Sketch of Theorem 2.** Under the adversarial setting, the policy of the opponent $\nu^t$ is not generated by the algorithm, which could be arbitrarily time-varying. We decompose $\text{Reg}^{\mathrm{adv}}(T) = \sum_{t=1}^T \Delta V^*_{f^t} + \sum_{t=1}^T [V^*_{f^t} - V^{\pi^t,\nu^t}_{f^*}]$ where $\Delta V^*_{f^t} := V^*_{f^*} - V^*_{f^t}$ relates to optimism. Since $(\pi^t, \overline{\nu}^t)$ is NE of $V^{\pi,\nu}_{f^t}$ as in Line 3 of Algorithm 2, we have $V^*_{f^t} = \min_\nu V^{\pi^t,\nu}_{f^t} \leq V^{\pi^t,\nu^t}_{f^t}$, which leads to

$$\text{Reg}^{\mathrm{adv}}(T) \leq \sum_{t=1}^T \Delta V^*_{f^t} + \sum_{t=1}^T \left[ V^{\pi^t,\nu^t}_{f^t} - V^{\pi^t,\nu^t}_{f^*} \right].$$

We can bound $\sum_{t=1}^T [V^{\pi^t,\nu^t}_{f^t} - V^{\pi^t,\nu^t}_{f^*}]$ via adversarial GEC in Definition 3 by

$$\left[ d_{\mathrm{GEC}} \sum_{h=1}^H \sum_{t=1}^T \left( \sum_{\iota=1}^{t-1} \mathbb{E}_{(\sigma^\iota_{\exp},h)} \ell(f^t,\xi^\iota_h) \right) \right]^{\frac{1}{2}} + 2H(d_{\mathrm{GEC}} HT)^{\frac{1}{2}} + \epsilon HT.$$

Connecting the loss $\ell(\overline{f}^t, \xi^\iota_h)$ to the likelihood function $L^t_h$ defined in (1) and (2) via Lemmas 11 and 12, we obtain $\mathbb{E}[\text{Reg}^{\mathrm{adv}}(T)] \leq \sum_{t=1}^T \mathbb{E}_{Z^{t-1}} \mathbb{E}_{f^t \sim p^t} \{ \gamma \sum_{t=1}^T \Delta V^*_{f^t} - \sum_{h=1}^H \sum_{\iota=1}^{t-1} [L^t_h(f^t) - L^t_h(f^*)] + \log \frac{p^t(f^t)}{p^0(f^t)} \} + 2H(d_{\mathrm{GEC}} HT)^{\frac{1}{2}} + \epsilon HT$. Lemma 10 shows $p^t$ in Line 3 of Algorithm 2 can minimize this bound. Thus, relaxing $p^t$ to be distribution defined around the true model $f^*$, with sufficiently large $T$ and $\eta = 1/2$, we have

$$\mathbb{E}[\text{Reg}^{\mathrm{adv}}(T)] \leq \omega(4HT, p^0)T/\gamma + \gamma d_{\mathrm{GEC}} H/4 + 2H(d_{\mathrm{GEC}} HT)^{\frac{1}{2}} + \epsilon HT.$$

Choosing proper values for $\epsilon$ and $\gamma$, we eventually obtain the bound for $\mathbb{E}[\text{Reg}^{\mathrm{adv}}(T)]$ in Theorem 2.

**Discussion of Limitations.** Our work has studied several but a limited number of tractable MG classes in both FOMGs and POMGs. It is interesting to further define new MG classes by generalizing their single-agent counterparts and explore the relation between these MG classes and low self-play/adversarial GEC classes. It is also intriguing to generalize our method to general-sum settings.

## Acknowledgments and Disclosure of Funding

The authors would like to thank the anonymous reviewers for their valuable comments. The authors would like to thank Wei Xiong for the helpful discussions. Shuang Qiu and Tong Zhang acknowledge the funding supported by the GRF 16310222.

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

# Appendix

## Contents

# A Omitted Algorithm in the Main Text

---

**Algorithm 3** Model-Based Posterior Sampling for Self-Play (Min-Player)

---

1: **Input:** Model class $\mathcal{F}$, prior distributions $p^0$ and $q^0$, $\gamma_1$, and $\gamma_2$.
2: **for** $t = 1, \ldots, T$ **do**
3:      Draw a model $\overline{f}^t \sim p^t(\cdot)$ with defining $p^t(f) \propto p^0(f) \exp[-\gamma_1 V_f^* + \sum_{\iota=1}^{t-1} \sum_{h=1}^{H} L_h^\tau(f)]$
4:      Compute $\nu^t$ by letting $(\overline{\pi}^t, \nu^t)$ be NE of $V_{\overline{f}^t}^{\pi, \nu}$
5:      Draw a model $\underline{f}^t \sim q^t(\cdot)$ with defining $q^t(f) \propto q^0(f) \exp[\gamma_2 V_f^{*, \nu^t} + \sum_{\iota=1}^{t-1} \sum_{h=1}^{H} L_h^\tau(f)]$
6:      Compute $\underline{\pi}^t$ by letting $\underline{\pi}^t$ be the best response of $\nu^t$ w.r.t. $V_{\underline{f}^t}^{\pi, \nu}$
7:      Collect data $\mathcal{D}^t$ by executing the joint exploration policy $\sigma^t$
8:      Define the likelihood functions $\{L_h^t(f)\}_{h=1}^{H}$ using the collected data $\mathcal{D}^t$
9: **end for**
10: **Return:** $(\nu^1, \ldots, \nu^T)$.

---

# B Computation of Self-Play and Adversarial GEC

In this section, we present the computation of the self-play and adversarial GEC for different FOMG and POMG classes, including linear MGs, linear mixture MGs, low-witness-rank MGs, weakly revealing POMGs, and a novel POMG class, dubbed decodable POMGs, newly proposed by this work.

Note that the GEC $d_{\mathrm{GEC}}$ essentially depends on $\epsilon$ as shown in Definition 1 and Definition 3 such that it should be expressed as $d_{\mathrm{GEC}}^\epsilon$. For convenience, we omit such a dependence in the notation. In our main theorems, we set $\epsilon = 1/\sqrt{HT}$, and according to our derivation in this section, $d_{\mathrm{GEC}}$ has an $O(\log(1/\epsilon)) = O(\cdot \log(HT))$ factor which only scales logarithmically in $T$.

## B.1 Linear MG

In this subsection, we show that the linear MG is in the class of MGs with low self-play and adversarial GEC and further calculate $d_{\mathrm{GEC}}$ for the linear MG.

**Definition 4** (Linear Markov Game [71]). *The linear MG is a FOMG admitting the following linear structures on the reward function and transition kernel,*

$$r_h(s, a, b) = \mathbf{w}_h^\top \phi(s, a, b), \quad \mathbb{P}_h(s' \mid s, a, b) = \boldsymbol{\theta}_h(s')^\top \phi(s, a, b),$$

*where there exist a known feature map $\phi : \mathcal{S} \times \mathcal{A} \times \mathcal{B} \mapsto \mathbb{R}^d$ and unknown coefficients $\mathbf{w}_h \in \mathbb{R}^d$ and $\boldsymbol{\theta}_h(s') \in \mathbb{R}^d$ that should be learned in algorithms. We assume that the feature map and the coefficients satisfy $\|\phi(s, a, b)\|_2 \le 1$, $\int_{\mathcal{S}} \|\boldsymbol{\theta}_h(s)\|_2 \mathrm{d}s \le \sqrt{d}$, and $\|\mathbf{w}_h\|_2 \le \sqrt{d}$.*

Then, $d_{\mathrm{GEC}}$ for the linear MG can be calculated in the following proposition. We can show that both low self-play and adversarial GEC classes subsume the class of linear MGs.

**Proposition 3** (Linear MG $\subset$ Low Self-Play/Adversarial GEC). *For linear MGs with $\phi(s, a, b) \in \mathbb{R}^d$ as defined in Definition 4, and for $\epsilon > 0$, when $T$ is sufficiently large, choosing $\ell(f, \xi_h) = D_{\mathrm{He}}^2\big(\mathbb{P}_{f,h}(\cdot \mid \xi_h), \mathbb{P}_{f^*,h}(\cdot \mid \xi_h)\big)$ as in (3) with $\xi_h = (s_h, a_h, b_h)$, we have*

$$\text{linear MG} \subset \text{MG with low self-play and adversarial GEC}$$

*with $d_{\mathrm{GEC}}$ satisfying*

$$d_{\mathrm{GEC}} = 16H^3 d \log\left(1 + \frac{HT}{\epsilon}\right).$$

*Proof.* Since the value function is defined as $V_{f,h}^{\pi, \nu}(s_h) := \mathbb{E}_{\pi, \nu, \mathbb{P}_f}[\sum_{h'=h}^{H} r_{h'}(s_{h'}, a_{h'}, b_{h'}) | s_h]$ at step $h$ with denoting $V_f^{\pi, \nu} = V_{f,1}^{\pi, \nu}(s_1)$ and we also have a relation $V_f^{\pi, \nu} =$

$\mathbb{E}_{\pi,\nu,\mathbb{P}_{f'}}[\sum_{h=1}^H (V_{f,h}^{\pi,\nu}(s_h) - V_{f,h+1}^{\pi,\nu}(s_{h+1}))]$ for any $f'$ using the telescoping summation as well as $V_{f,H+1}^{\pi,\nu}(\cdot) = \mathbf{0}$, we thus can decompose the value difference as follows,

$$V_{f^t}^{\pi^t,\nu^t} - V_{f^*}^{\pi^t,\nu^t}$$

$$= \mathbb{E}_{\pi^t,\nu^t,\mathbb{P}_{f^*}}\left[\sum_{h=1}^H (V_{f^t,h}^{\pi^t,\nu^t}(s_h) - V_{f^t,h+1}^{\pi^t,\nu^t}(s_{h+1}))\right] - \mathbb{E}_{\pi^t,\nu^t,\mathbb{P}_{f^*}}\left[\sum_{h=1}^H r_h(s_h, a_h, b_h)\right]$$

$$= \sum_{h=1}^H \mathbb{E}_{\pi^t,\nu^t,\mathbb{P}_{f^*}}\left[V_{f^t,h}^{\pi^t,\nu^t}(s_h) - \left(r_h(s_h, a_h, b_h) + V_{f^t,h+1}^{\pi^t,\nu^t}(s_{h+1})\right)\right]$$

$$= \sum_{h=1}^H \mathcal{E}_{f^t,h}^{\pi^t,\nu^t}, \tag{4}$$

where we define the following Bellman residual in expectation as

$$\mathcal{E}_{f,h}^{\pi,\nu} := \mathbb{E}_{\pi,\nu,\mathbb{P}_{f^*}}\left[Q_{f,h}^{\pi,\nu}(s_h, a_h, b_h) - \mathbb{E}_{s_{h+1}\sim\mathbb{P}_{f^*,h}(\cdot|s_h,a_h,b_h)}\left(r_h(s_h, a_h, b_h) + V_{f,h+1}^{\pi,\nu}(s_{h+1})\right)\right].$$

The Bellman residual for the linear MG can be linearly represented, whose form is

$$\mathcal{E}_{f,h}^{\pi,\nu} = \mathbb{E}_{\pi,\nu,\mathbb{P}_{f^*}}\int_{\mathcal{S}}[\mathbb{P}_{f,h}(s|s_h,a_h,b_h) - \mathbb{P}_{f^*,h}(s|s_h,a_h,b_h)]V_{f,h+1}^{\pi,\nu}(s)\mathrm{d}s$$

$$= \mathbb{E}_{\pi,\nu,\mathbb{P}_{f^*}}\int_{\mathcal{S}}[\boldsymbol{\phi}(s_h,a_h,b_h)^\top(\boldsymbol{\theta}_{f,h}(s) - \boldsymbol{\theta}_{f^*,h}(s))]V_{f,h+1}^{\pi,\nu}(s)\mathrm{d}s$$

$$= \mathbb{E}_{\pi,\nu,\mathbb{P}_{f^*}}\boldsymbol{\phi}(s_h,a_h,b_h)^\top\int_{\mathcal{S}}[(\boldsymbol{\theta}_{f,h}(s) - \boldsymbol{\theta}_{f^*,h}(s))]V_{f,h+1}^{\pi,\nu}(s)\mathrm{d}s$$

$$= \langle\mathbb{E}_{\pi,\nu,\mathbb{P}_{f^*}}\boldsymbol{\phi}(s_h,a_h,b_h), \boldsymbol{q}_h(f,\pi,\nu)\rangle,$$

where we let

$$\boldsymbol{q}_h(f,\pi,\nu) := \int_{\mathcal{S}}[(\boldsymbol{\theta}_{f,h}(s) - \boldsymbol{\theta}_{f^*,h}(s))]V_{f,h+1}^{\pi,\nu}(s)\mathrm{d}s,$$

such that we have $\|\boldsymbol{q}_h(f,\pi,\nu)\|_2 \leq 2\sqrt{d}H$ according to the definition of the linear MG. Therefore, the value difference term can also be rewritten as

$$V_{f^t}^{\pi^t,\nu^t} - V_{f^*}^{\pi^t,\nu^t} = \sum_{h=1}^H \mathcal{E}_{f^t,h}^{\pi^t,\nu^t} = \sum_{h=1}^H[\langle\mathbb{E}_{\pi^t,\nu^t,\mathbb{P}_{f^*}}\boldsymbol{\phi}(s_h^t, a_h^t, b_h^t), \boldsymbol{q}_h(f^t,\pi^t,\nu^t)\rangle],$$

where $(s_h^t, a_h^t, b_h^t)$ can be viewed as sampled data following $\pi^t, \nu^t$, and $\mathbb{P}_{f^*}$.

For simplicity, we denote $\boldsymbol{\phi}_h^t := \mathbb{E}_{\pi^t,\nu^t,\mathbb{P}_{f^*}}\boldsymbol{\phi}(s_h^t, a_h^t, b_h^t)$, $\boldsymbol{q}_h^t := \boldsymbol{q}_h(f^t,\pi^t,\nu^t)$, and $\xi_h^t := (s_h^t, a_h^t, b_h^t)$, and define $\Phi_h^t := \lambda I_d + \sum_{\iota=1}^t \boldsymbol{\phi}_h^t(\boldsymbol{\phi}_h^t)^\top$ with $\lambda > 0$ and $\varphi_h^t := \|\boldsymbol{\phi}_h^t\|_{(\Phi_h^{t-1})^{-1}}$. Note that we have

$$\|\boldsymbol{q}_h^t\|_{\Phi_h^{t-1}} = \sqrt{\sum_{\iota=1}^{t-1}\langle\boldsymbol{\phi}_h^t, \boldsymbol{q}_h^t\rangle^2 + \lambda\|\boldsymbol{q}_h^t\|_2^2}, \tag{5}$$

where the term $\langle\boldsymbol{\phi}_h^t, \boldsymbol{q}_h^t\rangle^2$ can be bounded as

$$[\langle\boldsymbol{\phi}_h^\iota, \boldsymbol{q}_h^t\rangle]^2 = \left[\mathbb{E}_{\pi^\iota,\nu^\iota,f^*}\int_{\mathcal{S}}(\mathbb{P}_{f^t,h}(s\,|\,s_h, a_h, b_h) - \mathbb{P}_{f^*,h}(s\,|\,s_h, a_h, b_h))V_{f^t,h+1}^{\pi^t,\nu^t}(s)\mathrm{d}s\right]^2$$

$$\leq H^2\mathbb{E}_{\pi^\iota,\nu^\iota,f^*}\|\mathbb{P}_{f,h}(\cdot|s_h, a_h, b_h) - \mathbb{P}_{f^*,h}(\cdot|s_h, a_h, b_h)\|_1^2$$

$$\leq 8H^2\mathbb{E}_{\pi^\iota,\nu^\iota,f^*}D_{\mathrm{He}}^2(\mathbb{P}_{f^t,h}(\cdot|s_h, a_h, b_h), \mathbb{P}_{f^*,h}(\cdot|s_h, a_h, b_h)), \tag{6}$$

where the first inequality uses $V_{f,h+1}^{\pi,\nu}(s) \leq H$ and the second inequality uses $\|P - Q\|_1^2 \leq 8D_{\mathrm{He}}^2(P, Q)$. Moreover, using $\boldsymbol{a}^\top\boldsymbol{b} \leq \|\boldsymbol{a}\|_A\|\boldsymbol{b}\|_{A^{-1}}$, we also have

$$\langle\boldsymbol{\phi}_h^t, \boldsymbol{q}_h^t\rangle = \langle\boldsymbol{\phi}_h^t, \boldsymbol{q}_h^t\rangle(\mathbf{1}_{\{\varphi_h^t<1\}} + \mathbf{1}_{\{\varphi_h^t\geq1\}})$$

$$\leq H\min\left\{\frac{1}{H}|\langle\boldsymbol{\phi}_h^t, \boldsymbol{q}_h^t\rangle|, 1\right\}\mathbf{1}_{\{\varphi_h^t<1\}} + H\cdot\mathbf{1}_{\{\varphi_h^t\geq1\}}$$

$$\leq \|\boldsymbol{q}_h^t\|_{\Phi_h^{t-1}}\min\{\varphi_h^t, 1\}\mathbf{1}_{\{\varphi_h^t<1\}} + H\cdot\mathbf{1}_{\{\varphi_h^t\geq1\}}, \tag{7}$$

where $\mathbf{1}_{\{.\}}$ denotes an indicator function.

Plugging (5) into (7) and taking a summation from $t = 1$ to $T$, we obtain

$$\sum_{h=1}^{H}\sum_{t=1}^{T}\|\boldsymbol{q}_h^t\|_{\Phi_h^{t-1}}\min\{\varphi_h^t,1\}\mathbf{1}_{\{\varphi_h^t<1\}}+H\cdot\mathbf{1}_{\{\varphi_h^t\geq1\}}$$

$$\leq\sum_{h=1}^{H}\sum_{t=1}^{T}\sqrt{\sum_{\iota=1}^{t-1}[\langle\boldsymbol{\phi}_h^\iota,\boldsymbol{q}_h^t\rangle]^2+4\lambda dH^2}\cdot\min\{\varphi_h^t,1\}\mathbf{1}_{\{\varphi_h^t<1\}}+H\cdot\mathbf{1}_{\{\varphi_h^t\geq1\}}$$

$$\leq\sum_{h=1}^{H}\sum_{t=1}^{T}\left(\sqrt{\sum_{\iota=1}^{t-1}[\langle\boldsymbol{\phi}_h^\iota,\boldsymbol{q}_h^t\rangle]^2}+\sqrt{4\lambda dH^2}\right)\cdot\min\{\varphi_h^t,1\}\mathbf{1}_{\{\varphi_h^t<1\}}+H\cdot\mathbf{1}_{\{\varphi_h^t\geq1\}}$$

$$\leq\sqrt{\sum_{h=1}^{H}\sum_{t=1}^{T}\sum_{\iota=1}^{t-1}[\langle\boldsymbol{\phi}_h^\iota,\boldsymbol{q}_h^t\rangle]^2}\sqrt{\sum_{h=1}^{H}\sum_{t=1}^{T}\min\{(\varphi_h^t)^2,1\}\mathbf{1}_{\{\varphi_h^t<1\}}}$$

$$+\sqrt{4\lambda dH^3T\sum_{h=1}^{H}\sum_{t=1}^{T}\min\{(\varphi_h^t)^2,1\}\mathbf{1}_{\{\varphi_h^t<1\}}}+H\sum_{h=1}^{H}\sum_{t=1}^{T}\mathbf{1}_{\{\varphi_h^t\geq1\}}, \tag{8}$$

where the second inequality uses $\sqrt{a+b}\leq\sqrt{a}+\sqrt{b}$ and the last inequality is by Cauchy-Schwarz.

According to the elliptical potential lemma (Lemma 15), we have the following bound,

$$\sum_{h=1}^{H}\sum_{t=1}^{T}\min\{(\varphi_h^t)^2,1\}\leq\sum_{h=1}^{H}2\log\left(\frac{\det\Phi_h^T}{\det\Phi_h^0}\right)\leq2Hd\log\left(1+\frac{T}{d\lambda}\right), \tag{9}$$

where the first inequality uses Lemma 15, and the last inequality uses $\log\det\Phi_h^t\leq d\log\frac{\operatorname{tr}(\Phi_h^T)}{d}\leq d\log\frac{\lambda d+T}{d}$. We also note that $\phi_h^t$ cannot exceed 1 too many times, as detailed in Lemma 16. By Lemma 16 and the fact that $\mathbf{1}_{\{\varphi_h^t\geq1\}}\leq1$, we obtain

$$\sum_{h=1}^{H}\sum_{t=1}^{T}\mathbf{1}_{\{\varphi_h^t\geq1\}}\leq\min\left\{\frac{3Hd}{\log2}\log\left(1+\frac{1}{\lambda\log2}\right),HT\right\}. \tag{10}$$

Combining (6), (7), (8),(9), and (10), we obtain

$$\left|\sum_{t=1}^{T}\left(V_{f^t}^{\pi^t,\nu^t}-V_{f^*}^{\pi^t,\nu^t}\right)\right|$$

$$\leq\sqrt{2Hd\log\left(1+\frac{T}{d\lambda}\right)\sum_{h=1}^{H}\sum_{t=1}^{T}\sum_{\iota=1}^{t-1}[\langle\boldsymbol{\phi}_h^\iota,\omega_h^t\rangle]^2}$$

$$+\sqrt{4\lambda dH^3T\min\left\{2Hd\log\left(1+\frac{T}{d\lambda}\right),HT\right\}}+\min\left\{\frac{3H^2d}{\log2}\log\left(1+\frac{1}{\lambda\log2}\right),H^2T\right\}$$

$$\leq H\sqrt{16Hd\log\left(1+\frac{T}{d\lambda}\right)\sum_{t=1}^{T}\sum_{h=1}^{H}\sum_{\iota=1}^{t-1}\mathbb{E}_{(\pi^\iota,\nu^\iota,h)}D_{\text{He}}^2\left(\mathbb{P}_{f^t,h}(\cdot\,|\,\xi_h^\iota),\mathbb{P}_{f^*,h}(\cdot\,|\,\xi_h^\iota)\right)}$$

$$+\lambda dH^2T+\min\left\{2H^2d\log\left(1+\frac{T}{d\lambda}\right),H^2T\right\}+\min\left\{\frac{3H^2d}{\log2}\log\left(1+\frac{1}{\lambda\log2}\right),H^2T\right\}$$

$$\leq\sqrt{16H^3d\log\left(1+\frac{T}{d\lambda}\right)\sum_{t=1}^{T}\sum_{h=1}^{H}\sum_{\iota=1}^{t-1}\mathbb{E}_{(\pi^\iota,\nu^\iota,h)}D_{\text{He}}^2\left(\mathbb{P}_{f^t,h}(\cdot\,|\,\xi_h^\iota),\mathbb{P}_{f^*,h}(\cdot\,|\,\xi_h^\iota)\right)}$$

$$+\lambda dH^2T+2\left[2H^2d\log\left(1+\frac{T}{d\lambda}\right)\wedge H^2T\right], \tag{11}$$

which further leads to

$$\left| \sum_{t=1}^{T} \left( V_{f^t}^{\pi^t, \nu^t} - V_{f^*}^{\pi^t, \nu^t} \right) \right| \leq \sqrt{16 H^3 d \log\left(1 + \frac{HT}{\epsilon}\right) \sum_{t=1}^{T} \sum_{h=1}^{H} \sum_{\iota=1}^{t-1} \mathbb{E}_{(\pi^\iota, \nu^\iota, h)} \ell(f^t, \xi_h^\iota)}$$

$$+ \epsilon H T + 2 \sqrt{2 H^4 T d \log\left(1 + \frac{HT}{\epsilon}\right)}$$

by $x \wedge y \leq \sqrt{xy}$ and setting $\epsilon = \lambda H d$ where $\lambda > 0$ and $\ell(f, \xi_h) := D_{\mathrm{He}}^2\big(\mathbb{P}_{f,h}(\cdot \mid \xi_h), \mathbb{P}_{f^*,h}(\cdot \mid \xi_h)\big)$ with $\xi_h = (s_h, a_h, b_h)$. Thus, we obtain that when $T$ is sufficiently large, we have

$$d_{\mathrm{GEC}} = 16 H^3 d \log\left(1 + \frac{HT}{\epsilon}\right).$$

Note that the above results hold for arbitrary policy pair $(\nu^t, \pi^t)$. This implies that linear MGs with the dimension $d$ is subsumed by the class of MGs with self-play and adversarial GEC $d_{\mathrm{GEC}} = 16 H^3 d \log(1 + \frac{HT}{\epsilon})$. This completes the proof. $\qquad \square$

## B.2 Linear Mixture MG

In this subsection, we discuss the linear mixture MG setting, where the transition kernel is a linear combination of some basis transition kernels and thus admits a linear structure. Specifically, we have the following definition of the linear mixture MG.

**Definition 5** (Linear Mixture Markov Game [16]). *There exists a $d$-dimensional feature map $\phi : \mathcal{S} \times \mathcal{A} \times \mathcal{B} \times \mathcal{S} \mapsto \mathbb{R}^d$ and an unknown coefficient $\boldsymbol{\theta}_h \in \mathbb{R}^d$, such that the transition of the Markov game can be represented by*

$$\mathbb{P}_h(s_{h+1} \mid s_h, a_h, b_h) = \boldsymbol{\theta}_h^\top \phi(s_{h+1} \mid s_h, a_h, b_h),$$

*where $\|\boldsymbol{\theta}_h\|_2 \leq B$ and $\| \int_{\mathcal{S}} \phi(s, a, b, s') V(s') \mathrm{d}s' \|_2 \leq H$ for any bounded function $V : \mathcal{S} \mapsto [0, H]$.*

In the context of linear mixture MGs, both self-play and adversarial GEC can be calculated.

**Proposition 4** (Linear Mixture MG $\subset$ Low Self-play/Adversarial GEC). *For linear mixture MGs with $\phi(s, a, b, s') \in \mathbb{R}^d$ defined as in Definition 4, and for $\epsilon > 0$, when $T$ is sufficiently large, choosing $\ell(f, \xi_h) = D_{\mathrm{He}}^2\big(\mathbb{P}_{f,h}(\cdot \mid \xi_h), \mathbb{P}_{f^*,h}(\cdot \mid \xi_h)\big)$ as in (3) with $\xi_h = (s_h, a_h, b_h)$, we have*

*linear mixture MG $\subset$ MG with low self-play and adversarial GEC*

*with $d_{\mathrm{GEC}}$ satisfying*

$$d_{\mathrm{GEC}} = 16 H^3 d \log\left(1 + \frac{TB^2}{d\epsilon H}\right).$$

*Proof.* The proof of GEC bound in a linear mixture Markov game is similar to the one in the linear Markov game case. We first define the Bellman residual in expectation as

$$\mathcal{E}_{f,h}^{\pi,\nu} := \mathbb{E}_{\pi,\nu,\mathbb{P}_{f^*}} \left[ Q_{f,h}^{\pi,\nu}(s_h, a_h, b_h) - \mathbb{E}_{s_{h+1} \sim \mathbb{P}_{f^*,h}(\cdot \mid s_h, a_h, b_h)} \left( r_h(s_h, a_h, b_h) + V_{f,h+1}^{\pi,\nu}(s_{h+1}) \right) \right].$$

One can derive the following property of $\mathcal{E}_{f,h}^{\pi,\nu}$ by precisely the same derivation as in (4), which is

$$V_{f^t}^{\pi^t, \nu^t} - V_{f^*}^{\pi^t, \nu^t} = \sum_{h=1}^{H} \mathcal{E}_{f^t, h}^{\pi^t, \nu^t}.$$

We next show that in a linear mixture Markov game, the Bellman residual $\mathcal{E}_{f,h}^{\pi,\nu}$ can be linearly represented, whose form is

$$\mathcal{E}_{f,h}^{\pi,\nu} = \mathbb{E}_{\pi,\nu,\mathbb{P}_{f^*}} \int_{\mathcal{S}} [\mathbb{P}_{f,h}(s \mid s_h, a_h, b_h) - \mathbb{P}_{f^*,h}(s \mid s_h, a_h, b_h)] V_{f,h+1}^{\pi,\nu}(s) \mathrm{d}s$$

$$= \mathbb{E}_{\pi,\nu,\mathbb{P}_{f^*}} \int_{\mathcal{S}} [\phi(s_h, a_h, b_h, s)^\top (\boldsymbol{\theta}_{f,h} - \boldsymbol{\theta}_{f^*,h})] V_{f,h+1}^{\pi,\nu}(s) \mathrm{d}s$$

$$= \langle \mathbb{E}_{\pi,\nu,\mathbb{P}_{f^*}} \int_{\mathcal{S}} \phi(s_h, a_h, b_h, s) V_{f,h+1}^{\pi,\nu}(s) \mathrm{d}s, \boldsymbol{\theta}_{f,h} - \boldsymbol{\theta}_{f^*,h} \rangle$$

$$= \langle \phi(s_h, a_h, b_h, f, \pi, \nu), \boldsymbol{q}_h(f) \rangle,$$

where we define $\phi(s_h, a_h, b_h, f, \pi, \nu) := \mathbb{E}_{\pi, \nu, \mathbb{P}_{f^*}} \int_{\mathcal{S}} \phi(s_h, a_h, b_h, s') V_{f,h+1}^{\pi, \nu}(s') \mathrm{d}s'$ and $\boldsymbol{q}_h(f) := \boldsymbol{\theta}_{f,h} - \boldsymbol{\theta}_{f^*,h}$. Note that $\|\phi(s_h, a_h, b_h, f, \pi, \nu)\| \leq H$ and $\|\boldsymbol{q}_h(f)\| \leq 2B$ by Definition 5.

We next define $\phi_h^t := \phi(s_h^t, a_h^t, b_h^t, f^t, \pi^t, \nu^t)$, $\boldsymbol{q}_h^t := \boldsymbol{q}_h(f^t)$, $\Phi_h^t := \lambda \mathrm{I}_d + \frac{1}{H^2} \sum_{\iota=1}^t \phi_h^\iota (\phi_h^\iota)^\top$ and $\varphi_h^t := \|\phi_h^t\|_{(\Phi_h^{t-1})^{-1}}$. Since we have

$$|\langle \phi_h^\iota, \boldsymbol{q}_h^t \rangle|^2 = \left| \mathbb{E}_{\pi^\iota, \nu^\iota, \mathbb{P}_{f^*}} \int_{\mathcal{S}} \left( \mathbb{P}_{f^t, h}(s \mid s_h^\iota, a_h^\iota, b_h^\iota) - \mathbb{P}_{f^*, h}(s \mid s_h^\iota, a_h^\iota, b_h^\iota) \right) V_{f^\iota, h+1}^{\pi^\iota, \nu^\iota}(s) \mathrm{d}s \right|^2$$

$$\leq H^2 \mathbb{E}_{\pi^\iota, \nu^\iota, \mathbb{P}_{f^*}} \|\mathbb{P}_{f^t, h}(\cdot \mid s_h^\iota, a_h^\iota, b_h^\iota) - \mathbb{P}_{f^*, h}(\cdot \mid s_h^\iota, a_h^\iota, b_h^\iota)\|_1^2$$

$$\leq 8H^2 \mathbb{E}_{\pi^\iota, \nu^\iota, \mathbb{P}_{f^*}} D_{\mathrm{He}}^2 \left( \mathbb{P}_{f^t, h}(\cdot \mid s_h^\iota, a_h^\iota, b_h^\iota), \mathbb{P}_{f^*, h}(\cdot \mid s_h^\iota, a_h^\iota, b_h^\iota) \right), \tag{12}$$

where the first inequality is due to $V_{f^\iota, h+1}^{\pi^\iota, \nu^\iota}(s) \leq H$ and the second inequality is due to $\|P - Q\|_2^2 \leq 8D_{\mathrm{He}}^2(P, Q)$. By utilizing $\|\boldsymbol{q}_h(f)\| \leq 2B$ and (12), we further obtain an upper bound of $\|\boldsymbol{q}_h^t\|_{\Phi_h^{t-1}}$ as follows,

$$\|\boldsymbol{q}_h^t\|_{\Phi_h^{t-1}} \leq \sqrt{\sum_{\iota=1}^{t-1} [\langle \phi_h^\iota, \boldsymbol{q}_h^t \rangle]^2 + \lambda \|\boldsymbol{q}_h^t\|_2^2}$$

$$\leq \sqrt{\sum_{\iota=1}^{t-1} 8H^2 \mathbb{E}_{\pi^\iota, \nu^\iota, \mathbb{P}_{f^*}} D_{\mathrm{He}}^2 \left( \mathbb{P}_{f^t, h}(\cdot \mid s_h^\iota, a_h^\iota, b_h^\iota), \mathbb{P}_{f^*, h}(\cdot \mid s_h^\iota, a_h^\iota, b_h^\iota) \right) + 4\lambda B^2}.$$

The rest of the proof is the same as the proof in the linear Markov game, from (5) to (11), except with a scaling factor. We provide the final result here as follows,

$$\left| \sum_{t=1}^T \left( V_{f^t}^{\pi^t, \nu^t} - V_{f^*}^{\pi^t, \nu^t} \right) \right|$$

$$\leq \sqrt{16H^3 d \log\left(1 + \frac{T}{d\lambda}\right) \sum_{t=1}^T \sum_{h=1}^H \sum_{\iota=1}^{t-1} \mathbb{E}_{\pi^\iota, \nu^\iota, \mathbb{P}_{f^*}} D_{\mathrm{He}}^2 \left( \mathbb{P}_{f^t, h}(\cdot \mid s_h^\iota, a_h^\iota, b_h^\iota), \mathbb{P}_{f^*, h}(\cdot \mid s_h^\iota, a_h^\iota, b_h^\iota) \right)}$$

$$+ \lambda B^2 T + \min\left\{ 2H^2 d \log\left(1 + \frac{T}{d\lambda}\right), H^2 T \right\} + \min\left\{ \frac{3H^2 d}{\log 2} \log\left(1 + \frac{1}{\lambda \log 2}\right), H^2 T \right\}$$

$$\leq \sqrt{16H^3 d \log\left(1 + \frac{T}{d\lambda}\right) \sum_{t=1}^T \sum_{h=1}^H \sum_{\iota=1}^{t-1} \mathbb{E}_{\pi^\iota, \nu^\iota, \mathbb{P}_{f^*}} D_{\mathrm{He}}^2 \left( \mathbb{P}_{f^t, h}(\cdot \mid s_h^\iota, a_h^\iota, b_h^\iota), \mathbb{P}_{f^*, h}(\cdot \mid s_h^\iota, a_h^\iota, b_h^\iota) \right)}$$

$$+ \lambda B^2 T + 2\left[ 2H^2 d \log\left(1 + \frac{T}{d\lambda}\right) \wedge H^2 T \right],$$

Note that here $(\pi^t, \nu^t)$ can be any arbitrary policy pair, including the ones required as in Definitions 1 and 3. Moreover, we will see that the above result satisfies the definitions of both self-play and adversarial GEC when using the inequality $x \wedge y \leq \sqrt{xy}$ and choosing $\lambda = \epsilon H / B^2$ and $\ell(f, \xi_h) = D_{\mathrm{He}}^2 \left( \mathbb{P}_{f, h}(\cdot \mid \xi_h), \mathbb{P}_{f^*, h}(\cdot \mid \xi_h) \right)$. Therefore, we have

$$d_{\mathrm{GEC}} = 16H^3 d \log\left(1 + \frac{TB^2}{d\epsilon H}\right)$$

for $T$ sufficiently large, which shows that linear mixture MGs with the dimension $d$ is subsumed by MGs with self-play and adversarial GEC $d_{\mathrm{GEC}} = 16H^3 d \log(1 + \frac{TB^2}{d\epsilon H})$. This completes the proof. $\square$

## B.3 Low-Witness-Rank MG

Witness rank is a complexity measure that absorbs a wide range of settings in single-agent RL[60]. The work [28] further studies the witness rank under the MG setting with self-play. In this subsection, we show that MGs with low self-play GEC subsumes the class of MGs with a low witness rank in the

self-play setting. Since the witness rank under the adversarial setting was not investigated in prior works, we here only analyze the self-play setting.

Before presenting our main proposition, we first show the definitions of witnessed model misfit and witness rank in MGs with self-play [28].

**Definition 6** (Witnessed model misfit in Markov game). *Suppose that there exists a discriminator class* $\mathcal{U}$ *with* $u \in \mathcal{U} : \mathcal{S} \times \mathcal{A} \times \mathcal{B} \times \mathcal{S} \mapsto [0, H]$. *The witnessed model misfit in Markov games* $\mathcal{W}(f, g, \rho, h)$ *for the models* $f, g, \rho \in \mathcal{F}$ *at step* $h$ *is characterized by*

$$
\mathcal{W}(f, g, \rho, h) = \sup_{u \in \mathcal{U}} \left| \mathbb{E}_{(s_h, a_h, b_h) \sim \mathbb{P}_{f^*}, \breve{\pi}, \breve{\nu}} \left[ \mathbb{E}_{s' \sim \mathbb{P}_{\rho, h}(\cdot \,|\, s_h, a_h, b_h)} u(s_h, a_h, b_h, s') \right. \right.
$$
$$
\left. \left. - \mathbb{E}_{s_{h+1} \sim \mathbb{P}_{f^*, h}(\cdot \,|\, s_h, a_h, b_h)} u(s_h, a_h, b_h, s_{h+1}) \right] \right|,
$$

*where* $\breve{\pi} := \operatorname{argmax}_\pi \min_\nu V_f^{\pi, \nu}$ *and* $\breve{\nu} := \operatorname{argmin}_\nu V_g^{\breve{\pi}, \nu}$ *with* $f, g \in \mathcal{F}$.

The definition of $(\breve{\pi}, \breve{\nu})$ matches the requirement for the policy pair in the self-play GEC, i.e., condition **(1)** in Definition 1. We note that condition **(2)** in our definition of self-play GEC is designed for Algorithm 3, which is thus a symmetric condition to **(1)**. We define the Bellman residual as

$$
\mathcal{E}_{f, h}^{\pi, \nu} := \mathbb{E}_{\pi, \nu, \mathbb{P}_{f^*}} \left[ Q_{f, h}^{\pi, \nu}(s_h, a_h, b_h) - \mathbb{E}_{s_{h+1} \sim \mathbb{P}_{f^*, h}(\cdot | s_h, a_h, b_h)} \left( r_h(s_h, a_h, b_h) + V_{f, h+1}^{\pi, \nu}(s_{h+1}) \right) \right].
$$

Then, we have the following definition of witness rank for MGs with self-play.

**Definition 7** (Witness Rank for Self-Play). *There exist two functions* $\boldsymbol{W}_h : \mathcal{F} \times \mathcal{F} \mapsto \mathbb{R}^d$ *and* $\boldsymbol{X}_h : \mathcal{F} \mapsto \mathbb{R}^d$ *and constant* $\kappa > 0$ *such that*

$$
\kappa_{\mathrm{wit}} |\mathcal{E}_{\rho, h}^{\breve{\pi}, \breve{\nu}}| \le \langle \boldsymbol{W}_h(f, g), \boldsymbol{X}_h(\rho) \rangle, \quad \mathcal{W}(f, g, \rho, h) \ge \langle \boldsymbol{W}_h(f, g), \boldsymbol{X}_h(\rho) \rangle,
$$

*where* $\breve{\pi} := \operatorname{argmax}_\pi \min_\nu V_f^{\pi, \nu}$ *and* $\breve{\nu} := \operatorname{argmin}_\nu V_g^{\breve{\pi}, \nu}$ *with* $f, g \in \mathcal{F}$, $\|\boldsymbol{W}_h(\cdot)\|_2 \le HB$ *for a constant* $B$, *and* $\|\boldsymbol{X}_h(\cdot)\|_2 \le 1$ *for any step* $h$. *Then, the witness rank of MGs with self-play is defined as the dimension* $d$ *of the range spaces for functions* $\boldsymbol{W}$ *and* $\boldsymbol{X}$.

We show that in the self-play setting, $d_{\mathrm{GEC}}$ is bounded for MGs with a low witness rank. It is worth noting that the proof of GEC bound for MGs with a low witness rank generalizes the proof for linear MGs in Appendix B.1.

**Proposition 5** (Low Self-Play Witness Rank $\subset$ Low Self-play GEC). *For a Markov game with witness rank* $d$ *as defined in Definition 7, when* $T$ *sufficiently large, choosing* $\ell(f, \xi_h) = D_{\mathrm{He}}^2 \big( \mathbb{P}_{f, h}(\cdot \,|\, \xi_h), \mathbb{P}_{f^*, h}(\cdot \,|\, \xi_h) \big)$ *as in* (3) *with* $\xi_h = (s_h, a_h, b_h)$, *we have*

*MG with low self-play witness rank* $\subset$ *MG with low self-play GEC*

*with* $d_{\mathrm{GEC}}$ *satisfying*

$$
d_{\mathrm{GEC}} = \frac{4H^3 d}{\kappa_{\mathrm{wit}}^2} \log \left( 1 + \frac{THB^2}{4d\epsilon \kappa_{\mathrm{wit}}^2} \right).
$$

*Proof.* We first give a bound of the model misfit in the Hellinger distance. By $0 \le u \le H$, we have

$$
\left[ \mathbb{E}_{s' \sim \mathbb{P}_{\rho, h}(\cdot \,|\, s_h, a_h, b_h)} u(s_h, a_h, b_h, s') - \mathbb{E}_{s_{h+1} \sim \mathbb{P}_{f^*, h}(\cdot \,|\, s_h, a_h, b_h)} u(s_h, a_h, b_h, s_{h+1}) \right]^2
$$
$$
\le H^2 \big\| \mathbb{P}_{\rho, h}(\cdot \,|\, s_h, a_h, b_h) - \mathbb{P}_{f^*, h}(\cdot \,|\, s_h, a_h, b_h) \big\|_1^2
$$
$$
\le 2H^2 D_{\mathrm{He}}^2 \big( \mathbb{P}_{\rho, h}(\cdot \,|\, s_h, a_h, b_h), \mathbb{P}_{f^*, h}(\cdot \,|\, s_h, a_h, b_h) \big),
$$

where $\rho$ can be $f$ or $g$ as in the definition of self-play GEC. By the definition of $\mathcal{W}(f, g, \rho, h)$ and Jensen's inequality, we further get

$$
\mathcal{W}(f, g, \rho, h)^2 \le 2H^2 \mathbb{E}_{(s_h, a_h, b_h) \sim \mathbb{P}_{f^*}, \breve{\pi}, \breve{\nu}} D_{\mathrm{He}}^2 \big( \mathbb{P}_{\rho, h}(\cdot \,|\, s_h, a_h, b_h), \mathbb{P}_{f^*, h}(\cdot \,|\, s_h, a_h, b_h) \big), \quad (13)
$$

where we use $\sup_v \mathbb{E}_x f(x, v) \le \mathbb{E}_x \sup_v f(x, v)$ and the definition of $(\breve{\pi}, \breve{\nu})$. Note that following our definition of self-play GEC, if we choose $f = f^t$ and $g = g^t$ in the model misfit, then correspondingly we have $\breve{\pi} = \pi^t$ and $\breve{\nu} = \nu^t$ as in the condition **(1)** of Definition 1.

By the definition of $\mathcal{E}_{\rho,h}^{\pi,\nu}$, we can decompose the value difference as

$$\left| \sum_{t=1}^{T} \left( V_{\rho^t}^{\pi^t,\nu^t} - V_{f^*}^{\pi^t,\nu^t} \right) \right| = \left| \sum_{t=1}^{T} \sum_{h=1}^{H} \mathcal{E}_{\rho^t,h}^{\pi^t,\nu^t} \right|.$$

Hereafter we define $\boldsymbol{W}_h^t := \boldsymbol{W}_h(f^t,g^t)$, $\boldsymbol{X}_h^t := \boldsymbol{X}_h(\rho^t)$, $\Sigma_h^t := \lambda I + \sum_{\iota=1}^{t} \boldsymbol{X}_h^\iota (\boldsymbol{X}_h^\iota)^\top$, and $\varphi_h^t := \|\boldsymbol{X}_h^t\|_{(\Sigma_h^t)^{-1}}^2$. Using the definition of $\boldsymbol{W}_h$ and $\boldsymbol{X}_h$ and $\mathcal{E}_{\rho^t,h}^{\pi^t,\nu^t} \leq H$, we obtain

$$\left| \sum_{t=1}^{T} \sum_{h=1}^{H} \mathcal{E}_{\rho^t,h}^{\pi^t,\nu^t} \right| \leq \sum_{t=1}^{T} \sum_{h=1}^{H} H \min \left\{ \frac{1}{H\kappa_{\mathrm{wit}}} \langle \boldsymbol{W}_h(f^t,g^t), \boldsymbol{X}_h(\rho^t) \rangle, 1 \right\}$$

$$= \sum_{t=1}^{T} \sum_{h=1}^{H} H \min \left\{ \frac{1}{H\kappa_{\mathrm{wit}}} \langle \boldsymbol{W}_h(f^t,g^t), \boldsymbol{X}_h(\rho^t) \rangle, 1 \right\} \left( \mathbf{1}_{\{\varphi_h^{t-1} \leq 1\}} + \mathbf{1}_{\{\varphi_h^{t-1} > 1\}} \right)$$

$$\leq \underbrace{\sum_{t=1}^{T} \sum_{h=1}^{H} \frac{1}{\kappa_{\mathrm{wit}}} \|\boldsymbol{W}_h^t\|_{\Sigma_h^{t-1}} \min \left\{ \|\boldsymbol{X}_h^t\|_{(\Sigma_h^{t-1})^{-1}}, 1 \right\} \mathbf{1}_{\{\varphi_h^{t-1} \leq 1\}}}_{\mathrm{Term(i)}}$$

$$+ \underbrace{H \sum_{t=1}^{T} \sum_{h=1}^{H} \mathbf{1}_{\{\varphi_h^{t-1} > 1\}}}_{\mathrm{Term(ii)}},$$

where $\mathbf{1}_{\{\cdot\}}$ denotes the indicator function. For Term(ii), we can apply Lemma 16 to obtain

$$H \sum_{t=1}^{T} \sum_{h=1}^{H} \mathbf{1}_{\{\varphi_h^{t-1} > 1\}} \leq \min \left\{ \frac{3H^2 d}{\log 2} \left( 1 + \frac{1}{\lambda \log 2} \right), H^2 T \right\}.$$

It remains to bound Term(i). We first calculate the sum of $\|\boldsymbol{W}_h^t\|_{\Sigma_h^t}^2$, which is

$$\sum_{t=1}^{T} \sum_{h=1}^{H} \|\boldsymbol{W}_h^{t-1}\|_{\Sigma_h^t}^2$$

$$= \sum_{t=1}^{T} \sum_{h=1}^{H} \left( \lambda \|\boldsymbol{W}_h^t\|_2^2 + \sum_{\iota=1}^{t-1} \langle \boldsymbol{W}_h^t, \boldsymbol{X}_h^\iota \rangle^2 \right)$$

$$\leq H^3 T B^2 \lambda + \sum_{t=1}^{T} \sum_{h=1}^{H} \sum_{\iota=1}^{t-1} \langle \boldsymbol{W}_h^t, \boldsymbol{X}_h^\iota \rangle^2$$

$$\leq H^3 T B^2 \lambda + 2H^2 \sum_{t=1}^{T} \sum_{h=1}^{H} \sum_{\iota=1}^{t-1} \mathbb{E}_{(\pi^\iota,\nu^\iota,h)} D_{\mathrm{He}}^2 \left( \mathbb{P}_{\rho^\iota,h}(\cdot \mid s_h, a_h, b_h), \mathbb{P}_{f^*,h}(\cdot \mid s_h, a_h, b_h) \right),$$

where the first inequality uses $\|\boldsymbol{W}_h^t\| \leq HB$, and the second inequality uses the definition of witness rank and (13). We also let $\xi_h^\iota := (s_h^\iota, a_h^\iota, b_h^\iota)$. We next use Lemma 15 to obtain

$$\sum_{h=1}^{H} \sum_{t=1}^{T} \min\{\varphi_h^t, 1\} \leq \min \left\{ 2Hd \log \left( 1 + \frac{T}{d\lambda} \right), HT \right\}.$$

Finally, by using the Cauchy-Schwarz inequality, we obtain the bound for Term(i) as

$$
\sum_{t=1}^{T}\sum_{h=1}^{H}\frac{1}{\kappa_{\mathrm{wit}}}\|\boldsymbol{W}_h^t\|_{\Sigma_h^{t-1}}\min\{\|\boldsymbol{X}_h^t\|_{(\Sigma_h^{t-1})^{-1}},1\}\mathbf{1}_{\{\varphi_h^{t-1}\le 1\}}
$$

$$
\le \sqrt{\sum_{t=1}^{T}\sum_{h=1}^{H}\left(\frac{1}{\kappa_{\mathrm{wit}}}\|\boldsymbol{W}_h^t\|_{\Sigma_h^{t-1}}\right)^2}\sqrt{\sum_{t=1}^{T}\sum_{h=1}^{H}\min\{\varphi_h^t,1\}}
$$

$$
\le \frac{1}{\kappa_{\mathrm{wit}}}\sqrt{\lambda H^3 T B^2 + 2H^2\sum_{t=1}^{T}\sum_{h=1}^{H}\sum_{\iota=1}^{t-1}\mathbb{E}_{(\pi^\iota,\nu^\iota,h)}D_{\mathrm{He}}^2\big(\mathbb{P}_{\rho^\iota,h}(\cdot\,|\,\xi_h^\iota),\mathbb{P}_{f^*,h}(\cdot\,|\,\xi_h^\iota)\big)}
$$

$$
\cdot\sqrt{\min\left\{2Hd\log\left(1+\frac{T}{d\lambda}\right),HT\right\}}
$$

$$
\le \left[\sqrt{\frac{\lambda H^3 T B^2}{\kappa_{\mathrm{wit}}^2}}+\sqrt{\frac{2H^2}{\kappa_{\mathrm{wit}}^2}\sum_{t=1}^{T}\sum_{h=1}^{H}\sum_{\iota=1}^{t-1}\mathbb{E}_{(\pi^\iota,\nu^\iota,h)}D_{\mathrm{He}}^2\big(\mathbb{P}_{\rho^\iota,h}(\cdot\,|\,\xi_h^\iota),\mathbb{P}_{f^*,h}(\cdot\,|\,\xi_h^\iota)\big)}\right]
$$

$$
\cdot\sqrt{\min\left\{2Hd\log\left(1+\frac{T}{d\lambda}\right),HT\right\}},
$$

where the first inequality is by Cauchy-Schwarz, and the last inequality is by $\sqrt{a+b}\le\sqrt{a}+\sqrt{b}$. We further invoke $\sqrt{ab}\le\frac{a}{4}+b$ to obtain that

$$
\mathrm{Term(i)}
$$

$$
\le \frac{\lambda H^2 T B^2}{4\kappa_{\mathrm{wit}}^2}+\min\left\{2H^2 d\log\left(1+\frac{T}{d\lambda}\right),H^2 T\right\}
$$

$$
+\sqrt{\frac{4H^3 d}{\kappa_{\mathrm{wit}}^2}\log\left(1+\frac{T}{d\lambda}\right)\sum_{t=1}^{T}\sum_{h=1}^{H}\sum_{\iota=1}^{t-1}\mathbb{E}_{(\pi^\iota,\nu^\iota,h)}D_{\mathrm{He}}^2\big(\mathbb{P}_{\rho^\iota,h}(\cdot\,|\,\xi_h^\iota),\mathbb{P}_{f^*,h}(\cdot\,|\,\xi_h^\iota)\big)}.
$$

Finally, we combine the bounds for term(i) and term(ii) to obtain that for $T$ sufficiently large,

$$
\left|\sum_{t=1}^{T}\left(V_{\rho^t}^{\pi^t,\nu^t}-V_{f^*}^{\pi^t,\nu^t}\right)\right|
$$

$$
\le \frac{\lambda H^2 T B^2}{4\kappa_{\mathrm{wit}}^2}+2\sqrt{2H^4 dT\log\left(1+\frac{T}{d\lambda}\right)}
$$

$$
+\sqrt{\frac{4H^3 d}{\kappa_{\mathrm{wit}}^2}\log\left(1+\frac{T}{d\lambda}\right)\sum_{t=1}^{T}\sum_{h=1}^{H}\sum_{\iota=1}^{t-1}\mathbb{E}_{(\pi^\iota,\nu^\iota,h)}D_{\mathrm{He}}^2\big(\mathbb{P}_{\rho^\iota,h}(\cdot\,|\,\xi_h^\iota),\mathbb{P}_{f^*,h}(\cdot\,|\,\xi_h^\iota)\big)},
$$

where we also use the inequality $x\wedge y\le\sqrt{xy}$. Then, we can see that the above results satisfy Definition 1 with choosing $\lambda=\frac{4\epsilon\kappa_{\mathrm{wit}}^2}{HB^2}$. Thus, we conclude that under the self-play setting, we have

$$
d_{\mathrm{GEC}}=\frac{4H^3 d}{\kappa_{\mathrm{wit}}^2}\log\left(1+\frac{THB^2}{4d\epsilon\kappa_{\mathrm{wit}}^2}\right).
$$

This concludes the proof. □

## B.4 Weakly Revealing POMG

In this section, we consider a weakly revealing POMG. Recall that $\{\mathbb{O}_h\}_{h=1}^{H}$ stands for the emission kernels, which can be viewed in the form of matrices $\mathbb{O}_h\in\mathbb{R}^{|\mathcal{O}|\times|\mathcal{S}|}$, with the $(o,s)$-th element being $\mathbb{O}_h(o\,|\,s)$ for step $h$. Similarly, the transition matrix $\mathbb{P}_{h,a,b}\in\mathbb{R}^{|\mathcal{S}|\times|\mathcal{S}|}$ is such that the $(s',s)$-th element is $\mathbb{P}_h(s'\,|\,s,a,b)$.

Here we consider undercomplete POMGs where there are more observations than hidden states, $|\mathcal{O}| \geq |\mathcal{S}|$. Formally, it can be defined as an $\alpha$-revealing POMG as follows:

**Definition 8** ($\alpha$-Revealing POMG [41]). *A POMG is $\alpha$-revealing if it satisfies that there exists $\alpha > 0$ such that $\min_h \sigma_S(\mathbb{O}_h) \geq \alpha$, where $\sigma_S(\cdot)$ denotes the $S$-th singular value of a matrix with $S = |\mathcal{S}|$.*

The $\alpha$-revealing condition measures how hard it is to recover states from observations. Before presenting the GEC bound for an $\alpha$-revealing POMG, we first define some notations. We define $\tau_{h:h'} := (o_i, a_i, b_i)_{i=h}^{h'}$ to be a set of observation-action tuples from step $h$ to $h'$. For simplicity, we denote $\tau_{1:h}$ as $\tau_h$. We define $\boldsymbol{q}_0 := \mathbb{O}_1 \boldsymbol{\mu}_1 \in \mathbb{R}^{|\mathcal{O}|}$, where $\boldsymbol{\mu}_1 \in \mathbb{R}^{|\mathcal{S}|}$ is a vector formed by the distribution of the initial state. We define $\boldsymbol{q}(\tau_h) := [\Pr\left(o_{1:h}, o_{h+1} \mid \mathbf{do}(a_{1:h}, b_{1:h})\right)]_{o \in \mathcal{O}} \in \mathbb{R}^{|\mathcal{O}|}$ when $\tau_h = (o_i, a_i, b_i)_{i=1}^{h}$. Here $\mathbf{do}(a_{1:h}, b_{1:h})$ means executing actions $a_{h'}$ and $b_{h'}$ at step $h'$. In other words, $\Pr\left(o_{1:h}, o_{h+1} \mid \mathbf{do}(a_{1:h}, b_{1:h})\right)$ describes the probability of receiving observations $o_{1:h}$ given the actions fixed as $(a_{1:h-1}, b_{1:h-1})$. We define $\bar{\boldsymbol{q}}(\tau_h) := \boldsymbol{q}(\tau_h)/\mathbf{P}(\tau_h)$, which is the concatenation of the probabilities under condition $\tau_h$. We have $\bar{\boldsymbol{q}}(\tau_h) = [\Pr(o \mid \tau_{1:h})]_{o \in \mathcal{O}}$, where $\Pr(o \mid \tau_{1:h})$ means the probability of receiving $o_{h+1} = o$ under the condition that the previous trajectory is $\tau_h$.

Based on the above definitions, we then derive the GEC bound for an $\alpha$-revealing POMG in the following proposition.

**Proposition 6** ($\alpha$-Revealing POMG $\subset$ Low Self-Play/Adversarial GEC). *For an $\alpha$-revealing POMG defined in Definition 8, with the loss $\ell(f, \xi_h)$ defined in (3) for POMGs, we have*

$$\alpha\text{-revealing POMG} \subset \text{MG with low self-play and adversarial GEC}$$

*with $d_{\mathrm{GEC}}$ satisfying*

$$d_{\mathrm{GEC}} = O\left(\frac{H^3 |\mathcal{O}|^3 |\mathcal{A}|^2 |\mathcal{B}|^2 |\mathcal{S}|^2 \varsigma}{\alpha^4}\right),$$

*where $\varsigma = 2 \log\left(1 + \frac{4|\mathcal{A}|^2 |\mathcal{B}|^2 |\mathcal{O}|^2 |\mathcal{S}|^2}{\alpha^2}\right)$.*

For our proof of this proposition, we prove several supporting lemmas presented in Appendix B.4.1.

*Proof.* For ease of notation, we let $S = |\mathcal{S}|$, $A = |\mathcal{A}|$, $B = |\mathcal{B}|$, and $O = |\mathcal{O}|$ in our proof. We define the following observed operator representation:

$$\boldsymbol{M}_h(o, a, b) := \mathbb{O}_{h+1} \mathbb{P}_{h,a,b} \mathrm{diag}\left(\mathbb{O}_h(o \mid \cdot)\right) \mathbb{O}_h^\dagger \quad \in \mathbb{R}^{O \times O}$$

for $1 \leq h \leq H$. One can verify that the previously defined $\boldsymbol{M}_h(o, a, b)$ has the following property,

$$\boldsymbol{q}(\tau_h) = \boldsymbol{M}_h(o_h, a_h, b_h) \boldsymbol{M}_{h-1}(o_{h-1}, a_{h-1}, b_{h-1}) \ldots \boldsymbol{M}_1(o_1, a_1, b_1) \boldsymbol{q}_0.$$

Specifically, $\mathbf{P}(\tau_H) = \boldsymbol{M}_H(o_H, a_H, b_H) \ldots \boldsymbol{M}_1(o_1, a_1, b_1) \boldsymbol{q}_0$. For simplicity, we define

$$\boldsymbol{M}_{h':h}(o_{h:h'}, a_{h:h'}, b_{h:h'}) := \boldsymbol{M}_{h'}(o_{h'}, a_{h'}, b_{h'}) \boldsymbol{M}_{h'-1}(o_{h'-1}, a_{h'-1}, b_{h'-1}) \ldots \boldsymbol{M}_h(o_h, a_h, b_h).$$

We also rewrite $\boldsymbol{M}_{H:h+1}(o_{h+1:H}, a_{h+1:H}, b_{h+1:H})$ as $\boldsymbol{m}(o_{h+1:H}, a_{h+1:H}, b_{h+1:H})$ to emphasize that $\boldsymbol{M}_{H:h+1}(o_{h+1:H}, a_{h+1:H}, b_{h+1:H})$ is a vector since $\boldsymbol{M}_H(o_H, a_H, b_H)$ is a vector.

We start by using Lemma 5 to decompose the regret as follows

$$\left|\sum_{t=1}^{T}\left(V_{f^t}^{\pi^t, \nu^t} - V_{f^*}^{\pi^t, \nu^t}\right)\right|$$
$$\leq \sum_{t=1}^{T} \frac{H\sqrt{S}}{\alpha}\left[\sum_{h=1}^{H} \sum_{\tau_h} \|(\boldsymbol{M}_h^t(o_h, a_h, b_h) - \boldsymbol{M}_h(o_h, a_h, b_h))\boldsymbol{q}(\tau_{h-1})\|_1 \sigma^t(\tau_h) + \|\boldsymbol{q}_0^t - \boldsymbol{q}_0\|_1\right].$$

Combine this with the fact that $V_{f^t}^{\pi^t,\nu^t} - V_{f^*}^{\pi^t,\nu^t} \leq H$, we have

$$\left| \sum_{t=1}^{T} \left( V_{f^t}^{\pi^t,\nu^t} - V_{f^*}^{\pi^t,\nu^t} \right) \right|$$

$$\leq \sum_{t=1}^{T} \frac{H\sqrt{S}}{\alpha} \Big[ \sum_{h=1}^{H} \min \Big\{ \frac{\alpha}{\sqrt{S}}, \sum_{\tau_h} \|(\boldsymbol{M}_h^t(o_h,a_h,b_h) - \boldsymbol{M}_h(o_h,a_h,b_h))\boldsymbol{q}(\tau_{h-1})\|_1 \sigma^t(\tau_h) \Big\}$$

$$+ \|\boldsymbol{q}_0^t - \boldsymbol{q}_0\|_1 \Big].$$

where we define

$$\pi(\tau_h) = \prod_{h'=1}^{h} \pi_{h'}(a_{h'} \mid \tau_{h'-1}, o_{h'}), \quad \nu(\tau_h) = \prod_{h'=1}^{h} \nu_{h'}(b_{h'} \mid \tau_{h'-1}, o_{h'}), \quad \sigma^t(\tau_h) = \pi^t(\tau_h)\nu^t(\tau_h),$$

and $M_h^t$ and $q_0^t$ correspond to the model $f^t$ sampled in time step $t$.

We next focus on $\sum_{\tau_h} \|(\boldsymbol{M}_h^t(o_h,a_h,b_h) - \boldsymbol{M}_h(o_h,a_h,b_h))\boldsymbol{q}(\tau_{h-1})\|_1 \sigma^t(\tau_h)$. By the definition of $\bar{\boldsymbol{q}}$ and $\boldsymbol{q}$, we obtain

$$\sum_{\tau_h} \|\big(\boldsymbol{M}_h^t(o_h,a_h,b_h) - \boldsymbol{M}_h(o_h,a_h,b_h)\big)\boldsymbol{q}(\tau_{h-1})\|_1 \sigma(\tau_h)$$

$$= \mathbb{E}_{\tau_{h-1} \sim \mathbf{P}_{f^*,h-1}^{\pi^t,\nu^t}} \sum_{o_h,a_h,b_h} \sum_{j=1}^{O} |\widetilde{\boldsymbol{e}}_j^\top (\boldsymbol{M}_h^t(o_h,a_h,b_h) - \boldsymbol{M}_h(o_h,a_h,b_h))\bar{\boldsymbol{q}}(\tau_{h-1})| \sigma^t(o_h,a_h,b_h;\tau_{h-1})$$

$$= \mathbb{E}_{\tau_{h-1} \sim \mathbf{P}_{f^*,h-1}^{\pi^t,\nu^t}} \sum_{o_h,a_h,b_h} \sum_{j=1}^{O} \Big| \widetilde{\boldsymbol{e}}_j^\top (\boldsymbol{M}_h^t(o_h,a_h,b_h) - \boldsymbol{M}_h(o_h,a_h,b_h))\mathbb{O}_{h-1}\mathbb{O}_{h-1}^\dagger$$

$$\cdot \bar{\boldsymbol{q}}(\tau_{h-1})\sigma^t(o_h,a_h,b_h;\tau_{h-1}) \Big|,$$

where $\widetilde{\boldsymbol{e}}_j$ denotes the $j$-th standard basis vector in $\mathbb{R}^O$, and $\sigma^t(\tau_{h:h'};\tau_{h-1}) := \pi^t(\tau_{h:h'};\tau_{h-1}) \cdot \nu^t(\tau_{h:h'};\tau_{h-1})$, where $\pi^t(\tau_{h:h'};\tau_{h-1}) = \prod_{h''=h}^{h'} \pi_{h''}^t(a_{h''} \mid \tau_{h''-1}, o_{h''})$, $\nu^t(\tau_{h:h'};\tau_{h-1}) = \prod_{h''=h}^{h'} \nu_{h''}^t(b_{h''} \mid \tau_{h''-1}, o_{h''})$. For simplicity, we define $w_{h,t,j,o,a,b} := (\widetilde{\boldsymbol{e}}_j^\top (\boldsymbol{M}_h^t(o,a,b) - \boldsymbol{M}_h(o,a,b))\mathbb{O}_{h-1})^\top$ and $x_{\tau_{h-1},\sigma,o,a,b} := \mathbb{O}_{h-1}^\dagger \bar{\boldsymbol{q}}(\tau_{h-1})\sigma(o,a,b;\tau_{h-1})$. Then, we have

$$\mathbb{E}_{\tau_{h-1} \sim \mathbf{P}_{f^*,h-1}^{\pi^t,\nu^t}} \sum_{o_h,a_h,b_h} \sum_{j=1}^{O} |\widetilde{\boldsymbol{e}}_j^\top (\boldsymbol{M}_h^t(o_h,a_h,b_h) - \boldsymbol{M}_h(o_h,a_h,b_h))\mathbb{O}_{h-1}\mathbb{O}_{h-1}^\dagger \bar{\boldsymbol{q}}(\tau_{h-1})\sigma^t(o_h,a_h,b_h;\tau_{h-1})|$$

$$= \sum_{o,a,b} \mathbb{E}_{\tau_{h-1} \sim \mathbf{P}_{f^*,h-1}^{\pi^t,\nu^t}} \sum_{j=1}^{O} |w_{h,t,j,o,a,b}^\top x_{\tau_{h-1},\sigma^t,o,a,b}|.$$

Next, we will apply the $\ell_2$ eluder technique [13, 83] in Lemma 17 to derive the upper bound. We first analyze the upper bounds of $\sum_{j=1}^{O} \|w_{h,t,j,o,a,b}\|_2$ and $\mathbb{E}_{\tau_{h-1} \sim \mathbf{P}_{f^*,h-1}^{\pi^t,\nu^t}} \|x_{\tau_{h-1},\sigma^t,o,a,b}\|_2^2$ for the usage of the $\ell_2$ eluder technique. According to Lemma 7, for any $(o,a,b) \in \mathcal{O} \times \mathcal{A} \times \mathcal{B}$, we have

$$\mathbb{E}_{\tau_{h-1} \sim \mathbf{P}_{f^*,h-1}^{\pi^t,\nu^t}} \|x_{\tau_{h-1},\sigma^t,o,a,b}\|_2^2 \leq S, \qquad \sum_{j=1}^{O} \|w_{h,t,j,o,a,b}\|_2 \leq \frac{2ABd\sqrt{S}}{\alpha}.$$

According to Lemma 17, setting $R = \frac{\alpha}{\sqrt{S}}$ and $R_x^2 R_w^2 \le \frac{4A^2 B^2 O^2 S^2}{\alpha^2}$ in the new $\ell_2$ eluder technique, we can obtain

$$\sum_{t=1}^{T} \min\left\{ \frac{\alpha}{\sqrt{S}}, \; \mathbb{E}_{\tau_{h-1} \sim \mathbf{P}_{f^*,h-1}^{\pi^t,\nu^t}} \sum_{j=1}^{O} \sum_{o,a,b} |w_{h,t,j,o,a,b}^\top x_{\tau_{h-1}, \sigma^t, o, a, b}| \right\}$$

$$\le \sum_{o_h, a_h, b_h} \left\{ O_\varsigma \left[ \frac{\alpha^2}{S} T + \sum_{t=1}^{T} \sum_{\iota=1}^{t-1} \mathbb{E}_{\tau_{h-1} \sim \mathbf{P}_{f^*,h-1}^{\pi^\iota,\nu^\iota}} \left( \| \left( \boldsymbol{M}_h^t(o_h, a_h, b_h) - \boldsymbol{M}_h(o_h, a_h, b_h) \right) \bar{\boldsymbol{q}}(\tau_{h-1}) \|_1 \right. \right. \right.$$

$$\left. \left. \left. \cdot \, \sigma^\iota(o_h, a_h, b_h; \tau_{h-1}) \right)^2 \right]^{\frac{1}{2}} \right\},$$

where $\varsigma = 2\log\left(1 + \frac{4A^2 B^2 O^2 S^2}{\alpha^2}\right)$. By Jensen's inequality, the last term above is further relaxed as

$$\left\{ O^2 AB\varsigma \left[ \frac{OAB\alpha^2}{S} T + \sum_{t=1}^{T} \sum_{\iota=1}^{t-1} \mathbb{E}_{\tau_{h-1} \sim \mathbf{P}_{f^*,h-1}^{\pi^\iota,\nu^\iota}} \left( \sum_{o_h, a_h, b_h} \| \left( \boldsymbol{M}_h^t(o_h, a_h, b_h) - \boldsymbol{M}_h(o_h, a_h, b_h) \right) \bar{\boldsymbol{q}}(\tau_{h-1}) \|_1 \right. \right. \right.$$

$$\left. \left. \left. \cdot \, \sigma^\iota(o_h, a_h, b_h; \tau_{h-1}) \right)^2 \right]^{\frac{1}{2}} \right\}.$$

Then, invoking Lemma 6, we obtain

$$\sum_{t=1}^{T} \min\left\{ \frac{\alpha}{\sqrt{S}}, \; \mathbb{E}_{\tau_{h-1} \sim \mathbf{P}_{f^*,h-1}^{\pi^t,\nu^t}} \sum_{j=1}^{O} \sum_{o,a,b} |w_{h,t,j,o,a,b}^\top x_{\tau_{h-1}, \sigma^t, o, a, b}| \right\}$$

$$\lesssim \left[ O^2 AB\varsigma \left( \frac{OAB\alpha^2}{S} T + \frac{S}{\alpha^2} \sum_{t=1}^{T} \sum_{\iota=1}^{t-1} D_{\mathrm{He}}^2 \left( \mathbf{P}_{f^t,h}^{\pi^\iota,\nu^\iota}, \mathbf{P}_{f^*,h}^{\pi^\iota,\nu^\iota} \right) \right) \right]^{\frac{1}{2}}, \tag{14}$$

We next focus on $\|\boldsymbol{q}_0^t - \boldsymbol{q}_0\|_1$. Using that $\|\boldsymbol{q}_0^t - \boldsymbol{q}_0\|_1 \le 1$ and $2(t-1) \ge t$ when $t \ge 2$, we obtain

$$\sum_{t=1}^{T} \|\boldsymbol{q}_0^t - \boldsymbol{q}_0\|_1 \le 1 + \sum_{t=2}^{T} \left[ \frac{2(t-1)}{t} \|\boldsymbol{q}_0^t - \boldsymbol{q}_0\|_1^2 \right]^{\frac{1}{2}}$$

$$\lesssim 1 + \left[ \sum_{t=2}^{T} \frac{2}{t} \right]^{\frac{1}{2}} \left[ \sum_{t=2}^{T} (t-1) D_{\mathrm{He}}^2(\boldsymbol{q}_0^t, \boldsymbol{q}_0) \right]^{\frac{1}{2}}$$

$$\lesssim 1 + \left[ \log T \sum_{t=1}^{T} \sum_{\iota=1}^{t-1} D_{\mathrm{He}}^2(\boldsymbol{q}_0^t, \boldsymbol{q}_0) \right]^{\frac{1}{2}}. \tag{15}$$

Thus, by invoking (14) and (15), we get the regret bound by

$$\left| \sum_{t=1}^{T} \left( V_{f^t}^{\pi^t,\nu^t} - V_{f^*}^{\pi^t,\nu^t} \right) \right|$$

$$\lesssim H \left( OABH\sqrt{O\varsigma T} + \sqrt{ \frac{O^2 AB\varsigma S^2 H}{\alpha^4} \sum_{h=0}^{H} \sum_{t=1}^{T} \sum_{\iota=1}^{t-1} D_{\mathrm{He}}^2(\mathbf{P}_{f^t,h}^{\pi^\iota,\nu^\iota}, \mathbf{P}_{f^*,h}^{\pi^\iota,\nu^\iota}) } \right)$$

$$\lesssim \left[ \frac{O^2 AB\varsigma S^2 H^3}{\alpha^4} \sum_{h=0}^{H} \sum_{t=1}^{T} \sum_{\iota=1}^{t-1} D_{\mathrm{He}}^2(\mathbf{P}_{f^t,h}^{\pi^\iota,\nu^\iota}, \mathbf{P}_{f^*,h}^{\pi^\iota,\nu^\iota}) \right]^{\frac{1}{2}} + (O^3 A^2 B^2 H^3 \varsigma \cdot HT)^{\frac{1}{2}}.$$

Since the above derivation is for any policy pair $(\pi^t, \nu^t)$, we can show that $\alpha$-revealing POMG is subsumed by the classes of both self-play and adversarial GEC class with a bounded $d_{\mathrm{GEC}}$. We finally prove

$$d_{\mathrm{GEC}} = O\left( \frac{O^3 A^2 B^2 H^3 S^2 \varsigma}{\alpha^4} \right),$$

where $\varsigma = 2\log\left(1 + \frac{4A^2 B^2 O^2 S^2}{\alpha^2}\right)$. This completes the proof. $\qquad\square$

### B.4.1 Lemmas for Proof of Proposition 6

Here, we present and prove all the lemmas used for the proof of Proposition 6. All the lemmas presented here follow the notations of Proposition 6.

**Lemma 3.** *In an $\alpha$-revealing POMG, for any policy pair $(\pi, \nu)$, any vector $x \in \mathbb{R}^O$, and any $\tau_{h-1}$, we have*

$$\sum_{\tau_{h:H}} |\boldsymbol{m}(\tau_{h:H})x|\sigma(\tau_{h:H}; \tau_{h-1}) \leq \frac{\sqrt{S}}{\alpha}\|x\|_1.$$

*Proof.* We first obtain

$$\sum_{\tau_{h:H}} |\boldsymbol{m}(\tau_{h:H})x|\sigma(\tau_{h:H}; \tau_{h-1})$$

$$= \sum_{\tau_{h:H}} |\boldsymbol{m}(\tau_{h:H})\mathbb{O}_h\mathbb{O}_h^{\dagger}x|\sigma(\tau_{h:H}; \tau_{h-1})$$

$$\leq \sum_{i=1}^{S}\sum_{\tau_{h:H}} |\boldsymbol{m}(\tau_{h:H})\mathbb{O}_h\boldsymbol{e}_i| \cdot |\boldsymbol{e}_i^{\top}\mathbb{O}_h^{\dagger}x|\sigma(\tau_{h:H}; \tau_{h-1}),$$

where $\boldsymbol{e}_i \in \mathbb{R}^S$ denotes the standard basis vector whose $i$-th element is 1 and 0 for others. Note that $\sum_{\tau_{h:H}} |\boldsymbol{m}(\tau_{h:H})\mathbb{O}_h\boldsymbol{e}_i|\sigma(\tau_{h:H}; \tau_{h-1}) = \sum_{\tau_{h:H}} \Pr^{\sigma}(\tau_{h:H} \mid \tau_{1:h-1}, s_i) = 1$. Thus we obtain

$$\sum_{i=1}^{S}\sum_{\tau_{h:H}} |\boldsymbol{m}(\tau_{h:H})\mathbb{O}_h\boldsymbol{e}_i| \cdot |\boldsymbol{e}_i^{\top}\mathbb{O}_h^{\dagger}x|\sigma(\tau_{h:H}; \tau_{h-1})$$

$$= \|\mathbb{O}_h^{\dagger}x\|_1 \leq \|\mathbb{O}_h^{\dagger}\|_1 \cdot \|x\|_1 \leq \sqrt{S}\|\mathbb{O}_h^{\dagger}\|_2 \cdot \|x\|_1 \leq \frac{\sqrt{S}}{\alpha}\|x\|_1,$$

which completes the proof. $\qquad\square$

**Lemma 4.** *In an $\alpha$-revealing POMG, for any policy pair $(\pi, \nu)$ and any $x \in \mathbb{R}^O$, we have*

$$\sum_{(o_h, a_h, b_h)\in\mathcal{O}\times\mathcal{A}\times\mathcal{B}} \|\boldsymbol{M}_h(o_h, a_h, b_h)x\|_1\sigma(o_h, a_h, b_h) \leq \frac{\sqrt{S}}{\alpha}\|x\|_1.$$

*Proof.* We first obtain

$$\sum_{(o_h, a_h, b_h)\in\mathcal{O}\times\mathcal{A}\times\mathcal{B}} \|\boldsymbol{M}_h(o_h, a_h, b_h)x\|_1\sigma(o_h, a_h, b_h)$$

$$= \sum_{o_h, a_h, b_h}\sum_{j=1}^{O} |\widetilde{\boldsymbol{e}}_j^{\top}\boldsymbol{M}_h(o_h, a_h, b_h)x|\sigma(o_h, a_h, b_h)$$

$$\leq \sum_{o_h, a_h, b_h}\sum_{j=1}^{O}\sum_{i=1}^{S} |\widetilde{\boldsymbol{e}}_j^{\top}\boldsymbol{M}_h(o_h, a_h, b_h)\mathbb{O}_h\boldsymbol{e}_i| \cdot |\boldsymbol{e}_i^{\top}\mathbb{O}_h^{\dagger}x|\sigma(o_h, a_h, b_h),$$

where $\widetilde{\boldsymbol{e}}_j$ is the $j$-th standard basis vector in $\mathbb{R}^O$ and $\boldsymbol{e}_i$ is the $i$-th standard basis vector in $\mathbb{R}^S$. Note that we have

$$\sum_{o_h, a_h, b_h}\sum_{j=1}^{O} |\widetilde{\boldsymbol{e}}_j^{\top}\boldsymbol{M}_h(o_h, a_h, b_h)\mathbb{O}_h\boldsymbol{e}_i|\sigma(o_h, a_h, b_h)$$

$$= \sum_{o_h, a_h, b_h}\sum_{j=1}^{O} \Pr(o_j, o_h, a_h, b_h \mid s_i)\sigma(o_h, a_h, b_h)$$

$$= \sum_{o_h, a_h, b_h} \Pr(o_h, a_h, b_h \mid s_i)\sigma(o_h, a_h, b_h)$$

$$\leq \sum_{o_h, a_h, b_h} \sigma(o_h, a_h, b_h) = 1.$$

Combining the above result, we obtain

$$\sum_{o_h, a_h, b_h} \sum_{j=1}^{O} \sum_{i=1}^{S} |\widetilde{e}_j^\top \boldsymbol{M}_h(o_h, a_h, b_h) \mathbb{O}_h \boldsymbol{e}_i| \cdot |\boldsymbol{e}_i^\top \mathbb{O}_h^\dagger x| \sigma(o_h, a_h, b_h)$$

$$\leq \sum_{i=1}^{S} |\boldsymbol{e}_i^\top \mathbb{O}_h^\dagger x|$$

$$= \|\mathbb{O}_h^\dagger x\|_1 \leq \|\mathbb{O}_h^\dagger\|_1 \cdot \|x\|_1 \leq \sqrt{S} \|\mathbb{O}_h^\dagger\|_2 \cdot \|x\|_1 \leq \frac{\sqrt{S}}{\alpha} \|x\|_1,$$

which completes the proof. $\qquad\square$

**Lemma 5.** *The value difference in a POMG can be decomposed as*

$$V_{f^t}^{\pi^t, \nu^t} - V_{f^*}^{\pi^t, \nu^t} \leq \frac{H\sqrt{S}}{\alpha} \sum_{h=1}^{H} \Big[ \sum_{\tau_h} \|(\boldsymbol{M}_h^t - \boldsymbol{M}_h) \boldsymbol{q}(\tau_{h-1})\|_1 \sigma^t(\tau_h) + \|q_0^t - q_0\|_1 \Big].$$

*Proof.* Denoting $r(\tau_H)$ as the sum of rewards from $h = 1$ to $H$ on $\tau_H$, we obtain

$$V_{f^t}^{\pi^t, \nu^t} - V_{f^*}^{\pi^t, \nu^t}$$

$$= \sum_{\tau_H} \Big( \mathbf{P}_{f^t}^{\pi^t, \nu^t}(\tau_H) - \mathbf{P}_{f^*}^{\pi^t, \nu^t}(\tau_H) \Big) r(\tau_H)$$

$$\leq H \sum_{\tau_H} |\mathbf{P}_{f^t}^{\pi^t, \nu^t}(\tau_H) - \mathbf{P}_{f^*}^{\pi^t, \nu^t}(\tau_H)|$$

$$= H \sum_{\tau_H} |\boldsymbol{M}_{H:1}^t(o_{1:H}, a_{1:H}, b_{1:H}) q_0^t - \boldsymbol{M}_{H:1}(o_{1:H}, a_{1:H}, b_{1:H}) q_0| \sigma^t(\tau_H).$$

Using the triangle inequality, we further obtain

$$H \sum_{\tau_H} |\boldsymbol{M}_{H:1}^t(o_{1:H}, a_{1:H}, b_{1:H}) q_0^t - \boldsymbol{M}_{H:1}(o_{1:H}, a_{1:H}, b_{1:H}) q_0| \sigma^t(\tau_H)$$

$$\leq H \sum_{\tau_H} \Big[ \sum_{h=1}^{H} |\boldsymbol{m}^t(\tau_{h+1:H}) \big( \boldsymbol{M}_h^t(o_h, a_h, b_h) - \boldsymbol{M}_h(o_h, a_h, b_h) \big) \boldsymbol{q}(\tau_{h-1})| \sigma^t(\tau_H)$$

$$+ |\boldsymbol{m}^t(\tau_H)(q_0^t - q_0)| \sigma^t(\tau_H) \Big],$$

where $\boldsymbol{m}(\tau_{h+1:H}) = \boldsymbol{M}_{H:h+1}(\tau_{h+1:H})$ and $\boldsymbol{q}(\tau_{h-1}) = \boldsymbol{M}_{h-1:1}(\tau_{h-1}) q_0$. Note that Lemma 3 shows that

$$\sum_{\tau_{h+1:H}} |\boldsymbol{m}^t(\tau_{h+1:H}) \big( \boldsymbol{M}_h^t(o_h, a_h, b_h) - \boldsymbol{M}_h(o_h, a_h, b_h) \big) \boldsymbol{q}(\tau_{h-1})| \sigma^t(\tau_{h+1:H}; \tau_{1:h})$$

$$\leq \frac{\sqrt{S}}{\alpha} \| \big( \boldsymbol{M}_h^t(o_h, a_h, b_h) - \boldsymbol{M}_h(o_h, a_h, b_h) \big) \boldsymbol{q}(\tau_{h-1})\|_1$$

and

$$\sum_{\tau_H} |\boldsymbol{m}^t(\tau_H)(q_0^t - q_0)| \sigma^t(\tau_H) \leq \frac{\sqrt{S}}{\alpha} \|q_0^t - q_0\|_1.$$

Thus we obtain

$$
H \sum_{\tau_H} \Big[ \sum_{h=1}^{H} |\boldsymbol{m}^t(\tau_{h+1:H}) \big( \boldsymbol{M}_h^t(o_h, a_h, b_h) - \boldsymbol{M}_h(o_h, a_h, b_h) \big) \boldsymbol{q}(\tau_{h-1})| \sigma^t(\tau_H)
$$

$$
+ |\boldsymbol{m}^t(\tau_H)(\boldsymbol{q}_0^t - \boldsymbol{q}_0)| \sigma^t(\tau_H) \Big]
$$

$$
= H \Big[ \sum_{h=1}^{H} \sum_{\tau_H} |\boldsymbol{m}^t(\tau_{h+1:H}) \big( \boldsymbol{M}_h^t(o_h, a_h, b_h) - \boldsymbol{M}_h(o_h, a_h, b_h) \big) \boldsymbol{q}(\tau_{h-1})| \sigma^t(\tau_H)
$$

$$
+ \sum_{\tau_H} |\boldsymbol{m}^t(\tau_H)(\boldsymbol{q}_0^t - \boldsymbol{q}_0)| \sigma^t(\tau_H) \Big]
$$

$$
\leq \frac{H\sqrt{S}}{\alpha} \sum_{h=1}^{H} \Big[ \sum_{\tau_h} \|(\boldsymbol{M}_h^t - \boldsymbol{M}_h)\boldsymbol{q}(\tau_{h-1})\|_1 \sigma(\tau_h) + \|\boldsymbol{q}_0^t - \boldsymbol{q}_0\|_1 \Big],
$$

which completes the proof. $\qquad\square$

The next lemma gives the bound of training error by Hellinger distances.

**Lemma 6.** *For any model $f^t$ and policy $\sigma^\iota$, we have*

$$
\sum_{h=1}^{H} \sum_{\iota=1}^{t-1} \mathop{\mathbb{E}}_{\tau_{h-1} \sim \mathbf{P}_{f^*,h-1}^{\pi^\iota,\nu^\iota}} \Big[ \sum_{o_h,a_h,b_h} \|\big( \boldsymbol{M}_h^t(o_h, a_h, b_h) - \boldsymbol{M}_h(o_h, a_h, b_h) \big) \bar{\boldsymbol{q}}(\tau_{h-1})\|_1 \sigma^\iota(o_h, a_h, b_h; \tau_{h-1}) \Big]^2
$$

$$
\lesssim \frac{S}{\alpha^2} \sum_{\iota=1}^{t-1} \sum_{h=1}^{H} D_{\mathrm{He}}^2(\mathbf{P}_{f^t,h}^{\pi^\iota,\nu^\iota}, \mathbf{P}_{f^*,h}^{\pi^\iota,\nu^\iota}).
$$

*where $\lesssim$ omits absolute constants.*

*Proof.* By using $(a+b)^2 \leq 2a^2 + 2b^2$ and the triangle inequality, we decompose the LHS into two parts to bound them separately. We have

$$
\sum_{\iota=1}^{t-1} \mathop{\mathbb{E}}_{\tau_{h-1} \sim \mathbf{P}_{f^*,h-1}^{\pi^\iota,\nu^\iota}} \Big[ \sum_{o_h,a_h,b_h} \|\big( \boldsymbol{M}_h^t(o_h, a_h, b_h) - \boldsymbol{M}_h(o_h, a_h, b_h) \big) \bar{\boldsymbol{q}}(\tau_{h-1})\|_1 \sigma^\iota(o_h, a_h, b_h; \tau_{h-1}) \Big]^2
$$

$$
\leq 2 \sum_{\iota=1}^{t-1} \underbrace{\Big[ \sum_{\tau_h} \|\boldsymbol{M}_h^t(o_h, a_h, b_h) \bar{\boldsymbol{q}}^t(\tau_{h-1}) - \boldsymbol{M}_h(o_h, a_h, b_h) \bar{\boldsymbol{q}}(\tau_{h-1})\|_1 \mathbf{P}(\tau_{h-1}) \sigma^\iota(\tau_h) \Big]^2}_{\text{Term(i)}}
$$

$$
+ 2 \sum_{\iota=1}^{t-1} \underbrace{\Big[ \sum_{\tau_h} \|\boldsymbol{M}_h^t(o_h, a_h, b_h) \big( \bar{\boldsymbol{q}}^t(\tau_{h-1}) - \bar{\boldsymbol{q}}(\tau_{h-1}) \big)\|_1 \mathbf{P}(\tau_{h-1}) \sigma^\iota(\tau_h) \Big]^2}_{\text{Term(ii)}}.
$$

For Term(i), we equivalently rewrite this term as

$$
\Big[ \sum_{\tau_h} \|\boldsymbol{M}_h^t(o_h, a_h, b_h) \bar{\boldsymbol{q}}^t(\tau_{h-1}) - \boldsymbol{M}_h(o_h, a_h, b_h) \bar{\boldsymbol{q}}(\tau_{h-1})\|_1 \mathbf{P}(\tau_{h-1}) \sigma^\iota(\tau_h) \Big]^2
$$

$$
= \Big[ \mathop{\mathbb{E}}_{\tau_{h-1} \sim \mathbf{P}_{f^*,h-1}^{\pi^\iota,\nu^\iota}} \sum_{o_h} \sum_{a_h,b_h} \|\boldsymbol{M}_h^t(o_h, a_h, b_h) \bar{\boldsymbol{q}}^t(\tau_{h-1}) - \boldsymbol{M}_h(o_h, a_h, b_h) \bar{\boldsymbol{q}}(\tau_{h-1})\|_1
$$

$$
\cdot \sigma^\iota(o_h, a_h, b_h; \tau_{h-1}) \Big]^2.
$$

We first consider the case when $h < H$. According to the definitions of $\bar{q}_h^t$ and $\bar{q}_h$, we obtain

$$\text{Term(i)} = \Big[ \mathop{\mathbb{E}}_{\tau_{h-1} \sim \mathbf{P}_{f^*,h-1}^{\pi^\iota,\nu^\iota}} \sum_{o_h} \sum_{a_h,b_h} \|M_h^t(o_h,a_h,b_h)\bar{q}^t(\tau_{h-1}) - M_h(o_h,a_h,b_h)\bar{q}(\tau_{h-1})\|_1$$

$$\cdot \sigma^\iota(o_h,a_h,b_h;\tau_{h-1})\Big]^2$$

$$= \Big[ \mathop{\mathbb{E}}_{\tau_{h-1} \sim \mathbf{P}_{f^*,h-1}^{\pi^\iota,\nu^\iota}} \sum_{o_h,a_h,b_h} \sum_{i=1}^{O} \Big| \Pr_{f^t}(o_h, o_i \mid \tau_{1:h-1}, \mathbf{do}(a_h,b_h))$$

$$- \Pr_{f^*}(o_h, o_i \mid \tau_{h-1}, \mathbf{do}(a_h,b_h)) \Big| \sigma^\iota(o_h,a_h,b_h;\tau_{h-1})\Big]^2$$

$$= \Big[ \sum_{\tau_{h-1},o_h} |\Pr_{f^t}^{\pi^\iota,\nu^\iota}(\tau_{h-1},o_h) - \Pr_{f^*}^{\pi^\iota,\nu^\iota}(\tau_{h-1},o_h)| \Big]^2$$

$$\leq \|\mathbf{P}_{f^t,h}^{\pi^\iota,\nu^\iota}(\cdot) - \mathbf{P}_{f^*,h}^{\pi^\iota,\nu^\iota}(\cdot)\|_1^2 \leq 8D_{\text{He}}^2\big(\mathbf{P}_{f^t,h}^{\pi^\iota,\nu^\iota}(\cdot), \mathbf{P}_{f^*,h}^{\pi^\iota,\nu^\iota}(\cdot)\big),$$

where $\Pr_{f^t}^{\pi^\iota,\nu^\iota}(\tau_{h-1},o_h)$ denotes the probability of $(\tau_{h-1},o_h)$ if the actions are taken following $(\pi^\iota,\nu^\iota)$ and the observations follow the omission and transition process in the model $f^t$. The first inequality is by the fact that the $L_1$ difference of two marginal distributions is no more than the $L_1$ difference of two uniformed distributions, and the second inequality is due to $\|P - Q\|_1^2 \leq 8D_{\text{He}}^2(P,Q)$. Similarly, when $h = H$, Term(i) can be bounded as

$$\text{Term(i)} = \Big[ \mathop{\mathbb{E}}_{\tau_{h-1} \sim \mathbf{P}_{f^*,h-1}^{\pi^\iota,\nu^\iota}} \sum_{o_h,a_h,b_h} |\Pr_{f^t}(o_H \mid \tau_{H-1}, \mathbf{do}(a_H,b_H)) - \Pr_{f^*}(o_H \mid \tau_{H-1}, \mathbf{do}(a_H,b_H))|$$

$$\cdot \sigma^\iota(o_H, a_H, b_H; \tau_{H-1})\Big]^2$$

$$= \Big[ \sum_{\tau_{H-1},o_H} |\Pr_{f^t}^{\pi^\iota,\nu^\iota}(\tau_{H-1},o_H) - \Pr_{f^*}^{\pi^\iota,\nu^\iota}(\tau_{H-1},o_H)| \Big]^2$$

$$\leq \|\mathbf{P}_{f^t,H}^{\pi^\iota,\nu^\iota}(\cdot) - \mathbf{P}_{f^*,H}^{\pi^\iota,\nu^\iota}(\cdot)\|_1^2 \leq 8D_{\text{He}}^2\big(\mathbf{P}_{f^t,H}^{\pi^\iota,\nu^\iota}(\cdot), \mathbf{P}_{f^*,H}^{\pi^\iota,\nu^\iota}(\cdot)\big).$$

For Term(ii), Lemma 4 is used to give an upper bound. Specifically, we obtain that

$$\text{Term(i)} = \Big[ \sum_{\tau_h} \|M_h^t(o_h,a_h,b_h)\big(\bar{q}^t(\tau_{h-1}) - \bar{q}(\tau_{h-1})\big)\|_1 \mathbf{P}(\tau_{h-1})\sigma^\iota(\tau_h) \Big]^2$$

$$\leq \frac{S}{\alpha^2} \Big[ \sum_{\tau_{h-1}} \|\bar{q}^t(\tau_{h-1}) - \bar{q}(\tau_{h-1})\|_1 \mathbf{P}(\tau_{h-1})\sigma^\iota(\tau_{h-1}) \Big]^2$$

$$\leq \frac{S}{\alpha^2} \|\mathbf{P}_{f^t,h}^{\pi^\iota,\nu^\iota} - \mathbf{P}_{f^t,h}^{\pi^\iota,\nu^\iota}\|_1^2 \leq \frac{8S}{\alpha^2} D_{\text{He}}^2(\mathbf{P}_{f^t,h}^{\pi^\iota,\nu^\iota}, \mathbf{P}_{f^t,h}^{\pi^\iota,\nu^\iota}).$$

Combining the above results completes the proof of this lemma. $\qquad\square$

**Lemma 7.** *For any $(o,a,b) \in \mathcal{O} \times \mathcal{A} \times \mathcal{B}$, we have*

$$\mathop{\mathbb{E}}_{\tau_{h-1} \sim \mathbb{P}^{\pi^t,\nu^t}} \|x_{\tau_{h-1},\sigma^t,o,a,b}\|_2^2 \leq S, \qquad \sum_{j=1}^{O} \|w_{h,t,j,o,a,b}\|_2 \leq \frac{2ABd\sqrt{S}}{\alpha}.$$

*Proof.* We prove the former statement first. By noting that the $\mathbb{O}_{h-1}^\dagger \bar{q}(\tau_{h-1}) = [\Pr(s_i \mid \tau_{h-1})]_{i=1}^{S}$ and $\sigma^t(o_h,a_h,b_h;\tau_{h-1}) \leq 1$, we obtain

$$\|x_{\tau_{h-1},\sigma^t,o_h,a_h,b_h}\|_2^2$$

$$= \|\mathbb{O}_{h-1}^\dagger \bar{q}(\tau_{h-1})\sigma^t(o_h,a_h,b_h;\tau_{h-1})\|_2^2$$

$$\leq \sum_{i=1}^{S} \Pr(s_i \mid \tau_{h-1})^2 \leq S.$$

Therefore the first statement in Lemma 7 is proven. To prove the second statement, we write

$$\sum_{j=1}^{O} \|w_{h,t,j,o,a,b}\|_2$$

$$\leq \sum_{o,a,b} \sum_{j=1}^{O} \|w_{h,t,j,o,a,b}\|_2$$

$$\leq \sum_{o,a,b} \sum_{j=1}^{O} \|w_{h,t,j,o,a,b}\|_1$$

$$= \sum_{o,a,b} \sum_{i=1}^{O} \|(\boldsymbol{M}_h^t(o_h, a_h, b_h) - \boldsymbol{M}_h(o_h, a_h, b_h))\mathbb{O}_{h-1}\widetilde{\boldsymbol{e}}_i\|_1,$$

where the second inequality is due to $\|\cdot\|_2 \leq \|\cdot\|_1$, and $\widetilde{\boldsymbol{e}}_i$ stands for the $i$-th standard basis vector in $\mathbb{R}^O$. Then we apply Lemma 4 to obtain

$$\sum_{o,a,b} \sum_{i=1}^{O} \|(\boldsymbol{M}_h^t(o_h, a_h, b_h) - \boldsymbol{M}_h(o_h, a_h, b_h))\mathbb{O}_{h-1}\widetilde{\boldsymbol{e}}_i\|_1$$

$$\leq \frac{2AB\sqrt{S}}{\alpha} \sum_{i=1}^{O} \|\mathbb{O}_{h-1}\widetilde{\boldsymbol{e}}_i\|_1 \leq \frac{2ABO\sqrt{S}}{\alpha}\|\mathbb{O}_{h-1}\|_1 = \frac{2ABO\sqrt{S}}{\alpha},$$

where the second inequality is by $\|Ax\|_1 \leq \|A\|_1\|x\|_1$. The proof is completed. $\qquad\square$

## B.5   Decodable POMG

In this section, we propose a new class of POMGs, dubbed decodable POMG, by generalizing decodable POMDPs [23, 22] from the single-agent setting to the multi-agent setting.

**Definition 9** (Decodable POMG). *We say a POMG is a decodable POMG if an unknown decoder function $\phi_h$ exists, which recovers the state at step $h$ from the observation at step $h$. We have for any $1 \leq h \leq H$ that*

$$\phi_h(o_h) = s_h.$$

Given the decoder, we can define the transition from observation to observation as follows,

$$\mathbb{P}_h(o_{h+1} \mid o_h, a_h, b_h)$$
$$= \sum_{s_{h+1} \in \mathcal{S}} \mathbb{O}(o_{h+1} \mid s_{h+1})\mathbb{P}(s_{h+1} \mid s_h = \phi_h(o_h), a_h, b_h).$$

Our next proposition will show that the class of decodable POMGs is subsumed by the class of MGs with low self-play and adversarial GEC.

**Proposition 7** (Decodable POMG $\subset$ Low Self-Play/Adversarial GEC). *For a decodable POMG as defined in Definition 9, with the loss $\ell(f, \xi_h)$ defined in Definition 3 for POMGs, we have*

*Decodable POMG $\subset$ POMG with low self-play and adversarial GEC*

*with $d_{\mathrm{GEC}}$ satisfying*

$$d_{\mathrm{GEC}} = O(H^3|\mathcal{O}|^3|\mathcal{A}|^2|\mathcal{B}|^2\varsigma),$$

*where $\varsigma = 2\log\left(1 + 4|\mathcal{A}|^2|\mathcal{B}|^2|\mathcal{O}|^2|\mathcal{S}|\right).$*

*Proof.* The proof of this proposition is largely the same as the proof of Proposition 6, except that the coefficient before $\|x\|_1$ in Lemma 3 and 4 is changed, from $\frac{\sqrt{S}}{\alpha}$ to 1, as is proved in Lemma 8 and Lemma 9. Following the proof of Proposition 6 and using Lemmas 8 and 9 yields our result. $\qquad\square$

### B.5.1 Lemmas for Poof of Proposition 7

We present the lemmas for Proposition 7. These lemmas are analogous to Lemma 3 and Lemma 4. For convenience in analysis, we define some notations as follows.

Similar to the case of $\alpha$-revealing POMG in Appendix B.4, we define $\boldsymbol{q}_0$, $\tau_{h:h'}$, $\boldsymbol{q}(\tau_h)$ and $\bar{\boldsymbol{q}}(\tau_h)$. We now define a different observable operator as follows,

$$\boldsymbol{M}_h(o, a, b) := \mathbb{P}_{h,a,b}\mathrm{diag}(\boldsymbol{e}_o),$$

where $\boldsymbol{e}_o \in \mathbb{R}^O$ is the basis vector with only the $o$-th entry being 1. With these definitions, one can show that

$$\boldsymbol{q}(\tau_h) = \boldsymbol{M}_h(o_h, a_h, b_h)\boldsymbol{M}_{h-1}(o_{h-1}, a_{h-1}, b_{h-1})\dots\boldsymbol{M}_1(o_1, a_1, b_1)\boldsymbol{q}_0.$$

We define $\boldsymbol{M}_{h':h}(o_{h:h'}, a_{h:h'}, b_{h:h'}) := \boldsymbol{M}_{h'}(o_{h'}, a_{h'}, b_{h'})\boldsymbol{M}_{h'}(o_{h'-1}, a_{h'-1}, b_{h'-1})\dots\boldsymbol{M}_h(o_h, a_h, b_h)$ and rewrite $\boldsymbol{M}_{H:h+1}(o_{h+1:H}, a_{h+1:H}, b_{h+1:H})$ as $\boldsymbol{m}(o_{h+1:H}, a_{h+1:H}, b_{h+1:H})$ to emphasize that $\boldsymbol{m}(o_{h+1:H}, a_{h+1:H}, b_{h+1:H})$ is a vector since $\boldsymbol{M}_H(o_H, a_H, b_H)$ is a vector.

**Lemma 8.** *For any $x \in \mathbb{R}^O$, any policy pair $(\pi, \nu)$, and any $\tau_{h-1}$, we have*

$$\sum_{\tau_{h:H}} |\boldsymbol{m}(\tau_{h:H})x|\sigma(\tau_{h:H}; \tau_{h-1}) \leq \|x\|_1.$$

*Proof.* We can first bound the LHS as

$$\sum_{\tau_{h:H}} |\boldsymbol{m}(\tau_{h:H})x|\sigma(\tau_{h:H}; \tau_{h-1})$$

$$= \sum_{\tau_{h:H}} |\sum_{i=1}^O \widetilde{\boldsymbol{e}}_i^\top x \Pr(\tau_{h+1:H} \,|\, o_i)|\sigma(\tau_{h:H}; \tau_{h-1})$$

$$\leq \sum_{\tau_{h:H}} \sum_{i=1}^O |\widetilde{\boldsymbol{e}}_i^\top x| \Pr(\tau_{h+1:H} \,|\, o_i)\sigma(\tau_{h:H}; \tau_{h-1}),$$

where $\widetilde{\boldsymbol{e}}_i$ is the $i$-th basis vector of the space $\mathbb{R}^O$. Since $\sum_{\tau_{h+1:H}} \Pr(\tau_{h+1:H} \,|\, o_i)\sigma(\tau_{h:H}; \tau_{h-1}) \leq 1$, we further obtain

$$\sum_{\tau_{h:H}} \sum_{i=1}^O |\widetilde{\boldsymbol{e}}_i^\top x| \Pr(\tau_{h+1:H} \,|\, o_i)\sigma(\tau_{h:H}; \tau_{h-1})$$

$$\leq \sum_{i=1}^O |\widetilde{\boldsymbol{e}}_i^\top x| = \|x\|_1.$$

This completes the proof. $\qquad\square$

**Lemma 9.** *For any $x \in \mathbb{R}^O$, policy pair $(\pi, \nu)$ and $\tau_{h-1}$, we have*

$$\sum_{o\in\mathcal{O}, a\in\mathcal{A}, b\in\mathcal{B}} \|\boldsymbol{M}_h(o, a, b)x\|_1 \sigma(o, a, b; \tau_{h-1}) \leq \|x\|_1.$$

*Proof.* We can first show that

$$\sum_{o\in\mathcal{O}, a\in\mathcal{A}, b\in\mathcal{B}} \|\boldsymbol{M}_h(o, a, b)x\|_1 \sigma(o, a, b; \tau_{h-1})$$

$$= \sum_{o\in\mathcal{O}, a\in\mathcal{A}, b\in\mathcal{B}} \sum_{i=1}^O |\widetilde{\boldsymbol{e}}_i^\top \boldsymbol{M}_h(o, a, b)x|\sigma(o, a, b; \tau_{h-1})$$

$$\leq \sum_{o\in\mathcal{O}, a\in\mathcal{A}, b\in\mathcal{B}} \sum_{i=1}^O |\widetilde{\boldsymbol{e}}_i^\top x| \Pr(o_i \,|\, o, a, b)\sigma(o, a, b; \tau_{h-1}).$$

Since $\sum_{o\in\mathcal{O},a\in\mathcal{A},b\in\mathcal{B}} \Pr(o_i \,|\, o, a, b)\sigma(o, a, b; \tau_{h-1}) \leq 1$, we further obtain

$$\sum_{o\in\mathcal{O},a\in\mathcal{A},b\in\mathcal{B}} \sum_{i=1}^{O} |\widetilde{e}_i^\top x| \mathbb{P}(o_i \,|\, o, a, b)\sigma(o, a, b; \tau_{h-1})$$

$$\leq \sum_{i=1}^{O} |e_i^\top x| = \|x\|_1.$$

This concludes the proof. $\qquad\square$

## C   Computation of $\omega(\beta, p^0)$

In this section, we discuss the upper bound of the quantity $\omega(\beta, p^0)$ for both FOMGs and POMGs. For FOMGs, the analysis can be adopted from Lemma 2 in [2]. For POMGs, we give the detailed analysis to show the bound of $\omega(\beta, p^0)$, which is the first proof for the partially observable setting.

### C.1   Fully Observable Markov Game

We can adopt the result from Lemma 2 in [2] to give a bound for $\omega(\beta, p^0)$ in the context of fully observable Markov games.

**Proposition 8** (Lemma 2 of [2]). *When $\mathcal{F}$ is finite, $p^0$ is a uniform distribution over $\mathcal{F}$, then $\omega(\beta, p^0) \leq \log|\mathcal{F}|$. When $\mathcal{F}$ is infinite, suppose a transition kernel $\mathbb{P}_0$ satisfies $\mathbb{P}_{f^*} \ll \mathbb{P}_0$ and $\left\|\frac{d\mathbb{P}_{f^*}}{d\mathbb{P}_0}\right\|_\infty \leq B$, then for $\epsilon \leq 2/3$ and $B \geq \log(6B^2/\epsilon)$, there exists a prior $p^0$ on $\mathcal{F}$ such that*

$$\omega(\beta, p^0) \leq \beta\epsilon + \log\left(\mathcal{N}\left(\frac{\epsilon}{6\log(B/\nu)}\right)\right),$$

*where $\nu = \epsilon/(6\log(6B^2/\epsilon))$ and $\mathcal{N}(\epsilon)$ stands for the $\epsilon$-covering number w.r.t. the distance*

$$d(f, f') := \sup_{s,a,b,h} \left| D_{\mathrm{He}}^2\big(\mathbb{P}_{f,h}(\cdot\,|\,s,a,b), \mathbb{P}_{f^*,h}(\cdot\,|\,s,a,b)\big) - D_{\mathrm{He}}^2\big(\mathbb{P}_{f',h}(\cdot\,|\,s,a,b), \mathbb{P}_{f^*,h}(\cdot\,|\,s,a,b)\big) \right|.$$

To apply Proposition 8 in FOMGs, we can further show that $|D_{\mathrm{He}}^2(P, R) - D_{\mathrm{He}}^2(Q, R)| \leq \sqrt{2}D_{\mathrm{He}}^2(P, R)| \leq \frac{\sqrt{2}}{2}\|P - Q\|_1$ for distributions $P, Q, R$, so that the covering number under the distance $d$ can be bounded by the covering number w.r.t. the $\ell_1$ distance denoted as $\mathcal{N}_1(\epsilon)$, i.e.,

$$\mathcal{N}(\epsilon) \leq \mathcal{N}_1(\sqrt{2}\epsilon),$$

where $\mathcal{N}_1(\epsilon)$ is defined w.r.t. the distance

$$d_1(f, f') = \sup_{s,a,b,h} \|\mathbb{P}_{f,h}(\cdot\,|\,s,a,b) - \mathbb{P}_{f',h}(\cdot\,|\,s,a,b)\|_1.$$

The covering number under the $\ell_1$ distance is more common and is readily applicable to many problems.

Taking linear mixture MGs for an instance (defined in Definition 5), we calculate $\omega(4HT, p^0)$, which is the term quantifying the coverage of the initial sampling distribution in our theorems. We first obtain that

$$\sup_{s,a,b,h} \|P_{f,h}(\cdot\,|\,s,a,b) - P_{f',h}(\cdot\,|\,s,a,b)\|_1$$

$$= \sup_{s,a,b,h} \int_{s'\in\mathcal{S}} \phi(s', s, a, b)^\top \left(\boldsymbol{\theta}_{f,h} - \boldsymbol{\theta}_{f',h}\right) ds'$$

$$\leq \sup_{s,a,b,h} \int_{s'\in\mathcal{S}} \|\phi(s', s, a, b)\|_2 ds' \left\|\boldsymbol{\theta}_{f,h} - \boldsymbol{\theta}_{f',h}\right\|_2$$

$$\leq \sup_{h} \left\|\boldsymbol{\theta}_{f,h} - \boldsymbol{\theta}_{f',h}\right\|_2, \tag{16}$$

where the first inequality is by Cauchy-Schwarz, and the second inequality is due to $\int_{s'\in\mathcal{S}}\|\phi(s',s,a,b)\|_2 ds' \leq 1$. Since the model space is $\{\boldsymbol{\theta}: \sup_h \|\boldsymbol{\theta}_h\|_2 \leq B\}$, under the $\ell_2$ distance measure of

$$d_2(f,f') = \sup_h \left\|\boldsymbol{\theta}_{f,h} - \boldsymbol{\theta}_{f',h}\right\|_2,$$

we have

$$\mathcal{N}(\epsilon) \leq \mathcal{N}_1(\sqrt{2}\epsilon) \leq \left(1 + \frac{2\sqrt{2}B}{\varepsilon}\right)^{Hd}. \tag{17}$$

according to (16) and the covering number for an Euclidean ball. Combining this result (17) with Proposition 8 together, we can show that there exists a prior distribution $p^0$ with a sufficiently large $T$ such that

$$\omega(4HT, p^0) \lesssim Hd \log\left(1 + BT\log\left(T\log(T)\right)\right).$$

## C.2 Partially Observable Markov Game

In this subsection, we prove the bound of $\omega(\beta, p^0)$ for the partially observable setting, inspired by the proof of Proposition 8 (Lemma 2 in [2]).

**Proposition 9.** *When $\mathcal{F}$ is finite, $p^0$ is a uniform distribution over $\mathcal{F}$, then $\omega(\beta, p^0) \leq \log|\mathcal{F}|$. When $\mathcal{F}$ is infinite, suppose a distribution $\mathbf{P}_0$ satisfies that for any policy pair $(\pi, \nu)$, $\mathbf{P}_{f^*,H}^{\pi,\nu} \ll \mathbf{P}_0^{\pi,\nu}$ and $\|d\mathbf{P}_{f^*,H}^{\pi,\nu}/d\mathbf{P}_0^{\pi,\nu}\|_\infty \leq B$, where $B \geq 1$. Then for $\epsilon \leq 2/3$ and $B \geq \log(6B^2/\epsilon)$, there exists a prior $p^0$ on $\mathcal{F}$ such that*

$$\omega(\beta, p^0) \leq \beta\epsilon + \log\left(\mathcal{N}\left(\frac{\epsilon}{6\log(B/\nu)}\right)\right),$$

*where $\nu = \epsilon/(6\log(6B^2/\epsilon))$ and $\mathcal{N}(\cdot)$ is the covering number w.r.t. the distance*

$$d(f,f') = \sup_{\pi,\nu}\left|D_{\mathrm{He}}^2(\mathbf{P}_{f,H}^{\pi,\nu}, \mathbf{P}_{f^*,H}^{\pi,\nu}) - D_{\mathrm{He}}^2(\mathbf{P}_{f',H}^{\pi,\nu}, \mathbf{P}_{f^*,H}^{\pi,\nu})\right|.$$

The assumption in the theorem above covers the case when $\mathcal{S}$ is finite, where we choose $P_0$ to be uniform on $\mathcal{S}$ regardless of the policy. We also note that

$$\|d\mathbf{P}_{f^*,H}^{\pi,\nu}/d\mathbf{P}_0^{\pi,\nu}\|_\infty = \sup_{\tau_H}\left|\frac{d\mathbf{P}_{f^*,H}^{\pi,\nu}(\tau_H)}{d\mathbf{P}_0^{\pi,\nu}(\tau_H)}\right| = \sup_{\tau_H}\left|\frac{d\mathbf{P}_{f^*,H}(\tau_H)\sigma(\tau_H)}{d\mathbf{P}_0(\tau_H)\sigma(\tau_H)}\right| = \sup_{\tau_H}\left|\frac{d\mathbf{P}_{f^*,H}(\tau_H)}{d\mathbf{P}_0(\tau_H)}\right|,$$

which does not depend on the joint policy $\sigma = (\pi, \nu)$.

To apply the above proposition in POMGs, by the relation between different distances: $|D_{\mathrm{He}}^2(P,R) - D_{\mathrm{He}}^2(Q,R)| \leq \sqrt{2}D_{\mathrm{He}}^2(P,R)| \leq \frac{\sqrt{2}}{2}\|P-Q\|_1$ for distributions $P, Q, R$, we can show the covering number under the distance $d$ can be bounded by the covering number w.r.t. the $\ell_1$ distance denoted as $\mathcal{N}_1(\epsilon)$, i.e.,

$$\mathcal{N}(\epsilon) \leq \mathcal{N}_1(\sqrt{2}\epsilon),$$

where $\mathcal{N}_1(\epsilon)$ is defined w.r.t. the distance

$$d_1(f,f') := \sup_{\pi,\nu}\left\|\mathbf{P}_{f,H}^{\pi,\nu} - \mathbf{P}_{f',H}^{\pi,\nu}\right\|_1.$$

Such a covering number under $\ell_1$ distance is analyzed in the work [78], generalizing whose results gives that POMGs with different structures admit a log-covering number $\log\mathcal{N}_1(\epsilon) = \mathrm{ploy}(|\mathcal{O}|, |\mathcal{A}|, |\mathcal{B}|, |\mathcal{S}|, H, \log(1/\epsilon))$. We refer readers to [78] for detailed calculation of log-covering numbers under $d_1(f,f')$. Therefore, we can eventually show that $\omega(4HT, p^0) = \mathrm{ploy}(|\mathcal{O}|, |\mathcal{A}|, |\mathcal{B}|, |\mathcal{S}|, H, \log(HT))$. Next, we show the detailed proof for Proposition 9.

*Proof.* When $\mathcal{F}$ is finite and $p^0$ is the uniform distribution on $\mathcal{F}$, the proof is straightforward as we have $\omega(\beta, p^0) \leq \beta\varepsilon + \log|\mathcal{F}|$ for any $\varepsilon \geq 0$ and a uniform distribution $p^0$. Setting $\varepsilon$ to approach $0^+$ completes the proof.

When $\mathcal{F}$ is infinite, we start by setting up a $\gamma$-covering $\mathcal{C}(\gamma) \subset \mathcal{F}$ w.r.t. the distance $d(f, f') = \sup_{\pi,\nu} \left| D^2_{\mathrm{He}}(\mathbf{P}^{\pi,\nu}_{f,H}, \mathbf{P}^{\pi,\nu}_{f^*,H}) - D^2_{\mathrm{He}}(\mathbf{P}^{\pi,\nu}_{f',H}, \mathbf{P}^{\pi,\nu}_{f^*,H}) \right|$, where $\gamma > 0$ is a variable to be specified. Since $f^*$ is covered, an $\widetilde{f} \in \mathcal{C}(\gamma)$ satisfies $d(\widetilde{f}, f^*) = \sup_{\pi,\nu} D^2_{\mathrm{He}}(\mathbf{P}^{\pi,\nu}_{\widetilde{f},H}, \mathbf{P}^{\pi,\nu}_{f^*,H}) \leq \gamma$. We further define $\mathcal{C}_\nu(\gamma) := \{\nu\mathbf{P}_0 + (1-\nu)\mathbf{P}_f | f \in \mathcal{C}(\gamma)\}$ and also $\mathbf{P}_{f'} := \nu\mathbf{P}_0 + (1-\nu)\mathbf{P}_{\widetilde{f}}$. We note that $\sup_{\pi,\nu} \|\frac{\mathrm{d}\mathbf{P}^{\pi,\nu}_{f^*,H}}{\mathrm{d}\mathbf{P}^{\pi,\nu}_{f',H}}\|_\infty \leq \frac{B}{\nu}$. Then, we obtain

$$
\begin{aligned}
&D^2_{\mathrm{He}}(\mathbf{P}^{\pi,\nu}_{f',H}, \mathbf{P}^{\pi,\nu}_{f^*,H}) \\
&= 1 - \int \sqrt{\mathrm{d}(\nu\mathbf{P}^{\pi,\nu}_0 + (1-\nu)\mathbf{P}^{\pi,\nu}_{\widetilde{f},H})\mathrm{d}\mathbf{P}^{\pi,\nu}_{f^*,H}} \\
&\leq 1 - \left(\nu\int \sqrt{\mathrm{d}\mathbf{P}^{\pi,\nu}_0\mathrm{d}\mathbf{P}^{\pi,\nu}_{f^*,H}} + (1-\nu)\int \sqrt{\mathrm{d}\mathbf{P}^{\pi,\nu}_{\widetilde{f},H}\mathrm{d}\mathbf{P}^{\pi,\nu}_{f^*,H}}\right) \\
&= D^2_{\mathrm{He}}(\mathbf{P}^{\pi,\nu}_{\widetilde{f},H}, \mathbf{P}^{\pi,\nu}_{f^*,H}) + \nu\int \sqrt{\mathrm{d}\mathbf{P}^{\pi,\nu}_{\widetilde{f},H}\mathrm{d}\mathbf{P}^{\pi,\nu}_{f^*,H}} - \nu\int \sqrt{\mathrm{d}\mathbf{P}^{\pi,\nu}_0\mathrm{d}\mathbf{P}^{\pi,\nu}_{f^*,H}} \\
&\leq \gamma + \nu,
\end{aligned}
$$

where the first inequality uses Jensen's inequality and the second inequality is by $\sup_{\pi,\nu} D^2_{\mathrm{He}}(\mathbf{P}^{\pi,\nu}_{\widetilde{f},H}, \mathbf{P}^{\pi,\nu}_{f^*,H}) \leq \gamma$ and $0 \leq \int \sqrt{\mathrm{d}P\mathrm{d}Q} \leq 1$. To connect to the definition of $\mathcal{F}(\varepsilon)$, we further invoke Theorem 9 from [55] and obtain

$$
\mathrm{KL}(\mathbf{P}^{\pi,\nu}_{f^*,H} || \mathbf{P}^{\pi,\nu}_{f',H}) \leq \zeta(B/\nu)D^2_{\mathrm{He}}(\mathbf{P}^{\pi,\nu}_{f',H}, \mathbf{P}^{\pi,\nu}_{f^*,H})
$$

for any policy pair $(\pi, \nu)$, where $\zeta(b) \leq \max\left\{1, \frac{b\log b}{(1-\sqrt{b})^2}\right\}$ for $b > 1$. Plugging the above inequalities together, we obtain

$$
\mathrm{KL}(\mathbf{P}^{\pi,\nu}_{f^*,H} || \mathbf{P}^{\pi,\nu}_{f',H}) \leq \zeta(B/\nu)(\gamma + \nu). \tag{18}
$$

It remains to find a proper choice of $\gamma$ and $\nu$ to obtain $\zeta(B/\nu)(\gamma + \nu) \leq \varepsilon$. We choose $\nu = \varepsilon/\big(6\log(6B^2/\varepsilon)\big)$ and $\gamma = \varepsilon/\big(6\log(B/\nu)\big)$. Given the condition $B \geq \log(6B^2/\varepsilon)$, we have

$$
\nu = \frac{\varepsilon}{6\log(\frac{6B^2}{\varepsilon})} \leq \frac{\varepsilon}{6B},
$$

so that

$$
\nu = \frac{\varepsilon}{6\log(\frac{6B^2}{\varepsilon})} \leq \frac{\varepsilon}{6\log(\frac{B}{\nu})} = \gamma. \tag{19}
$$

Given $\varepsilon \leq 2/3$ and $B \geq 1$, we obtain

$$
\nu \leq \frac{\varepsilon}{6B} \leq \frac{1}{9B} \leq \frac{B}{9},
$$

namely $B/\nu \geq 9$. We note that when $b > 9$, it holds that

$$
\zeta(b) \leq \frac{b\log b}{(1-\sqrt{b})^2} \leq \frac{b\log b}{b - 2\sqrt{b}} \leq 3\log b.
$$

Thus we have $\zeta(B/\nu) \leq 3\log(B/\nu)$. Finally, we obtain

$$
\zeta(B/\nu)(\gamma + \nu) \leq 2\gamma\zeta(B/\nu) = \frac{\varepsilon}{3\log(\frac{B}{\nu})}\zeta(B/\nu) \leq \varepsilon,
$$

where the first inequality uses (19), the first equation uses the choice of $\gamma$, and the second inequality uses $\zeta(B/\nu) \leq 3\log(B/\nu)$. Choosing $p^0$ to be a uniform distribution on $\mathcal{C}_\nu(\gamma)$ completes the proof. $\qquad\square$

# D Technical Lemmas for Main Theorems

In this section, we first provide several important supporting lemmas used in the proofs of Theorems 1 and 2. We then present detailed proofs for these lemmas.

## D.1 Lemmas

**Lemma 10.** *Let $\upsilon$ be any probability distribution over $f \in \mathcal{F}$ where $\mathcal{F}$ is an arbitrary set. Then, $\mathbb{E}_{f \sim \upsilon(\cdot)}[G(f) + \log \upsilon(f)]$ is minimized at $\upsilon(f) \propto \exp(-G(f))$.*

*Proof.* This lemma is a corollary of Gibbs variational principle. For the detailed proof of this lemma, we refer the readers to the proof of Lemma 4.10 in [64]. This completes the proof. $\square$

The above lemma states that $\upsilon(f)$ in the above-described form solves the minimization problem $\min_{\upsilon \in \Delta(\mathcal{F})} \mathbb{E}_{f \sim \upsilon}[G(f) + \log \upsilon(f)]$, which helps to understand the design of the posterior sampling steps in our proposed algorithms. This lemma is also used in the proofs of the following two lemmas.

The following two lemmas provide the upper bounds for the expectation of the Hellinger distance by the likelihood functions defined in (1) and (2) for FOMGs and POMGs respectively.

**Lemma 11.** *Under the FOMG setting, for any $t \geq 1$, let $Z^t$ be the system randomness history up to the $t$-th episode, $p^t(\cdot|Z^{t-1})$ be any posterior distribution over the function class $\mathcal{F}$ with $p^0(\cdot)$ denoting an initial distribution, and $(\pi^t, \nu^t)$ be any Markovian policy pair for Player 1 and Player 2 depending on $f^t \sim p^t$. Suppose that $(s_h^t, a_h^t, b_h^t, s_{h+1}^t)$ is a data point sampled independently by executing the policy pair $(\pi^t, \nu^t)$ to the $h$-th step of the $t$-th episode. If we define $L_h^t(f) := \eta \log \mathbb{P}_{f,h}(s_{h+1}^t \mid s_h^t, a_h^t, b_h^t)$ with $\eta = 1/2$ as in (1), we have the following relation*

$$\sum_{h=1}^{H} \sum_{\iota=1}^{t-1} \mathbb{E}_{Z^{t-1}} \mathbb{E}_{f^t \sim p^t} \mathbb{E}_{(\pi^\iota, \nu^\iota, h)}[D_{\mathrm{He}}^2(\mathbb{P}_{f^t,h}(\cdot|s_h^\iota, a_h^\iota, b_h^\iota), \mathbb{P}_{f^*,h}(\cdot|s_h^\iota, a_h^\iota, b_h^\iota))]$$

$$\leq \mathbb{E}_{Z^{t-1}} \mathbb{E}_{f^t \sim p^t} \left[ -\sum_{h=1}^{H} \sum_{\iota=1}^{t-1} \left(L_h^\iota(f^t) - L_h^\iota(f^*)\right) + \log \frac{p^t(f^t)}{p^0(f^t)} \right],$$

*where $\mathbb{E}_{(\pi^t, \nu^t, h)}$ denotes taking an expectation over the data $(s_h^t, a_h^t, b_h^t)$ sampled following the policy pair $(\pi^t, \nu^t)$ and the true model $\mathbb{P}_{f^*}$ up to the $h$-th step at any $t$.*

*Proof.* Please see Appendix D.2 for a detailed proof. $\square$

**Lemma 12.** *Under the POMG setting, for any $t \geq 1$, let $Z^t$ be the system randomness history up to the $t$-th episode, $p^t(\cdot|Z^{t-1})$ be any posterior distribution over the function class $\mathcal{F}$ with $p^0(\cdot)$ denoting an initial distribution, and $(\pi^t, \nu^t)$ be any general history-dependent policy pair for Player 1 and Player 2 depending on $f^t \sim p^t$. Suppose that $\tau_h^t = (o_1^t, a_1^t, b_1^t \ldots, o_h^t, a_h^t, b_h^t)$ is a data point sampled independently by executing the policy pair $(\pi^t, \nu^t)$ to the $h$-th step of the $t$-th episode. If we define $L_h^t(f) := \eta \log \mathbb{P}_f(\tau_h^t)$ with $\eta = 1/2$ as in (2), we have the following relation*

$$\sum_{h=1}^{H} \sum_{\iota=1}^{t-1} \mathbb{E}_{Z^{t-1}} \mathbb{E}_{f^t \sim p^t}[D_{\mathrm{He}}^2(\mathbf{P}_{f^t,h}^{\pi^\iota, \nu^\iota}, \mathbf{P}_{f^*,h}^{\pi^\iota, \nu^\iota})]$$

$$\leq \mathbb{E}_{Z^{t-1}} \mathbb{E}_{f^t \sim p^t} \left[ -\sum_{h=1}^{H} \sum_{\iota=1}^{t-1} \left(L_h^\iota(f^t) - L_h^\iota(f^*)\right) + \log \frac{p^t(f^t)}{p^0(f^t)} \right],$$

*where $\mathbb{P}_{f,h}^{\pi, \nu}$ denotes the distribution for $\tau_h = (o_1, a_1, b_1, \ldots, o_h, a_h, b_h)$ under the model $\theta_f$ and the policy pair $(\pi, \nu)$ up to the $h$-th step.*

*Proof.* Please see Appendix D.3 for a detailed proof. $\square$

The next lemma shows that when the model is sufficiently close to $f^*$ with the distances employed in Definition 2, the following value differences are small enough under both FOMG and POMG settings.

**Lemma 13.** *If the model $f$ satisfies $\sup_{h,s,a,b} \mathrm{KL}^{\frac{1}{2}}(\mathbb{P}_{f^*,h}(\cdot\,|\,s,a,b)\|\mathbb{P}_{f,h}(\cdot\,|\,s,a,b)) \le \varepsilon$ for FOMGs and $\sup_{\pi,\nu} \mathrm{KL}^{\frac{1}{2}}(\mathbf{P}_{f^*,H}^{\pi,\nu}\|\mathbf{P}_{f,H}^{\pi,\nu}) \le \varepsilon$ for POMGs, we have that their corresponding value function satisfies*

$$V_{f^*}^* - V_f^* \le 3H\varepsilon, \qquad \sup_{\pi}(V_{f^*}^{\pi,*} - V_f^{\pi,*}) \le 3H\varepsilon, \qquad \sup_{\nu}(V_{f^*}^{*,\nu} - V_f^{*,\nu}) \le 3H\varepsilon.$$

*Proof.* Please see Appendix D.4 for detailed proof. $\qquad\square$

Finally, we show that when the model is sufficiently close to $f^*$, we will obtain that the following likelihood function difference is small under both FOMG and POMG settings.

**Lemma 14.** *If the model $f$ satisfies $\sup_{h,s,a,b} \mathrm{KL}^{\frac{1}{2}}(\mathbb{P}_{f^*,h}(\cdot\,|\,s,a,b)\|\mathbb{P}_{f,h}(\cdot\,|\,s,a,b)) \le \varepsilon$ for FOMGs and $\sup_{\pi,\nu} \mathrm{KL}^{\frac{1}{2}}(\mathbf{P}_{f^*,H}^{\pi,\nu}\|\mathbf{P}_{f,H}^{\pi,\nu}) \le \varepsilon$ for POMGs, we have that their corresponding likelihood function defined in (1) and (2) satisfies*

$$|\mathbb{E}(L_h^t(f) - L_h^t(f^*))| \le \eta\varepsilon^2,$$

*where the expectation is taken with respect to the randomness in $L_h^t$.*

*Proof.* Please see Appendix D.5 for detailed proof. $\qquad\square$

## D.2 Proof of Lemma 11

*Proof.* The proof of Lemma 11 can be viewed as a multi-agent generalization of the proof for Lemma E.5 in [83]. We start our proof by first considering the following equality

$$\mathbb{E}_{Z^{t-1}}\mathbb{E}_{f^t\sim p^t}\left[-\sum_{h=1}^{H}\sum_{\iota=1}^{t-1}\left(L_h^\iota(f^t) - L_h^\iota(f^*)\right) + \log\frac{p^t(f^t)}{p^0(f^t)}\right]$$

$$= \mathbb{E}_{Z^{t-1}}\mathbb{E}_{f^t\sim p^t}\left[\sum_{h=1}^{H}\sum_{\iota=1}^{t-1}\eta\log\frac{\mathbb{P}_{f^*,h}(s_{h+1}^\iota|s_h^\iota,a_h^\iota,b_h^\iota)}{\mathbb{P}_{f^t,h}(s_{h+1}^\iota|s_h^\iota,a_h^\iota,b_h^\iota)} + \log\frac{p^t(f^t)}{p^0(f^t)}\right].$$

Next, we lower bound RHS of the above equality. We define

$$\overline{L}_h^\iota(f) := \eta\log\frac{\mathbb{P}_{f,h}(s_{h+1}^\iota|s_h^\iota,a_h^\iota,b_h^\iota)}{\mathbb{P}_{f^*,h}(s_{h+1}^\iota|s_h^\iota,a_h^\iota,b_h^\iota)}, \qquad \widetilde{L}_h^\iota(f) := \overline{L}_h^\iota(f) - \log\mathbb{E}_{(\pi^\iota,\nu^\iota,\mathbb{P}_{f^*},h)}[\exp(\overline{L}_h^\iota(f))],$$

where $\mathbb{E}_{(\pi^\iota,\nu^\iota,\mathbb{P}_{f^*},h)}$ denotes taking an expectation over the data $(s_h^\iota, a_h^\iota, b_h^\iota, s_{h+1}^\iota)$ sampled following the policy pair $(\pi^\iota, \nu^\iota)$ and the true model $\mathbb{P}_{f^*}$ to the $h$-th step with $s_{h+1}^\iota \sim \mathbb{P}_{f^*,h}(\cdot|s_h^\iota, a_h^\iota, b_h^\iota)$ at round $\iota$. Then, we will show that $\mathbb{E}_{Z^{t-1}}[\exp(\sum_{h=1}^{H}\sum_{\iota=1}^{t-1}\widetilde{L}_h^\iota(f))] = 1$ by induction, following from [80]. Suppose that for any $k$, we have at $k-1$ that $\mathbb{E}_{Z^{k-1}}[\exp(\sum_{h=1}^{H}\sum_{\iota=1}^{k-1}\widetilde{L}_h^\iota(f))] = 1$. Then, at $k$, we have

$$\mathbb{E}_{Z^k}\left[\exp\left(\sum_{h=1}^{H}\sum_{\iota=1}^{k}\widetilde{L}_h^\iota(f)\right)\right] = \mathbb{E}_{Z^{k-1}}\left[\exp\left(\sum_{h=1}^{H}\sum_{\iota=1}^{k-1}\widetilde{L}_h^\iota(f)\right)\exp\left(\sum_{h=1}^{H}\widetilde{L}_h^k(f)\right)\right]$$

$$= \mathbb{E}_{Z^{k-1}}\left[\exp\left(\sum_{h=1}^{H}\sum_{\iota=1}^{k-1}\widetilde{L}_h^\iota(f)\right)\mathbb{E}_{f^k\sim p^k}\prod_{h=1}^{H}\mathbb{E}_{(\pi^k,\nu^k,\mathbb{P}_{f^*},h)}\exp\left(\widetilde{L}_h^k(f)\right)\right]$$

$$= \mathbb{E}_{Z^{k-1}}\left[\exp\left(\sum_{h=1}^{H}\sum_{\iota=1}^{k-1}\widetilde{L}_h^\iota(f)\right)\right] = 1,$$

where the second equality uses the fact that the data is sampled independently, the third equality is due to $\mathbb{E}_{(\pi^k,\nu^k,\mathbb{P}_{f^*},h)}\exp(\widetilde{L}_h^k(f)) = \mathbb{E}_{(\pi^k,\nu^k,\mathbb{P}_{f^*},h)}\exp(\overline{L}_h^k(f))/\mathbb{E}_{(\pi^k,\nu^k,\mathbb{P}_{f^*},h)}[\exp(\overline{L}_h^k(f))] = 1$ by the definition of $\widetilde{L}_h^k(f)$, and the last equality is by $\mathbb{E}_{Z^{k-1}}[\exp(\sum_{h=1}^{H}\sum_{\iota=1}^{k-1}\widetilde{L}_h^\iota(f))] = 1$ in the above assumption. Moreover, for $k = 1$, we have a trivial result that $\mathbb{E}_{Z^1}[\exp(\sum_{h=1}^{H}\widetilde{L}_h^1(f))] =$

$\mathbb{E}_{f^1 \sim p^1} \prod_{h=1}^{H} \mathbb{E}_{(\pi^1, \nu^1, \mathbb{P}_{f^*}, h)} \exp(\widetilde{L}_h^1(f)) = 1$. Consequently, we conclude that for any $k$, the above equality holds. Then, when $k = t - 1$, we have

$$\mathbb{E}_{Z^{t-1}} \left[ \exp \left( \sum_{h=1}^{H} \sum_{\iota=1}^{t-1} \widetilde{L}_h^\iota(f) \right) \right] = 1. \tag{20}$$

Furthermore, we have

$$\mathbb{E}_{Z^{t-1}} \mathbb{E}_{f^t \sim p^t} \left[ - \sum_{h=1}^{H} \sum_{\iota=1}^{t-1} \widetilde{L}_h^\iota(f^t) + \log \frac{p^t(f^t)}{p^0(f^t)} \right]$$

$$\geq \mathbb{E}_{Z^{t-1}} \inf_p \mathbb{E}_{f \sim p} \left[ - \sum_{h=1}^{H} \sum_{\iota=1}^{t-1} \widetilde{L}_h^\iota(f) + \log \frac{p(f)}{p^0(f)} \right]$$

$$= -\mathbb{E}_{Z^{t-1}} \log \mathbb{E}_{f \sim p^0} \exp \left[ \sum_{h=1}^{H} \sum_{\iota=1}^{t-1} \widetilde{L}_h^\iota(f) \right]$$

$$\geq -\log \mathbb{E}_{f \sim p^0} \mathbb{E}_{Z^{t-1}} \exp \left[ \sum_{h=1}^{H} \sum_{\iota=1}^{t-1} \widetilde{L}_h^\iota(f) \right] = 0,$$

where the last inequality is by Jensen's inequality and the last equality is due to (20). For the first equality, we use the fact that the following distribution is the minimizer

$$p(f) \propto \exp \left( \sum_{h=1}^{H} \sum_{\iota=1}^{t-1} \widetilde{L}_h^\iota(f) + \log p^0(f) \right) = p^0(f) \exp \left( \sum_{h=1}^{H} \sum_{\iota=1}^{t-1} \widetilde{L}_h^\iota(f) \right)$$

$$\iff p(f) = \frac{p^0(f) \exp \left( \sum_{h=1}^{H} \sum_{\iota=1}^{t-1} \widetilde{L}_h^\iota(f) \right)}{\int_{\mathcal{F}} p^0(f) \exp \left( \sum_{h=1}^{H} \sum_{\iota=1}^{t-1} \widetilde{L}_h^\iota(f) \right) df} = \frac{p^0(f) \exp \left( \sum_{h=1}^{H} \sum_{\iota=1}^{t-1} \widetilde{L}_h^\iota(f) \right)}{\mathbb{E}_{f \sim p^0} \left[ \exp \left( \sum_{h=1}^{H} \sum_{\iota=1}^{t-1} \widetilde{L}_h^\iota(f) \right) \right]}$$

according to Lemma 10, such that plugging in the above distribution leads to the first equality. Thus, according to the definitions of $\widetilde{L}_h^t$ and $\overline{L}_h^t$, we have

$$\mathbb{E}_{Z^{t-1}} \mathbb{E}_{f^t \sim p^t} \left[ - \sum_{h=1}^{H} \sum_{\iota=1}^{t-1} \eta \log \frac{\mathbb{P}_{f,h}(s_{h+1}^\iota | s_h^\iota, a_h^\iota, b_h^\iota)}{\mathbb{P}_{f^*,h}(s_{h+1}^\iota | s_h^\iota, a_h^\iota, b_h^\iota)} + \log \frac{p^t(f^t)}{p^0(f^t)} \right]$$

$$\geq \mathbb{E}_{Z^{t-1}} \mathbb{E}_{f^t \sim p^t} \left[ - \sum_{h=1}^{H} \sum_{\iota=1}^{t-1} \log \mathbb{E}_{(\pi^\iota, \nu^\iota, \mathbb{P}_{f^*}, h)} \exp \left( \eta \log \frac{\mathbb{P}_{f,h}(s_{h+1}^\iota | s_h^\iota, a_h^\iota, b_h^\iota)}{\mathbb{P}_{f^*,h}(s_{h+1}^\iota | s_h^\iota, a_h^\iota, b_h^\iota)} \right) \right].$$

Moreover, to further lower bound the RHS of the above inequality, by the inequality that $\log x \leq x - 1$ and the setting $\eta = \frac{1}{2}$, we have

$$-\log \mathbb{E}_{(\pi^\iota, \nu^\iota, \mathbb{P}_{f^*}, h)} \exp \left( \log \frac{\sqrt{\mathbb{P}_{f,h}(s_{h+1}^\iota | s_h^\iota, a_h^\iota, b_h^\iota)}}{\sqrt{\mathbb{P}_{f^*,h}(s_{h+1}^\iota | s_h^\iota, a_h^\iota, b_h^\iota)}} \right)$$

$$= -\log \mathbb{E}_{(\pi^\iota, \nu^\iota, \mathbb{P}_{f^*}, h)} \frac{\sqrt{\mathbb{P}_{f,h}(s_{h+1}^\iota | s_h^\iota, a_h^\iota, b_h^\iota)}}{\sqrt{\mathbb{P}_{f^*,h}(s_{h+1}^\iota | s_h^\iota, a_h^\iota, b_h^\iota)}}$$

$$\geq 1 - \mathbb{E}_{(\pi^\iota, \nu^\iota, \mathbb{P}_{f^*}, h)} \frac{\sqrt{\mathbb{P}_{f,h}(s_{h+1}^\iota | s_h^\iota, a_h^\iota, b_h^\iota)}}{\sqrt{\mathbb{P}_{f^*,h}(s_{h+1}^\iota | s_h^\iota, a_h^\iota, b_h^\iota)}}$$

$$= 1 - \mathbb{E}_{(\pi^\iota, \nu^\iota, h)} \int_{\mathcal{S}} \sqrt{\mathbb{P}_{f,h}(s | s_h^\iota, a_h^\iota, b_h^\iota) \mathbb{P}_{f^*,h}(s | s_h^\iota, a_h^\iota, b_h^\iota)} ds$$

$$= \mathbb{E}_{(\pi^\iota, \nu^\iota, h)} [D_{\text{He}}^2(\mathbb{P}_{f,h}(\cdot | s_h^\iota, a_h^\iota, b_h^\iota), \mathbb{P}_{f^*,h}(\cdot | s_h^\iota, a_h^\iota, b_h^\iota))],$$

where the last equality is by $D_{\text{He}}^2(P,Q) = \frac{1}{2}\int(\sqrt{\mathrm{d}P(x)} - \sqrt{\mathrm{d}Q(x)})^2 = 1 - \int\sqrt{\mathrm{d}P(x)\mathrm{d}Q(x)}$.
Here $\mathbb{E}_{(\pi^\iota,\nu^\iota,h)}$ denotes taking expectation over $(s_h^\iota, a_h^\iota, b_h^\iota)$ sampled following the policy pair $(\pi^\iota, \nu^\iota)$
and the true model $\mathbb{P}_{f^*}$ to the $h$-th step but without the next state $s_{h+1}^\iota$ generated by $\mathbb{P}_{f^*,h}(\cdot|s_h^\iota, a_h^\iota, b_h^\iota)$
at round $\iota$. In the sequel, combining the above results, we have

$$\mathbb{E}_{Z^{t-1}}\mathbb{E}_{f^t\sim p^t}\left[-\sum_{h=1}^{H}\sum_{\iota=1}^{t-1}\left(L_h^\iota(f^t) - L_h^\iota(f^*)\right) + \log\frac{p^t(f^t)}{p^0(f^t)}\right]$$
$$\geq \mathbb{E}_{Z^{t-1}}\mathbb{E}_{f^t\sim p^t}\left[\sum_{h=1}^{H}\sum_{\iota=1}^{t-1}\mathbb{E}_{(\pi^\iota,\nu^\iota,h)}[D_{\text{He}}^2(\mathbb{P}_{f,h}(\cdot|s_h^\iota,a_h^\iota,b_h^\iota),\mathbb{P}_{f^*,h}(\cdot|s_h^\iota,a_h^\iota,b_h^\iota))]\right].$$

This completes the proof. $\qquad\square$

### D.3    Proof of Lemma 12

*Proof.* The proof of Lemma 12 is similar to the proof of Lemma 11. We will give a brief description
of the main steps for our proof of Lemma 12. We start our proof by considering the following equality

$$\mathbb{E}_{Z^{t-1}}\mathbb{E}_{f^t\sim p^t}\left[-\sum_{h=1}^{H}\sum_{\iota=1}^{t-1}\left(L_h^\iota(f^t) - L_h^\iota(f^*)\right) + \log\frac{p^t(f^t)}{p^0(f^t)}\right]$$
$$= \mathbb{E}_{Z^{t-1}}\mathbb{E}_{f^t\sim p^t}\left[\sum_{h=1}^{H}\sum_{\iota=1}^{t-1}\eta\log\frac{\mathbf{P}_{f^*,h}(\tau_h^\iota)}{\mathbf{P}_{f^t,h}(\tau_h^\iota)} + \log\frac{p^t(f^t)}{p^0(f^t)}\right].$$

Next, we lower bound RHS of the above equality. We define

$$\overline{L}_h^\iota(f) := \eta\log\frac{\mathbf{P}_{f,h}(\tau_h^\iota)}{\mathbf{P}_{f^*,h}(\tau_h^\iota)}, \qquad \widetilde{L}_h^\iota(f) := \overline{L}_h^\iota(f) - \log\mathbb{E}_{(\pi^\iota,\nu^\iota,h)}[\exp(\overline{L}_h^\iota(f))],$$

where $\mathbb{E}_{(\pi^\iota,\nu^\iota,h)}$ denotes taking an expectation over the data $\tau_h^\iota$ sampled following the policy pair
$(\pi^\iota,\nu^\iota)$ and the true model $\theta_{f^*}$ to the $h$-th step at round $\iota$. Then, we can show that

$$\mathbb{E}_{Z^{t-1}}\mathbb{E}_{f^t\sim p^t}\left[-\sum_{h=1}^{H}\sum_{\iota=1}^{t-1}\widetilde{L}_h^\iota(f^t) + \log\frac{p^t(f^t)}{p^0(f^t)}\right] \geq 0,$$

following a similar derivation as (20) in the proof of Lemma 11. With setting $\eta = \frac{1}{2}$, this result
further leads to

$$-\log\mathbb{E}_{(\pi^\iota,\nu^\iota,h)}\exp\left(\log\frac{\sqrt{\mathbf{P}_{f,h}(\tau_h^\iota)}}{\sqrt{\mathbf{P}_{f^*,h}(\tau_h^\iota)}}\right)$$
$$\geq 1 - \mathbb{E}_{(\pi^\iota,\nu^\iota,h)}\frac{\sqrt{\mathbf{P}_{f,h}(\tau_h^\iota)}}{\sqrt{\mathbf{P}_{f^*,h}(\tau_h^\iota)}} = 1 - \mathbb{E}_{(\pi^\iota,\nu^\iota,h)}\frac{\sqrt{\mathbf{P}_{f,h}^{\pi^\iota,\nu^\iota}(\tau_h^\iota)}}{\sqrt{\mathbf{P}_{f^*,h}^{\pi^\iota,\nu^\iota}(\tau_h^\iota)}}$$
$$= 1 - \int_{(\mathcal{O}\times\mathcal{A})^h}\sqrt{\mathbf{P}_{f,h}^{\pi^\iota,\nu^\iota}(\tau_h)\mathbf{P}_{f^*,h}^{\pi^\iota,\nu^\iota}(\tau_h)}\mathrm{d}\tau_h$$
$$= D_{\text{He}}^2(\mathbf{P}_{f,h}^{\pi^\iota,\nu^\iota}, \mathbf{P}_{f^*,h}^{\pi^\iota,\nu^\iota}).$$

In the sequel, combining the above results, we have

$$\sum_{h=1}^{H}\sum_{\iota=1}^{t-1}\mathbb{E}_{Z^{t-1}}\mathbb{E}_{f^t\sim p^t}[D_{\text{He}}^2(\mathbb{P}_{f^t,h}^{\pi^\iota,\nu^\iota}, \mathbb{P}_{f^*,h}^{\pi^\iota,\nu^\iota})]$$
$$\leq \mathbb{E}_{Z^{t-1}}\mathbb{E}_{f^t\sim p^t}\left[-\sum_{h=1}^{H}\sum_{\iota=1}^{t-1}\left(L_h^\iota(f^t) - L_h^\iota(f^*)\right) + \log\frac{p^t(f^t)}{p^0(f^t)}\right].$$

This completes the proof. $\qquad\square$

### D.4 Proof of Lemma 13

*Proof.* We first prove the upper bound of $|V_{f^*}^{\pi,\nu} - V_f^{\pi,\nu}|$ for any $(\pi, \nu)$ and $f$ under FOMG and POMG settings separately.

For FOMGs, we let $V_{f^*}^{\pi,\nu} = V_{f^*}^{\pi,\nu}$ and $V_f^{\pi,\nu} = V_f^{\pi,\nu}$. Then, according to the Bellman equation that $Q_{f,h}^{\pi,\nu}(s,a,b) = r_h(s,a,b) + \langle \mathbb{P}_{f,h}(\cdot|s,a,b), V_{f,h+1}^{\pi,\nu}(\cdot) \rangle$ and also $V_{f,h}^{\pi,\nu}(s,a,b) = \mathbb{E}_{a \sim \pi_h(\cdot|s), b \sim \nu_h(\cdot|s)}[Q_{f,h}^{\pi,\nu}(s,a,b)]$, we have

$$\left| V_{f^*}^{\pi,\nu} - V_f^{\pi,\nu} \right|$$

$$\leq \mathbb{E}_{\pi,\nu} \left| \langle \mathbb{P}_{f^*}(\cdot|s_1,a_1,b_1), V_{f^*,2}^{\pi,\nu}(\cdot) \rangle - \langle \mathbb{P}_{f,1}(\cdot|s_1,a_1,b_1), V_{f,2}^{\pi,\nu}(\cdot) \rangle \right|$$

$$\leq H \mathbb{E}_{\pi,\nu} \left\| \mathbb{P}_{f^*}(\cdot|s_1,a_1,b_1) - \mathbb{P}_{f,1}(\cdot|s_1,a_1,b_1) \right\|_1 + \mathbb{E}_{\pi,\nu,\mathbb{P}_{f^*}} \left| V_{f^*,2}^{\pi,\nu}(s_2) - V_{f,2}^{\pi,\nu}(s_2) \right|$$

$$\vdots \quad \text{(recursively applying the above derivation)}$$

$$\leq H \mathbb{E}_{\pi,\nu,\mathbb{P}_{f^*}} \sum_{h=1}^{H} \left\| \mathbb{P}_{f^*,h}(\cdot|s_h,a_h,b_h) - \mathbb{P}_{f,h}(\cdot|s_h,a_h,b_h) \right\|_1,$$

which further leads to

$$\left| V_{f^*}^{\pi,\nu} - V_f^{\pi,\nu} \right|$$

$$\leq H \mathbb{E}_{\pi,\nu,\mathbb{P}_{f^*}} \sum_{h=1}^{H} \left\| \mathbb{P}_{f^*,h}(\cdot|s_h,a_h,b_h) - \mathbb{P}_{f,h}(\cdot|s_h,a_h,b_h) \right\|_1$$

$$\leq H^2 \sup_{h,s,a,b} \left\| \mathbb{P}_{f^*,h}(\cdot|s,a,b) - \mathbb{P}_{f,h}(\cdot|s,a,b) \right\|_1$$

$$\leq 3H^2 \sup_{h,s,a,b} \mathrm{KL}^{\frac{1}{2}}(\mathbb{P}_{f^*,h}(\cdot|s,a,b) \| \mathbb{P}_{f,h}(\cdot|s,a,b)) \leq 3H^2 \varepsilon,$$

where the third inequality is by Pinsker's inequality and the last inequality is by the assumption of this lemma. On the other hand, we can show that the above result also holds for POMGs. Then, for this setting, we have

$$\left| V_{f^*}^{\pi,\nu} - V_f^{\pi,\nu} \right| = \int_{(\mathcal{O} \times \mathcal{A} \times \mathcal{B})^H} (\mathbf{P}_{f^*,H}^{\pi,\nu}(\tau_H) - \mathbf{P}_{f,H}^{\pi,\nu}(\tau_H)) \left( \sum_{h=1}^{H} r_h(o_h,a_h,b_h) \right) \mathrm{d}\tau_H$$

$$= H \int_{(\mathcal{O} \times \mathcal{A} \times \mathcal{B})^H} \left| \mathbf{P}_{f^*,H}^{\pi,\nu}(\tau_H) - \mathbf{P}_{f,H}^{\pi,\nu}(\tau_H) \right| \mathrm{d}\tau_H$$

$$= H \| \mathbf{P}_{f^*,H}^{\pi,\nu}(\tau_H) - \mathbf{P}_{f,H}^{\pi,\nu}(\tau_H) \|_1$$

$$= 3H \sup_{\pi,\nu} \mathrm{KL}^{\frac{1}{2}}(\mathbf{P}_{f^*,H}^{\pi,\nu} \| \mathbf{P}_{f,H}^{\pi,\nu}) \leq 3H\varepsilon.$$

To unify our results, we enlarge the above bound by a factor of $H$ and eventually obtain that

$$\left| V_{f^*}^{\pi,\nu} - V_f^{\pi,\nu} \right| \leq 3H^2 \varepsilon,$$

for both FOMGs and POMGs. Moreover, by the properties of the operators $\min$, $\max$, and $\max \min$, we have

$$V_{f^*}^* - V_f^* = \max_\pi \min_\nu V_{f^*}^{\pi,\nu} - \max_\pi \min_\nu V_f^{\pi,\nu} \leq \sup_{\pi,\nu} |V_{f^*}^{\pi,\nu} - V_f^{\pi,\nu}| \leq 3H^2 \varepsilon.$$

$$\sup_\pi (V_{f^*}^{\pi,*} - V_f^{\pi,*}) = \sup_\pi (\min_\nu V_{f^*}^{\pi,\nu} - \min_\nu V_f^{\pi,\nu}) \leq \sup_{\pi,\nu} |V_{f^*}^{\pi,\nu} - V_f^{\pi,\nu}| \leq 3H^2 \varepsilon.$$

$$\sup_\nu (V_{f^*}^{*,\nu} - V_f^{*,\nu}) = \sup_\nu (\sum_\pi V_{f^*}^{\pi,\nu} - \sum_\pi V_f^{\pi,\nu}) \leq \sup_{\pi,\nu} |V_{f^*}^{\pi,\nu} - V_f^{\pi,\nu}| \leq 3H^2 \varepsilon.$$

This concludes the proof of this lemma. $\qquad \square$

### D.5 Proof of Lemma 14

*Proof.* We first prove the upper bound under the FOMG setting. By (1), we know that

$$L_h^t(f^*) - L_h^t(f) = \eta \log \mathbb{P}_{f^*,h}(s_{h+1}^t \mid s_h^t, a_h^t, b_h^t) - \eta \log \mathbb{P}_{f,h}(s_{h+1}^t \mid s_h^t, a_h^t, b_h^t)$$
$$= \eta \log \frac{\mathbb{P}_{f^*,h}(s_{h+1}^t \mid s_h^t, a_h^t, b_h^t)}{\mathbb{P}_{f,h}(s_{h+1}^t \mid s_h^t, a_h^t, b_h^t)},$$

which further leads to

$$|\mathbb{E}(L_h^t(f^*) - L_h^t(f))| = \eta \left| \mathbb{E} \log \frac{\mathbb{P}_{f^*,h}(s_{h+1}^t \mid s_h^t, a_h^t, b_h^t)}{\mathbb{P}_{f,h}(s_{h+1}^t \mid s_h^t, a_h^t, b_h^t)} \right|$$
$$= \eta \left| \mathbb{E}_{(s_h^t, a_h^t, b_h^t)} \mathbb{E}_{s_{h+1}^t \sim \mathbb{P}_{f^*,h}(\cdot \mid s_h^t, a_h^t, b_h^t)} \log \frac{\mathbb{P}_{f^*,h}(s_{h+1}^t \mid s_h^t, a_h^t, b_h^t)}{\mathbb{P}_{f,h}(s_{h+1}^t \mid s_h^t, a_h^t, b_h^t)} \right|$$
$$\leq \eta \sup_{s,a,b} \mathrm{KL}(\mathbb{P}_{f^*,h}(\cdot \mid s, a, b) \| \mathbb{P}_{f,h}(\cdot \mid s, a, b))$$
$$\leq \eta \varepsilon^2,$$

where we use the definition of KL divergence in the first inequality and the last inequality is by the condition that $\sup_{h,s,a,b} \mathrm{KL}^{\frac{1}{2}}(\mathbb{P}_{f^*,h}(\cdot \mid s, a, b) \| \mathbb{P}_{f,h}(\cdot \mid s, a, b)) \leq \varepsilon$.

We next prove the upper bound under the POMG setting. According to (2), we have

$$L_h^t(f^*) - L_h^t(f) = \eta \log \mathbf{P}_{f^*,h}(\tau_h^t) - \eta \log \mathbf{P}_{f,h}(\tau_h^t) = \eta \log \frac{\mathbf{P}_{f^*,h}(\tau_h^t)}{\mathbf{P}_{f,h}(\tau_h^t)},$$

which further leads to

$$|\mathbb{E}(L_h^t(f^*) - L_h^t(f))| = \eta \left| \mathbb{E}_{\tau_h^t \sim \mathbf{P}_{f^*,h}^{\pi^t,\nu^t}(\cdot)} \log \frac{\mathbf{P}_{f^*,h}(\tau_h^t)}{\mathbf{P}_{f,h}(\tau_h^t)} \right|$$
$$= \eta \left| \mathbb{E}_{\tau_h^t \sim \mathbf{P}_{f^*,h}^{\pi^t,\nu^t}(\cdot)} \log \frac{\mathbf{P}_{f^*,h}^{\pi^t,\nu^t}(\tau_h^t)}{\mathbf{P}_{f,h}^{\pi^t,\nu^t}(\tau_h^t)} \right|$$
$$= \eta \mathrm{KL}(\mathbf{P}_{f^*,h}^{\pi^t,\nu^t} \| \mathbf{P}_{f,h}^{\pi^t,\nu^t}),$$

where we use the definition of KL divergence and the relation between $\mathbf{P}_{f,h}$ and $\mathbf{P}_{f,h}^{\pi,\nu}$. Now we consider to lower bound $\mathrm{KL}(\mathbf{P}_{f^*,H}^{\pi^t,\nu^t} \| \mathbf{P}_{f,H}^{\pi^t,\nu^t})$. We have

$$\mathrm{KL}(\mathbf{P}_{f^*,H}^{\pi^t,\nu^t} \| \mathbf{P}_{f,H}^{\pi^t,\nu^t}) = \mathbb{E}_{\tau_H^t \sim \mathbf{P}_{f^*,H}^{\pi^t,\nu^t}(\cdot)} \log \frac{\mathbf{P}_{f^*,H}^{\pi^t,\nu^t}(\tau_H^t)}{\mathbf{P}_{f,H}^{\pi^t,\nu^t}(\tau_H^t)}$$
$$= \mathbb{E}_{\tau_H^t \sim \mathbf{P}_{f^*,H}^{\pi^t,\nu^t}(\cdot)} \left[ \log \frac{\mathbf{P}_{f^*,h}^{\pi^t,\nu^t}(\tau_h^t)}{\mathbf{P}_{f,h}^{\pi^t,\nu^t}(\tau_h^t)} + \log \frac{\mathrm{Pr}_{f^*}^{\pi^t,\nu^t}(\tau_{h+1:H}^t \mid \tau_h^t)}{\mathrm{Pr}_f^{\pi^t,\nu^t}(\tau_{h+1:H}^t \mid \tau_h^t)} \right]$$
$$= \mathrm{KL}(\mathbf{P}_{f^*,h}^{\pi^t,\nu^t} \| \mathbf{P}_{f,h}^{\pi^t,\nu^t}) + \mathbb{E}_{\tau_h^t \sim \mathbf{P}_{f^*,h}^{\pi^t,\nu^t}(\cdot)} \mathbb{E}_{\tau_{h+1:H}^t \sim \mathbf{P}_{f^*,h}^{\pi^t,\nu^t}(\cdot \mid \tau_h^t)} \log \frac{\mathrm{Pr}_{f^*}^{\pi^t,\nu^t}(\tau_{h+1:H}^t \mid \tau_h^t)}{\mathrm{Pr}_f^{\pi^t,\nu^t}(\tau_{h+1:H}^t \mid \tau_h^t)}$$
$$= \mathrm{KL}(\mathbf{P}_{f^*,h}^{\pi^t,\nu^t} \| \mathbf{P}_{f,h}^{\pi^t,\nu^t}) + \mathbb{E}_{\tau_h^t \sim \mathbf{P}_{f^*,h}^{\pi^t,\nu^t}(\cdot)} \mathrm{KL}(\mathrm{Pr}_{f^*}^{\pi^t,\nu^t}(\cdot \mid \tau_h^t) \| \mathrm{Pr}_f^{\pi^t,\nu^t}(\cdot \mid \tau_h^t))$$
$$\geq \mathrm{KL}(\mathbf{P}_{f^*,h}^{\pi^t,\nu^t} \| \mathbf{P}_{f,h}^{\pi^t,\nu^t}),$$

where $\mathrm{Pr}_f^{\pi^t,\nu^t}$ denote the probability under the model $f$ and the policy pair $\pi^t, \nu^t$, the third equality is by the definition of KL divergence, and the inequality is by the non-negativity of KL divergence. Therefore, combining the above results and the condition $\sup_{\pi,\nu} \mathrm{KL}^{\frac{1}{2}}(\mathbf{P}_{f^*,H}^{\pi,\nu} \| \mathbf{P}_{f,H}^{\pi,\nu}) \leq \varepsilon$, we obtain

$$|\mathbb{E}(L_h^t(f^*) - L_h^t(f))| \leq \eta \mathrm{KL}(\mathbf{P}_{f^*,H}^{\pi^t,\nu^t} \| \mathbf{P}_{f,H}^{\pi^t,\nu^t}) \leq \eta \varepsilon^2.$$

This concludes the proof of this lemma. $\square$

# E   Proofs for Algorithms 1 and 3

In this section, we provide the detailed proof for Theorem 1. In particular, our proof is compatible with both the FOMG and the POMG settings. Thus, we unify both setups in a single theorem and present their proof together in this section.

To characterize the value difference under models $f$ and $f^*$ we define the following terms, which are

$$\Delta V_f^{\pi,*}(s) := V_f^{\pi,*}(s) - V_{f^*}^{\pi,*}(s), \tag{21}$$

and

$$\Delta V_f^*(s) := V_f^*(s) - V_{f^*}^*(s). \tag{22}$$

In addition, we define the difference of likelihood functions at step $h$ of time $t$ as

$$\Delta L_h^t(f) = L_h^t(f) - L_h^t(f^*). \tag{23}$$

Then, the updating rules of the posterior distribution in Algorithm 1 have the following equivalent forms

$$p^t(f) \propto p^0(f) \exp\left[\gamma_1 V_f^* + \sum_{\iota=1}^{t-1}\sum_{h=1}^{H} L_h^\tau(f)\right]$$

$$\iff p^t(f) \propto p^0(f) \exp\left[\gamma_1 \Delta V_f^* + \sum_{\iota=1}^{t-1}\sum_{h=1}^{H} \Delta L_h^\tau(f)\right], \tag{24}$$

and

$$q^t(f) \propto q^0(f) \exp\left[-\gamma_2 V_f^{\pi^t,*} + \sum_{\iota=1}^{t-1}\sum_{h=1}^{H} L_h^\tau(f)\right]$$

$$\iff q^t(f) \propto q^0(f) \exp\left[-\gamma_2 \Delta V_f^{\pi^t,*} + \sum_{\iota=1}^{t-1}\sum_{h=1}^{H} \Delta L_h^\tau(f)\right]. \tag{25}$$

since here adding or subtracting terms irrelevant to $f$, i.e., $V_{f^*}^{\pi,*}$, $V_{f^*}^*$, and $L_h^t(f^*)$, within the power of all exponential terms will not change the posterior distribution of $f$.

To learning a sublinear upper bound for the expected value of the total regret, i.e., $\mathbb{E}[\mathrm{Reg}^{\mathrm{sp}}(T)] = \mathbb{E}\sum_{t=1}^{T}[V_{f^*}^{*,\nu^t} - V_{f^*}^{\pi^t,*}]$ for the self-play setting, we need to execute both Algorithm 1 and 3, which are two symmetric algorithms. We can decompose $\mathrm{Reg}^{\mathrm{sp}}(T)$ as

$$\mathrm{Reg}^{\mathrm{sp}}(T) = \mathrm{Reg}_1^{\mathrm{sp}}(T) + \mathrm{Reg}_2^{\mathrm{sp}}(T),$$

where we define

$$\mathrm{Reg}_1^{\mathrm{sp}}(T) := \sum_{t=1}^{T}[V_{f^*}^* - V_{f^*}^{\pi^t,*}], \qquad \mathrm{Reg}_2^{\mathrm{sp}}(T) := \sum_{t=1}^{T}[V_{f^*}^{*,\nu^t} - V_{f^*}^*].$$

In fact, executing Algorithm 1 leads to a low regret upper bound for $\mathbb{E}[\mathrm{Reg}_1^{\mathrm{sp}}(T)]$ while running Algorithm 3 incurs a low regret upper bound for $\mathbb{E}[\mathrm{Reg}_2^{\mathrm{sp}}(T)]$. Since the two algorithms are symmetric, the derivation of their respective regret bounds is thus similar. In the following subsections, we present the proofs of Proposition 1, Proposition 2, and Theorem 1 sequentially.

## E.1   Proof of Proposition 1

*Proof.* We start our proof by first decomposing the regret as follows

$$\mathrm{Reg}_1^{\mathrm{sp}}(T) = \underbrace{\sum_{t=1}^{T}[V_{f^*}^* - V_{f^*}^{\pi^t,\nu^t}]}_{\text{Term(i)}} + \underbrace{\sum_{t=1}^{T}[V_{f^*}^{\pi^t,\nu^t} - V_{f^*}^{\pi^t,*}]}_{\text{Term(ii)}}.$$

Our goal is to give the upper bound for the expected value of the total regret, which is $\mathbb{E}[\mathrm{Reg}_1^{\mathrm{sp}}(T)]$. Thus, we need to derive the upper bounds of $\mathbb{E}[\mathrm{Term(i)}]$ and $\mathbb{E}[\mathrm{Term(ii)}]$ separately. Intuitively, according to our analysis below, $\mathbb{E}[\mathrm{Term(i)}]$ can be viewed as the regret incurred by the updating rules for the main player in Line 3 and Line 4 of Algorithm 1, and $\mathbb{E}[\mathrm{Term(ii)}]$ is associated with the exploiter's updating rules in Line 5 and Line 6 of Algorithm 1,

**Bound $\mathbb{E}[\mathbf{Term(i)}]$.** To bound $\mathbb{E}[\mathrm{Term(i)}]$, we give the following decomposition

$$
\mathrm{Term(i)} = \sum_{t=1}^{T}[V_{f^*}^* - V_{\overline{f}^t}^{\pi^t,\overline{\nu}^t} + V_{\overline{f}^t}^{\pi^t,\overline{\nu}^t} - V_{\overline{f}^t}^{\pi^t,\underline{\nu}^t} + V_{\overline{f}^t}^{\pi^t,\underline{\nu}^t} - V_{f^*}^{\pi^t,\underline{\nu}^t}]
$$

$$
\leq \sum_{t=1}^{T}[V_{f^*}^* - V_{\overline{f}^t}^{\pi^t,\overline{\nu}^t} + V_{\overline{f}^t}^{\pi^t,\underline{\nu}^t} - V_{f^*}^{\pi^t,\underline{\nu}^t}]
$$

$$
= -\sum_{t=1}^{T}\Delta V_{\overline{f}^t}^* + \sum_{t=1}^{T}[V_{\overline{f}^t}^{\pi^t,\underline{\nu}^t} - V_{f^*}^{\pi^t,\underline{\nu}^t}],
$$

where according to Algorithm 1, we have $(\pi^t,\overline{\nu}^t) = \mathrm{argmax}_\pi\,\mathrm{argmin}_\nu\,V_{\overline{f}^t}^{\pi,\nu}$, i.e., $(\pi^t,\overline{\nu}^t)$ is the NE of $V_{\overline{f}^t}^{\pi,\nu}$. Thus, the second equality is by (22), and the inequality is due to

$$
V_{\overline{f}^t}^{\pi^t,\overline{\nu}^t} = \min_\nu V_{\overline{f}^t}^{\pi^t,\nu} \leq V_{\overline{f}^t}^{\pi^t,\underline{\nu}^t}.
$$

According to the condition **(1)** in Definition 1 for self-play GEC and the updating rules in Algorithm 1, setting the exploration policy pair as $\sigma^t = (\pi^t,\underline{\nu}^t)$, and $\rho^t = \overline{f}^t$ for Definition 1, we have

$$
\left|\sum_{t=1}^{T}\left(V_{\overline{f}^t,1}^{\pi^t,\underline{\nu}^t} - V_{f^*}^{\pi^t,\underline{\nu}^t}\right)\right|
$$

$$
\leq \left[d_{\mathrm{GEC}}\sum_{h=1}^{H}\sum_{t=1}^{T}\left(\sum_{\iota=1}^{t-1}\mathbb{E}_{(\sigma^\iota,h)}\ell(\overline{f}^t,\xi_h^\iota)\right)\right]^{1/2} + 2H(d_{\mathrm{GEC}}HT)^{\frac{1}{2}} + \epsilon HT
$$

$$
\leq \frac{1}{\gamma_1}\sum_{h=1}^{H}\sum_{t=1}^{T}\left(\sum_{\iota=1}^{t-1}\mathbb{E}_{(\pi^\iota,\underline{\nu}^\iota,h)}\ell(\overline{f}^t,\xi_h^\iota)\right) + \frac{\gamma_1 d_{\mathrm{GEC}}}{4} + 2H(d_{\mathrm{GEC}}HT)^{\frac{1}{2}} + \epsilon HT,
$$

where the second inequality is due to $\sqrt{xy} \leq \frac{1}{\gamma_1}x^2 + \frac{\gamma_1}{4}y^2$. Combining the above results, we have

$$
\mathrm{Term(i)} = \frac{1}{\gamma_1}\sum_{t=1}^{T}\sum_{h=1}^{H}\left(\sum_{\iota=1}^{t-1}\mathbb{E}_{(\pi^\iota,\underline{\nu}^\iota,h)}\ell(\overline{f}^t,\xi_h^\iota)\right) - \sum_{t=1}^{T}\Delta V_{\overline{f}^t}^*
$$

$$
+ \frac{\gamma_1 d_{\mathrm{GEC}}}{4} + 2H(d_{\mathrm{GEC}}HT)^{\frac{1}{2}} + \epsilon HT.
$$

We need to further bound the RHS of the above equality. Note that for FOMGs and POMGs, we have different definitions of $\ell(f,\xi_h)$. Specifically, according to our definition of $\ell(f,\xi_h^\iota)$, for FOMGs, we define

$$
\ell(\overline{f}^t,\xi_h^\iota) = D_{\mathrm{He}}^2(\mathbb{P}_{\overline{f}^t,h}(\cdot|s_h^\iota,a_h^\iota,b_h^\iota),\mathbb{P}_{f^*,h}(\cdot|s_h^\iota,a_h^\iota,b_h^\iota)),
$$

and for POMGs, we have the relation

$$
\mathbb{E}_{(\pi^\iota,\underline{\nu}^\iota,h)}\ell(\overline{f}^t,\xi_h^\iota) = D_{\mathrm{He}}^2(\mathbb{P}_{\overline{f}^t,h}^{\pi^\iota,\underline{\nu}^\iota},\mathbb{P}_{f^*,h}^{\pi^\iota,\underline{\nu}^\iota}).
$$

On the other hand, for the two MG settings, we also define $L_h^t(f)$ as in (1) and (2). According to Lemmas 11 and 12, unifying their results, we can obtain

$$
\mathbb{E}_{Z^{t-1}}\mathbb{E}_{\overline{f}^t\sim p^t}\left[\sum_{h=1}^{H}\sum_{\iota=1}^{t-1}\mathbb{E}_{(\pi^\iota,\underline{\nu}^\iota,h)}\ell(\overline{f}^t,\xi_h^\iota)\right]
$$

$$
\leq \mathbb{E}_{Z^{t-1}}\mathbb{E}_{\overline{f}^t\sim p^t}\left[-\sum_{h=1}^{H}\sum_{\iota=1}^{t-1}[L_h^t(\overline{f}^t) - L_h^t(f^*)] + \log\frac{p^t(\overline{f}^t)}{p^0(\overline{f}^t)}\right],
$$

which further gives

$$\mathbb{E}_{Z^{t-1}}\mathbb{E}_{\overline{f}^t \sim p^t}\left[-\gamma_1 \Delta V_{\overline{f}^t}^* + \sum_{h=1}^H \sum_{\iota=1}^{t-1} \mathbb{E}_{(\pi^\iota,\underline{\nu}^\iota,h)}\ell(\overline{f}^t,\xi_h^\iota)\right]$$

$$\leq \mathbb{E}_{Z^{t-1}}\mathbb{E}_{\overline{f}^t \sim p^t}\left[-\gamma_1 \Delta V_{\overline{f}^t}^* - \sum_{h=1}^H \sum_{\iota=1}^{t-1} \Delta L_h^\iota(\overline{f}^t) + \log \frac{p^t(\overline{f}^t)}{p^0(\overline{f}^t)}\right]$$

$$= \mathbb{E}_{Z^{t-1}}\mathbb{E}_{\overline{f}^t \sim p^t}\left[-\gamma_1 \Delta V_{\overline{f}^t}^* - \sum_{h=1}^H \sum_{\iota=1}^{t-1} \Delta L_h^\iota(\overline{f}^t) - \log p^0(\overline{f}^t) + \log p^t(\overline{f}^t)\right].$$

where $\Delta L_h^t(f) := L_h^t(f) - L_h^t(f^*)$ is defined in (23). Note that in the above analysis, we slightly abuse the notation of $\pi^t$ and $\underline{\nu}^t$, which denote the Markovian policies for FOMG and history-dependent policies for POMGs. Therefore, according to Lemma 10, we obtain that the distribution

$$p^t(f) \propto \exp\left(\gamma_1 \Delta V_f^* + \sum_{h=1}^H \sum_{\iota=1}^{t-1} \Delta L_h^\iota(f) + \log p^0(f)\right)$$

$$= p^0(f)\exp\left(\gamma_1 \Delta V_f^* + \sum_{h=1}^H \sum_{\iota=1}^{t-1} \Delta L_h^\iota(f)\right)$$

minimizes the last term in the above inequality. As discussed in (24), this distribution is an equivalent form of the posterior distribution updating rule for $p^t$ adopted in Line 3 of Algorithm 1. This result implies that

$$\mathbb{E}_{Z^{t-1}}\mathbb{E}_{\overline{f}^t \sim p^t}\left[-\gamma_1 \Delta V_{\overline{f}^t}^* - \sum_{h=1}^H \sum_{\iota=1}^{t-1} \Delta L_h^\iota(\overline{f}^t) + \log \frac{p^t(\overline{f}^t)}{p^0(\overline{f}^t)}\right]$$

$$= \mathbb{E}_{Z^{t-1}}\inf_p \mathbb{E}_{f \sim p}\left[-\gamma_1 \Delta V_f^* - \sum_{h=1}^H \sum_{\iota=1}^{t-1} \Delta L_h^\iota(f) + \log \frac{p(f)}{p^0(f)}\right].$$

By Definition 2, further letting $\widetilde{p} := p^0(f)\cdot\mathbf{1}(f \in \mathcal{F}(\varepsilon))/p^0(\mathcal{F}(\varepsilon))$, we have

$$\mathbb{E}_{Z^{t-1}}\inf_p \mathbb{E}_{f \sim p}\left[-\gamma_1 \Delta V_f^* - \sum_{h=1}^H \sum_{\iota=1}^{t-1} \Delta L_h^\iota(f) + \log \frac{p(f)}{p^0(f)}\right]$$

$$\leq \mathbb{E}_{Z^{t-1}}\mathbb{E}_{f \sim \widetilde{p}}\left[-\gamma_1 \Delta V_f^* - \sum_{h=1}^H \sum_{\iota=1}^{t-1} \Delta L_h^\iota(f) + \log \frac{p(f)}{p^0(f)}\right]$$

$$\leq (3H^2\gamma_1 + \eta HT)\varepsilon - \log p^0(\mathcal{F}(\varepsilon))$$

$$\leq \omega(4HT,p^0),$$

where the second inequality is by Lemma 13 and Lemma 14 as well as $\varepsilon \leq 1$, and the last inequality is by our setting that $\eta = \frac{1}{2}$ and $H\gamma_1 \leq T$. Therefore, by combining the above results, we obtain

$$\mathbb{E}[\text{Term(i)}] \leq \frac{\omega(HT,p^0)T}{\gamma_1} + \frac{\gamma_1 d_{\text{GEC}}H}{4} + 2H(d_{\text{GEC}}HT)^{\frac{1}{2}} + \epsilon HT$$

$$\leq 4\sqrt{d_{\text{GEC}}HT\omega(4HT,p^0)}, \tag{26}$$

where we set $HT > d_{\text{GEC}}$, $\epsilon = 1/\sqrt{HT}$ for self-play GEC in Definition 1, and $\gamma_1 = 2\sqrt{\omega(4HT,p^0)T/d_{\text{GEC}}}$. Since we set $\epsilon = 1/\sqrt{HT}$, the self-play GEC $d_{\text{GEC}}$ is defined associated with $\epsilon = 1/\sqrt{HT}$ by Definition 1. However, our analysis in Appendix B shows that this setting only introduces $\log T$ factors in $d_{\text{GEC}}$ in the worst case.

**Bound $\mathbb{E}[\textbf{Term(ii)}]$.** Next, we consider to bound $\mathbb{E}[\text{Term(ii)}]$. We start by decomposing Term(ii) as follows,

$$\text{Term(ii)} = \sum_{t=1}^{T}[V_{f^*}^{\pi^t,\underline{\nu}^t} - V_{\underline{f}^t}^{\pi^t,\underline{\nu}^t} + V_{\underline{f}^t}^{\pi^t,\underline{\nu}^t} - V_{f^*}^{\pi^t,*}]$$

$$= \sum_{t=1}^{T}[V_{f^*}^{\pi^t,\underline{\nu}^t} - V_{\underline{f}^t}^{\pi^t,\underline{\nu}^t}] + \sum_{t=1}^{T}\Delta V_{\underline{f}^t}^{\pi^t,*},$$

where the second equality is by (21) and the fact that $\underline{\nu}^t = \arg\min_{\nu} V_{\underline{f}^t}^{\pi^t,\nu}$.

In addition, according to Definition 1 and the updating rules in Algorithm 1, setting $\sigma^t = (\pi^t, \underline{\nu}^t)$, $\rho^t = \underline{f}^t$, we have

$$\left|\sum_{t=1}^{T}\left(V_{\underline{f}^t}^{\pi^t,\underline{\nu}^t} - V_{f^*}^{\pi^t,\underline{\nu}^t}\right)\right|$$

$$\leq \left[d_{\text{GEC}}\sum_{h=1}^{H}\sum_{t=1}^{T}\left(\sum_{\iota=1}^{t-1}\mathbb{E}_{(\sigma^\iota,h)}\ell(\underline{f}^t,\xi_h^\iota)\right)\right]^{1/2} + 2H(d_{\text{GEC}}HT)^{\frac{1}{2}} + \epsilon HT$$

$$\leq \frac{1}{\gamma_2}\sum_{h=1}^{H}\sum_{t=1}^{T}\left(\sum_{\iota=1}^{t-1}\mathbb{E}_{(\pi^\iota,\underline{\nu}^\iota,h)}\ell(\underline{f}^t,\xi_h^\iota)\right) + \frac{\gamma_2 d_{\text{GEC}}}{4} + 2H(d_{\text{GEC}}HT)^{\frac{1}{2}} + \epsilon HT,$$

where the second inequality is due to $\sqrt{xy} \leq \frac{1}{\gamma_2}x^2 + \frac{\gamma_2}{4}y^2$. Thus, we obtain that

$$\text{Term(ii)} = \frac{1}{\gamma_2}\sum_{t=1}^{T}\sum_{h=1}^{H}\left(\sum_{\iota=1}^{t-1}\mathbb{E}_{(\pi^\iota,\underline{\nu}^\iota,h)}\ell(\underline{f}^t,\xi_h^\iota)\right)$$

$$+ \sum_{t=1}^{T}\Delta V_{\underline{f}^t}^{\pi^t,*} + \frac{\gamma_2 d_{\text{GEC}}}{4} + 2H(d_{\text{GEC}}HT)^{\frac{1}{2}} + \epsilon HT.$$

We further bound the RHS of the above equality. By the definitions of $\ell(f,\xi_h)$ for FOMGs and POMGs and also the definitions of $L_h^t(f)$ as in (1) and (2), according to Lemmas 11 and 12, we can obtain

$$\mathbb{E}_{Z^{t-1},\overline{f}^t\sim p^t}\mathbb{E}_{\underline{f}^t\sim q^t}\left[\gamma_2\Delta V_{\underline{f}^t}^{\pi^t,*} + \sum_{h=1}^{H}\sum_{\iota=1}^{t-1}\mathbb{E}_{(\pi^\iota,\underline{\nu}^\iota,h)}\ell(\underline{f}^t,\xi_h^\iota)\right]$$

$$\leq \mathbb{E}_{Z^{t-1},\overline{f}^t\sim p^t}\mathbb{E}_{\underline{f}^t\sim q^t}\left[\gamma_2\Delta V_{\underline{f}^t}^{\pi^t,*} - \sum_{h=1}^{H}\sum_{\iota=1}^{t-1}\Delta L_h^\iota(\underline{f}^t) + \log\frac{q^t(\underline{f}^t)}{q^0(\underline{f}^t)}\right]$$

$$= \mathbb{E}_{Z^{t-1},\overline{f}^t\sim p^t}\inf_{q}\mathbb{E}_{f\sim q}\left[\gamma_2\Delta V_f^{\pi^t,*} - \sum_{h=1}^{H}\sum_{\iota=1}^{t-1}\Delta L_h^\iota(f) + \log\frac{q(f)}{q^0(f)}\right],$$

where the expectation for $\overline{f}^t \sim p^t$ exists due to that $\pi^t$ is computed based on $\overline{f}^t$, and also according to Lemma 10 and (25), the last equality can be achieved by that the updating rule for the distribution $q^t$ in Algorithm 1, i.e., equivalently

$$q^t(f) \propto \exp\left(-\gamma_2\Delta V_f^{\pi^t,*} + \sum_{h=1}^{H}\sum_{\iota=1}^{t-1}\Delta L_h^\iota(f) + \log q^0(f)\right)$$

$$= q^0(f)\exp\left(-\gamma_2\Delta V_f^{\pi^t,*} + \sum_{h=1}^{H}\sum_{\iota=1}^{t-1}\Delta L_h^\iota(f)\right),$$

can minimize $\mathbb{E}_{f\sim q}\left[\gamma_2\Delta V_f^{\pi^t,*} - \sum_{h=1}^{H}\sum_{\iota=1}^{t-1}\Delta L_h^\iota(f) + \log\frac{q(f)}{q^0(f)}\right]$.

By Definition 2, further letting $\widetilde{q} := q^0(f) \cdot \mathbf{1}(f \in \mathcal{F}(\varepsilon))/q^0(\mathcal{F}(\varepsilon))$ with $\varepsilon \le 1$, we have

$$\mathbb{E}_{Z^{t-1}, \overline{f}^t \sim p^t} \inf_q \mathbb{E}_{f \sim q} \left[ \gamma_2 \Delta V_f^{\pi^t, *} - \sum_{h=1}^{H} \sum_{\iota=1}^{t-1} \Delta L_h^{\iota}(f) + \log \frac{q(f)}{q^0(f)} \right]$$

$$\le \mathbb{E}_{Z^{t-1}, \overline{f}^t \sim p^t} \mathbb{E}_{f \sim \widetilde{q}} \left[ \gamma_2 \Delta V_f^{\pi^t, *} - \sum_{h=1}^{H} \sum_{\iota=1}^{t-1} \Delta L_h^{\iota}(f) + \log \frac{q(f)}{q^0(f)} \right]$$

$$\le (3H^2 \gamma_2 + \eta HT)\varepsilon - \log q^0(\mathcal{F}(\varepsilon))$$

$$\le \omega(4HT, q^0),$$

where the second inequality is by Lemma 13 and Lemma 14 as well as $\varepsilon \le 1$, and the last inequality is by our setting that $\eta = \frac{1}{2}$ and $H\gamma_2 \le T$. Therefore, combining the above results, we have

$$\mathbb{E}[\text{Term(ii)}] \le \frac{\omega(HT, q^0)T}{\gamma_2} + \frac{\gamma_2 d_{\text{GEC}} H}{4} + 2H(d_{\text{GEC}} HT)^{\frac{1}{2}} + \epsilon HT$$

$$\le 4\sqrt{d_{\text{GEC}} HT \cdot \omega(4HT, q^0)}, \tag{27}$$

where we set $HT > d_{\text{GEC}}$, $\epsilon = 1/\sqrt{HT}$ for self-play GEC in Definition 1, and $\gamma_2 = 2\sqrt{\omega(4HT, q^0)T/d_{\text{GEC}}}$.

**Combining Results.** Finally, combining the results in (26) and (27), we eventually obtain

$$\mathbb{E}[\text{Reg}_1^{\text{sp}}(T)] = \mathbb{E}[\text{Term(i)}] + \mathbb{E}[\text{Term(ii)}] \le 6\sqrt{d_{\text{GEC}} HT(\omega(4HT, p^0) + \omega(4HT, q^0))}.$$

This completes the proof. $\qquad\qquad\square$

### E.2 Proof of Proposition 2

*Proof.* Due to the symmetry of Algorithms 1 and 3, we can similarly bound $\mathbb{E}[\text{Reg}_2^{\text{sp}}(T)]$ in the way of bounding $\mathbb{E}[\text{Reg}_2^{\text{sp}}(T)]$. Therefore, in this subsection, we only present the main steps for the proof. Specifically, by Algorithm 3, we have a decomposition as

$$\text{Reg}_2^{\text{sp}}(T) = \underbrace{\sum_{t=1}^{T} [V_{f^*}^{*, \nu^t} - V_{f^*}^{\pi^t, \nu^t}]}_{\text{Term(iv)}} + \underbrace{\sum_{t=1}^{T} [V_{f^*}^{\pi^t, \nu^t} - V_{f^*}^*]}_{\text{Term(iii)}}.$$

**Bound $\mathbb{E}[\text{Term(iii)}]$.** To bound $\mathbb{E}[\text{Term(iii)}]$, we give the following decomposition

$$\text{Term(iii)} = \sum_{t=1}^{T} [V_{f^*}^{\underline{\pi}^t, \nu^t} - V_{\overline{f}^t}^{\underline{\pi}^t, \nu^t} + V_{\overline{f}^t}^{\underline{\pi}^t, \nu^t} - V_{\overline{f}^t}^{\overline{\pi}^t, \nu^t} + V_{\overline{f}^t}^{\overline{\pi}^t, \nu^t} - V_{f^*}^*]$$

$$\le \sum_{t=1}^{T} [V_{f^*}^{\underline{\pi}^t, \nu^t} - V_{\overline{f}^t}^{\underline{\pi}^t, \nu^t} + V_{\overline{f}^t}^{\underline{\pi}^t, \nu^t} - V_{f^*}^*]$$

$$= \sum_{t=1}^{T} \Delta V_{\overline{f}^t}^* + \sum_{t=1}^{T} [V_{f^*}^{\underline{\pi}^t, \nu^t} - V_{\overline{f}^t}^{\underline{\pi}^t, \nu^t}],$$

where according to Algorithm 3, we have $(\overline{\pi}^t, \nu^t) = \text{argmax}_\pi \text{argmin}_\nu V_{\overline{f}^t}^{\pi, \nu}$, which thus leads to

$$V_{\overline{f}^t}^{\overline{\pi}^t, \nu^t} = \max_\pi V_{\overline{f}^t}^{\pi, \nu^t} \ge V_{\overline{f}^t}^{\underline{\pi}^t, \nu^t}.$$

According to the condition **(2)** in Definition 1 for self-play GEC and the updating rules in Algorithm 3, setting the exploration policy pair as $\sigma^t = (\underline{\pi}^t, \nu^t)$, we have

$$\left| \sum_{t=1}^{T} \left( V_{\overline{f}^t, 1}^{\underline{\pi}^t, \nu^t} - V_{f^*}^{\underline{\pi}^t, \nu^t} \right) \right|$$

$$\le \frac{1}{\gamma_1} \sum_{h=1}^{H} \sum_{t=1}^{T} \left( \sum_{\iota=1}^{t-1} \mathbb{E}_{(\underline{\pi}^\iota, \nu^\iota, h)} \ell(\overline{f}^t, \xi_h^\iota) \right) + \frac{\gamma_1 d_{\text{GEC}}}{4} + 2H(d_{\text{GEC}} HT)^{\frac{1}{2}} + \epsilon HT.$$

We need to further bound the RHS of the above equality. For FOMGs and POMGs, by their definitions of $\ell(f, \xi_h)$ as well as $L_h^t(f)$, according to Lemmas 11 and 12, we obtain

$$\mathbb{E}_{Z^{t-1}} \mathbb{E}_{\overline{f}^t \sim p^t} \left[ \gamma_1 \Delta V_{\overline{f}^t}^* + \sum_{h=1}^{H} \sum_{\iota=1}^{t-1} \mathbb{E}_{(\pi^\iota, \underline{\nu}^\iota, h)} \ell(\overline{f}^t, \xi_h^\iota) \right]$$

$$\leq \mathbb{E}_{Z^{t-1}} \mathbb{E}_{\overline{f}^t \sim p^t} \left[ \gamma_1 \Delta V_{\overline{f}^t}^* - \sum_{h=1}^{H} \sum_{\iota=1}^{t-1} \Delta L_h^\iota(\overline{f}^t) + \log \frac{p^t(\overline{f}^t)}{p^0(\overline{f}^t)} \right],$$

where $\Delta L_h^t(f) := L_h^t(f) - L_h^t(f^*)$ is defined in (23). According to Lemma 10, we obtain that the distribution $p^t$ defined in Algorithm 3, equivalently

$$p^t(f) \propto p^0(f) \exp \left( -\gamma_1 \Delta V_f^* + \sum_{h=1}^{H} \sum_{\iota=1}^{t-1} \Delta L_h^\iota(f) \right)$$

minimizes the last term in the above inequality. This result implies that

$$\mathbb{E}_{Z^{t-1}} \mathbb{E}_{\overline{f}^t \sim p^t} \left[ \gamma_1 \Delta V_{\overline{f}^t}^* - \sum_{h=1}^{H} \sum_{\iota=1}^{t-1} \Delta L_h^\iota(\overline{f}^t) + \log \frac{p^t(\overline{f}^t)}{p^0(\overline{f}^t)} \right]$$

$$= \mathbb{E}_{Z^{t-1}} \inf_p \mathbb{E}_{f \sim p} \left[ \gamma_1 \Delta V_f^* - \sum_{h=1}^{H} \sum_{\iota=1}^{t-1} \Delta L_h^\iota(f) + \log \frac{p(f)}{p^0(f)} \right].$$

By Definition 2, further letting $\widetilde{p} := p^0(f) \cdot \mathbf{1}(f \in \mathcal{F}(\varepsilon)) / p^0(\mathcal{F}(\varepsilon))$, we have

$$\mathbb{E}_{Z^{t-1}} \inf_p \mathbb{E}_{f \sim p} \left[ \gamma_1 \Delta V_f^* - \sum_{h=1}^{H} \sum_{\iota=1}^{t-1} \Delta L_h^\iota(f) + \log \frac{p(f)}{p^0(f)} \right]$$

$$\leq \mathbb{E}_{Z^{t-1}} \mathbb{E}_{f \sim \widetilde{p}} \left[ \gamma_1 \Delta V_f^* - \sum_{h=1}^{H} \sum_{\iota=1}^{t-1} \Delta L_h^\iota(f) + \log \frac{p(f)}{p^0(f)} \right]$$

$$\leq (3H^2 \gamma_1 + \eta HT) \varepsilon - \log p^0(\mathcal{F}(\varepsilon))$$

$$\leq \omega(4HT, p^0),$$

where the second inequality is by Lemma 13 and Lemma 14 as well as $\varepsilon \leq 1$, and the last inequality is by our setting that $\eta = \frac{1}{2}$ and $H\gamma_1 \leq T$. Therefore, by combining the above results, we obtain

$$\mathbb{E}[\text{Term(iii)}] \leq \frac{\omega(HT, p^0)T}{\gamma_1} + \frac{\gamma_1 d_{\text{GEC}} H}{4} + 2H(d_{\text{GEC}} HT)^{\frac{1}{2}} + \epsilon HT$$

$$\leq 4\sqrt{d_{\text{GEC}} HT \omega(4HT, p^0)}, \tag{28}$$

where we set $HT > d_{\text{GEC}}$, $\epsilon = 1/\sqrt{HT}$ for self-play GEC in Definition 1, and $\gamma_1 = 2\sqrt{\omega(4HT, p^0)T/d_{\text{GEC}}}$.

**Bound $\mathbb{E}[\text{Term(iv)}]$.** Next, we consider to bound $\mathbb{E}[\text{Term(iv)}]$ as follows,

$$\text{Term(iv)} = \sum_{t=1}^{T} [V_{f^*}^{*, \nu^t} - V_{\underline{f}^t}^{\underline{\pi}^t, \nu^t} + V_{\underline{f}^t}^{\underline{\pi}^t, \nu^t} - V_{f^*}^{\underline{\pi}^t, \nu^t}]$$

$$= -\sum_{t=1}^{T} \Delta V_{\underline{f}^t}^{*, \nu^t} + \sum_{t=1}^{T} [V_{\underline{f}^t}^{\underline{\pi}^t, \nu^t} - V_{f^*}^{\underline{\pi}^t, \nu^t}],$$

where the second equality is by the fact that $\underline{\pi}^t = \text{argmax}_\pi V_{\underline{f}^t}^{\pi, \nu^t}$ according to the updating rule.

By Definition 1 and the updating rules in Algorithm 3, we have

$$\left| \sum_{t=1}^{T} \left( V_{\underline{f}^t}^{\underline{\pi}^t, \nu^t} - V_{f^*}^{\underline{\pi}^t, \nu^t} \right) \right|$$

$$\leq \frac{1}{\gamma_2} \sum_{h=1}^{H} \sum_{t=1}^{T} \left( \sum_{\iota=1}^{t-1} \mathbb{E}_{(\underline{\pi}^\iota, \nu^\iota, h)} \ell(\underline{f}^t, \xi_h^\iota) \right) + \frac{\gamma_2 d_{\text{GEC}}}{4} + 2H(d_{\text{GEC}} HT)^{\frac{1}{2}} + \epsilon HT.$$

By the definitions of $\ell(f, \xi_h)$ for FOMGs and POMGs and also the definitions of $L_h^t(f)$ as in (1) and (2), we have

$$\mathbb{E}_{Z^{t-1}, \overline{f}^t \sim p^t} \mathbb{E}_{\underline{f}^t \sim q^t} \left[ -\gamma_2 \Delta V_{\underline{f}^t}^{*, \nu^t} + \sum_{h=1}^{H} \sum_{\iota=1}^{t-1} \mathbb{E}_{(\underline{\pi}^\iota, \nu^\iota, h)} \ell(\underline{f}^t, \xi_h^\iota) \right]$$

$$\leq \mathbb{E}_{Z^{t-1}, \overline{f}^t \sim p^t} \mathbb{E}_{\underline{f}^t \sim q^t} \left[ -\gamma_2 \Delta V_{\underline{f}^t}^{*, \nu^t} - \sum_{h=1}^{H} \sum_{\iota=1}^{t-1} \Delta L_h^\iota(\underline{f}^t) + \log \frac{q^t(\underline{f}^t)}{q^0(\underline{f}^t)} \right]$$

$$= \mathbb{E}_{Z^{t-1}, \overline{f}^t \sim p^t} \inf_q \mathbb{E}_{f \sim q} \left[ -\gamma_2 \Delta V_f^{\pi^t, *} - \sum_{h=1}^{H} \sum_{\iota=1}^{t-1} \Delta L_h^\iota(f) + \log \frac{q(f)}{q^0(f)} \right],$$

where the expectation for $\overline{f}^t \sim p^t$ exists due to that $\nu^t$ is computed based on $\overline{f}^t$, and also according to Lemma 10, the last equality can be achieved by that the updating rule for the distribution $q^t$ in Algorithm 3, i.e., equivalently

$$q^t(f) \propto q^0(f) \exp \left( \gamma_2 \Delta V_f^{\pi^t, *} + \sum_{h=1}^{H} \sum_{\iota=1}^{t-1} \Delta L_h^\iota(f) \right),$$

can minimize $\mathbb{E}_{f \sim q} \left[ -\gamma_2 \Delta V_f^{* \pi^t, \nu^t} - \sum_{h=1}^{H} \sum_{\iota=1}^{t-1} \Delta L_h^\iota(f) + \log \frac{q(f)}{q^0(f)} \right]$. By Definition 2, further letting $\widetilde{q} := q^0(f) \cdot \mathbf{1}(f \in \mathcal{F}(\varepsilon)) / q^0(\mathcal{F}(\varepsilon))$ with $\varepsilon \leq 1$, we have

$$\mathbb{E}_{Z^{t-1}, \overline{f}^t \sim p^t} \inf_q \mathbb{E}_{f \sim q} \left[ \gamma_2 \Delta V_f^{\pi^t, *} - \sum_{h=1}^{H} \sum_{\iota=1}^{t-1} \Delta L_h^\iota(f) + \log \frac{q(f)}{q^0(f)} \right]$$

$$\leq \mathbb{E}_{Z^{t-1}, \overline{f}^t \sim p^t} \mathbb{E}_{f \sim \widetilde{q}} \left[ \gamma_2 \Delta V_f^{\pi^t, *} - \sum_{h=1}^{H} \sum_{\iota=1}^{t-1} \Delta L_h^\iota(f) + \log \frac{q(f)}{q^0(f)} \right]$$

$$\leq (3H^2 \gamma_2 + \eta HT) \varepsilon - \log q^0(\mathcal{F}(\varepsilon))$$

$$\leq \omega(4HT, q^0),$$

where the second inequality is by Lemma 13 and Lemma 14 as well as $\varepsilon \leq 1$, and the last inequality is by our setting that $\eta = \frac{1}{2}$ and $H\gamma_2 \leq T$. Therefore, combining the above results, we have

$$\mathbb{E}[\text{Term(iv)}] \leq \frac{\omega(HT, q^0) T}{\gamma_2} + \frac{\gamma_2 d_{\text{GEC}} H}{4} + 2H(d_{\text{GEC}} HT)^{\frac{1}{2}} + \epsilon HT$$

$$\leq 4\sqrt{d_{\text{GEC}} HT \cdot \omega(4HT, q^0)}, \tag{29}$$

where we set $HT > d_{\text{GEC}}$, $\epsilon = 1/\sqrt{HT}$ for self-play GEC in Definition 1, and $\gamma_2 = 2\sqrt{\omega(4HT, q^0) T / d_{\text{GEC}}}$.

**Combining Results.** Finally, combining the results in (28) and (29), we eventually obtain

$$\mathbb{E}[\text{Reg}_2^{\text{sp}}(T)] = \mathbb{E}[\text{Term(iii)}] + \mathbb{E}[\text{Term(iv)}] \leq 6\sqrt{d_{\text{GEC}} HT (\omega(4HT, p^0) + \omega(4HT, q^0))}.$$

This completes the proof. $\qquad \square$

### E.3 Proof of Theorem 1

*Proof.* The proof of Theorem 1 is immediate by the relation that $\text{Reg}^{\text{sp}}(T) = \text{Reg}_1^{\text{sp}}(T) + \text{Reg}_2^{\text{sp}}(T)$. Thus, according to the above proofs of Proposition 1 and Proposition 2, under the conditions in both propositions, we have that

$$\mathbb{E}[\text{Reg}^{\text{sp}}(T)] = \mathbb{E}[\text{Reg}_1^{\text{sp}}(T)] + \mathbb{E}[\text{Reg}_2^{\text{sp}}(T)]$$

$$\leq 12\sqrt{d_{\text{GEC}} HT \cdot (\omega(4HT, p^0) + \omega(4HT, q^0))}.$$

This completes the proof. $\qquad \square$

# F Proofs for Algorithm 2

In this section, we provide the detailed proof for Theorem 2 and unify the proofs for both the FOMG and POMG settings together.

We recall that the value difference and the likelihood function difference are already defined in (22) and (23), which are

$$
\Delta V_f^* := V_f^* - V_{f^*}^*,
$$
$$
\Delta L_h^t(f) = L_h^t(f) - L_h^t(f^*).
$$

Then, similar to (24), the updating rules of posterior distribution in Algorithm 2 have the following equivalent form as

$$
p^t(f) \propto p^0(f) \exp\left[\gamma V_f^* + \sum_{\iota=1}^{t-1}\sum_{h=1}^{H} L_h^\tau(f)\right]
$$
$$
\iff p^t(f) \propto p^0(f) \exp\left[\gamma \Delta V_f^* + \sum_{\iota=1}^{t-1}\sum_{h=1}^{H} \Delta L_h^\tau(f)\right], \tag{30}
$$

since adding or subtracting terms irrelevant to $f$, i.e., $V_{f^*}^*$ and $L_h^t(f^*)$, within the power of all exponential terms will not change the posterior distribution of $f$.

## F.1 Proof of Theorem 2

*Proof.* To bound the expected value of the adversarial regret $\mathrm{Reg}^{\mathrm{adv}}(T)$, i.e., $\mathbb{E}[\mathrm{Reg}^{\mathrm{adv}}(T)]$, we first decompose the regret $\mathrm{Reg}_w^{\mathrm{adv}}$ as follows

$$
\mathrm{Reg}^{\mathrm{adv}}(T) = \sum_{t=1}^{T}[V_{f^*}^* - V_{f^t}^*] + \sum_{t=1}^{T}[V_{f^*}^* - V_{f^*}^{\pi^t,\nu^t}]
$$
$$
= -\sum_{t=1}^{T}\Delta V_{f^t}^* + \sum_{t=1}^{T}[V_{f^t}^* - V_{f^*}^{\pi^t,\nu^t}].
$$

Furthermore, for the term $\sum_{t=1}^{T}[V_{f^t}^* - V_{f^*}^{\pi^t,\nu^t}]$, we have

$$
\sum_{t=1}^{T}[V_{f^t}^* - V_{f^*}^{\pi^t,\nu^t}] = \sum_{t=1}^{T}[V_{f^t}^{\pi^t,\overline{\nu}^t} - V_{f^*}^{\pi^t,\nu^t}]
$$
$$
\leq \sum_{t=1}^{T}[V_{f^t}^{\pi^t,\nu^t} - V_{f^*}^{\pi^t,\nu^t}],
$$

where the above result is by $(\pi^t, \overline{\nu}^t) = \mathrm{argmax}_\pi \mathrm{argmin}_\nu V_{f^t}^{\pi,\nu}$, i.e., $(\pi^t, \overline{\nu}^t)$ is the NE of $V_{f^t}^{\pi,\nu}$, according to Algorithm 2, and also due the relation that

$$
V_{f^t}^{\pi^t,\overline{\nu}^t} = \min_\nu V_{f^t}^{\pi^t,\nu} \leq V_{f^t}^{\pi^t,\nu^t}.
$$

Thus, we have

$$
\mathrm{Reg}^{\mathrm{adv}}(T) = -\sum_{t=1}^{T}\Delta V_{f^t}^* + \sum_{t=1}^{T}[V_{f^t}^{\pi^t,\nu^t} - V_{f^*}^{\pi^t,\nu^t}].
$$

According to Definition 3 and the updating rules in Algorithm 2, setting $\sigma_{\exp}^t = (\pi^t, \nu^t)$, and $\ell(f^t, \xi_h^\iota) = D_{\mathrm{He}}^2(\mathbb{P}_{f^t, h}(\cdot \mid s_h^\iota, a_h^\iota, b_h^\iota), \mathbb{P}_{f^*, h}(\cdot \mid s_h^\iota, a_h^\iota, b_h^\iota))$ for Definition 1, we have

$$
\left| \sum_{t=1}^{T} \left( V_{f^t}^{\pi^t, \nu^t} - V_{f^*}^{\pi^t, \nu^t} \right) \right|
$$

$$
\leq \left[ d_{\mathrm{GEC}} \sum_{h=1}^{H} \sum_{t=1}^{T} \left( \sum_{\iota=1}^{t-1} \mathbb{E}_{(\sigma_{\exp}^\iota, h)} \ell(f^t, \xi_h^\iota) \right) \right]^{\frac{1}{2}} + 2H(d_{\mathrm{GEC}} HT)^{\frac{1}{2}} + \epsilon HT
$$

$$
\leq \frac{1}{\gamma} \sum_{h=1}^{H} \sum_{t=1}^{T} \left( \sum_{\iota=1}^{t-1} \mathbb{E}_{(\pi^\iota, \nu^\iota, h)} \ell(f^t, \xi_h^\iota) \right) + \frac{\gamma d_{\mathrm{GEC}}}{4} + 2H(d_{\mathrm{GEC}} HT)^{\frac{1}{2}} + \epsilon HT,
$$

where the second inequality is due to $\sqrt{xy} \leq \frac{1}{\gamma} x^2 + \frac{\gamma}{4} y^2$. Combining the above results, we have

$$
\mathrm{Reg}^{\mathrm{adv}}(T) \leq \frac{1}{\gamma} \sum_{t=1}^{T} \sum_{h=1}^{H} \left( \sum_{\iota=1}^{t-1} \mathbb{E}_{(\pi^\iota, \nu^\iota)} \ell(f^t, \xi_h^\iota) \right) - \sum_{t=1}^{T} \Delta V_{f^t}^*
$$

$$
+ \frac{\gamma d_{\mathrm{GEC}}}{4} + 2H(d_{\mathrm{GEC}} HT)^{\frac{1}{2}} + \epsilon HT.
$$

For FOMGs and POMGs, we have different definitions of $\ell(f, \xi_h)$. According to (3), for FOMGs, we define

$$
\ell(f^t, \xi_h^\iota) = D_{\mathrm{He}}^2(\mathbb{P}_{f^t, h}(\cdot \mid s_h^\iota, a_h^\iota, b_h^\iota), \mathbb{P}_{f^*, h}(\cdot \mid s_h^\iota, a_h^\iota, b_h^\iota)),
$$

and for POMGs, the definition of $\ell$ ensures

$$
\mathbb{E}_{(\pi^\iota, \nu^\iota, h)} \ell(f^t, \xi_h^\iota) = D_{\mathrm{He}}^2(\mathbf{P}_{f^*, h}^{\pi^\iota, \nu^\iota}, \mathbf{P}_{f^t, h}^{\pi^\iota, \nu^\iota}).
$$

For different MG settings, we also define $L_h^t(f)$ as in (1) and (2) which are

$$
L_h^t(f) = \eta \log \mathbb{P}_{f, h}(s_{h+1}^t \mid s_h^t, a_h^t, b_h^t), \qquad L_h^t(f) = \eta \log \mathbf{P}_{f, h}(\tau_h^t).
$$

Then, combining Lemmas 11 and 12, we know that the following result holds for both FOMGs and POMGs,

$$
\mathbb{E}_{Z^{t-1}} \mathbb{E}_{f^t \sim p^t} \left[ \sum_{h=1}^{H} \sum_{\iota=1}^{t-1} \mathbb{E}_{(\pi^\iota, \nu^\iota, h)} \ell(f^t, \xi_h^\iota) \right]
$$

$$
\leq \mathbb{E}_{Z^{t-1}} \mathbb{E}_{f^t \sim p^t} \left[ -\sum_{h=1}^{H} \sum_{\iota=1}^{t-1} [L_h^\iota(f^t) - L_h^\iota(f^*)] + \log \frac{p^t(f^t)}{p^0(f^t)} \right],
$$

which further leads to

$$
\mathbb{E}_{Z^{t-1}} \mathbb{E}_{f^t \sim p^t} \left[ -\gamma \Delta V_{f^t}^* + \sum_{h=1}^{H} \sum_{\iota=1}^{t-1} \mathbb{E}_{(\pi^\iota, \nu^\iota)} \ell(f^t, \xi_h^\iota) \right]
$$

$$
\leq \mathbb{E}_{Z^{t-1}} \mathbb{E}_{f^t \sim p^t} \left[ -\gamma \Delta V_{f^t}^* - \sum_{h=1}^{H} \sum_{\iota=1}^{t-1} \Delta L_h^\iota(f^t) + \log \frac{p^t(f^t)}{p^0(f^t)} \right]
$$

$$
= \mathbb{E}_{Z^{t-1}} \mathbb{E}_{f^t \sim p^t} \left[ -\gamma \Delta V_{f^t}^* - \sum_{h=1}^{H} \sum_{\iota=1}^{t-1} \Delta L_h^\iota(f^t) - \log p^0(f^t) + \log p^t(f^t) \right],
$$

where $\Delta L_h^\iota(f) = L_h^\iota(f) - L_h^\iota(f^*)$ as in (23). Therefore, according to Lemma 10 and (30), we know that the distribution

$$
p^t(f) \propto \exp \left( \gamma \Delta V_f^* + \sum_{h=1}^{H} \sum_{\iota=1}^{t-1} \Delta L_h^\iota(f) + \log p^0(f) \right)
$$

$$
= p^0(f) \exp \left( \gamma \Delta V_f^* + \sum_{h=1}^{H} \sum_{\iota=1}^{t-1} \Delta L_h^\iota(f) \right)
$$

minimizes the last term in the above inequality, which is the posterior distribution updating rule for $p^t$ adopted in Algorithm 2. This result implies that

$$\mathbb{E}_{Z^{t-1}}\mathbb{E}_{f^t\sim p^t}\left[-\gamma\Delta V_{f^t}^* - \sum_{h=1}^{H}\sum_{\iota=1}^{t-1}\Delta L_h^\iota(f^t) + \log\frac{p^t(f^t)}{p^0(f^t)}\right]$$

$$= \mathbb{E}_{Z^{t-1}}\inf_p\mathbb{E}_{f\sim p}\left[-\gamma\Delta V_f^* - \sum_{h=1}^{H}\sum_{\iota=1}^{t-1}\Delta L_h^\iota(f^t) + \log\frac{p(f)}{p^0(f)}\right].$$

For the adversarial setting, we define $\mathcal{F}(\varepsilon)$ as in Definition 2 similar to the definition of $\mathcal{F}(\varepsilon)$ in the self-play setting. Further letting $\widetilde{p} := p^0(f)\cdot\mathbf{1}(f\in\mathcal{F}(\varepsilon))/p^0(\mathcal{F}(\varepsilon))$ with $\varepsilon\leq 1$ and $\omega$ be associated with $\mathcal{F}(\varepsilon)$, we have

$$\mathbb{E}_{Z^{t-1}}\inf_p\mathbb{E}_{f\sim p}\left[-\gamma\Delta V_f^* - \sum_{h=1}^{H}\sum_{\iota=1}^{t-1}\Delta L_h^\iota(f) + \log\frac{p(f)}{p^0(f)}\right]$$

$$\leq \mathbb{E}_{Z^{t-1}}\mathbb{E}_{f\sim\widetilde{p}}\left[-\gamma\Delta V_f^* - \sum_{h=1}^{H}\sum_{\iota=1}^{t-1}\Delta L_h^\iota(f) + \log\frac{p(f)}{p^0(f)}\right]$$

$$\leq (3H^2\gamma + \eta HT)\varepsilon - \log p^0(\mathcal{F}(\varepsilon))$$

$$\leq \omega(4HT, p^0),$$

where the second inequality is by Lemma 13 and Lemma 14 as well as $\varepsilon\leq 1$, and the last inequality is by our setting that $\eta = \frac{1}{2}$ and $\gamma\leq T$. Therefore, by combining the above results, we obtain

$$\mathbb{E}[\text{Reg}^{\text{adv}}(T)] \leq \frac{\omega(4HT, p^0)T}{\gamma} + \frac{\gamma d_{\text{GEC}}H}{4} + 2H(d_{\text{GEC}}HT)^{\frac{1}{2}} + \epsilon HT$$

$$\leq 4\sqrt{d_{\text{GEC}}HT\cdot\omega(4HT, p^0)},$$

where we set $HT > d_{\text{GEC}}$, $\epsilon = 1/\sqrt{HT}$ for adversarial GEC in Definition 3, and $\gamma = 2\sqrt{\omega(4HT, p^0)T/d_{\text{GEC}}}$. Note that since we set $\epsilon = 1/\sqrt{HT}$, the adversarial GEC $d_{\text{GEC}}$ is now calculated associated with $\epsilon = 1/\sqrt{HT}$ by Definition 3. Further by our analysis in Appendix B, we know that this setting only introduces $\log T$ factors in $d_{\text{GEC}}$ in the worst case. This concludes the proof. $\square$

# G  Other Supporting Lemmas

**Lemma 15** (Elliptical Potential Lemma in [1], Lemma 11). *Suppose $\{\phi_t\}_{t\geq 0}$ is a sequence in $\mathbb{R}^d$ satisfying $\|\phi_t\|_2\leq 1$, and $\Lambda_0$ is a positive definite $d\times d$ matrix. Let $\Lambda_t = \Lambda_0 + \sum_{\iota=1}^{t}\phi_\iota\phi_\iota^\top$. Then, the following inequalities hold*

$$\log\left(\frac{\det\Lambda_t}{\det\Lambda_0}\right) \leq \sum_{\iota=1}^{t}\min\left\{\phi_\iota^\top(\Lambda_{\iota-1})^{-1}\phi_\iota, 1\right\} \leq 2\log\left(\frac{\det\Lambda_t}{\det\Lambda_0}\right).$$

**Lemma 16** (Estimation of Elliptical Potential [38]). *Given $\lambda > 0$ and $\{\phi_t\}_{t\geq 0}$ with $\|\phi_t\|_2\leq 1$, denoting $\Lambda_t = \lambda\mathrm{I} + \sum_{\iota=1}^{t}\phi_\iota\phi_\iota^\top$, then $\phi_\iota^\top(\Lambda_{\iota-1})^{-1}\phi_\iota$ is upper bounded by*

$$\frac{3d}{\log 2}\log\left(1 + \frac{1}{\lambda\log 2}\right).$$

**Lemma 17** ($\ell_2$ Eluder Technique [13, 83]). *Suppose that $\{w_{t,j}\in\mathbb{R}^d\}_{(t,j)\in[T]\times[J]}$, $\{x_{t,i}\in\mathbb{R}^d_{(t,i)\in[T]\times[I]}\}_{(t,i)}$, and distributions $\{p_t\in\Delta_{[I]}\}_{[T]}$ satisfy*

- $\sum_{s=1}^{t-1}\mathbb{E}_{i\sim p_s}(\sum_{j=1}^{J}|w_{t,j}^\top x_{s,i}|)^2\leq\gamma_t,$

- $\mathbb{E}_{i\sim p_t}\|x_{t,i}\|_2^2\leq R_x^2,$

- $\sum_{j=1}^{J} \|w_{t,j}\|_2 \leq R_w,$

*for any $1 \leq t \leq T$. Then, for $R > 0$, we have*

$$\sum_{t=1}^{T} \min\{R, \mathbb{E}_{i \sim p_t}\Big(\sum_{j=1}^{J} |w_{t,j}^{\top} x_{t,i}|\Big)\} \leq \left[2d\Big(R^2 T + \sum_{t=1}^{T} \gamma_t\Big) \cdot \log\Big(1 + \frac{T R_x^2 R_w^2}{R^2}\Big)\right]^{1/2}.$$

