# OpenReview forum: "Posterior Sampling for Competitive RL: Function Approximation and Partial Observation"
_NeurIPS.cc/2023/Conference — NeurIPS 2023 poster_

### Official Review · Reviewer_qywY · 2023-07-02

**Soundness:** 3 good
**Presentation:** 3 good
**Contribution:** 2 fair
**Rating:** 5
**Confidence:** 2

**Summary:**

The paper considers a zero-sum Markov game with unknown dynamics in the case of full and partial observations. The authors propose algorithms for finding a Nash equilibrium in the games, in which, at each iteration, virtual games with dynamics sampled from certain distributions are solved. The paper's main result is theoretical and consists of estimates for the rate of the algorithms' convergence.

**Strengths:**

The paper is aimed at solving an important problem. It is well structured and written in clear mathematical language.

**Weaknesses:**

Unfortunately, as a non-specialist, it is rather difficult for me to assess the significance of the obtained theoretical results. It is not entirely clear what useful conclusion the reader can draw from them concerning practical methods for solving zero-sum Markov games. The proposed algorithms seem very abstract since it is difficult to calculate the indicated distributions in experimental tasks. If I'm wrong and this is feasible, an example confirming this would significantly strengthen the paper.

One more thing I doubt is the fact that in the algorithms, the authors apparently assume that a Nash equilibrium in Markov games exists. This is true if we consider games with an infinite horizon, but I'm not sure if this is true for games with a fixed number of steps $H$. Usually, in such games, it is assumed that the policy depends not only on the state but also on the step number. Otherwise, the existence of a Nash equilibrium is not obvious to me, and I would welcome a reference to this fact.

The paper also contains typos:

178 - $V^*$ instead of $V_1^*$.
189 - $P^{\pi,\nu}_f$ instead of $P^{\pi,\nu}_h$
246 - “begin” instead of “bening”

**Questions:**

On line 151, the authors introduce the reward function $r_h(o,a,b)$. It seems a bit exotic that it depends on an observation $o$ rather than on a state $s$. Is it important for the obtained results?


**Limitations:**

The paper does not have potentially negative societal impact.

---

> ### Author Rebuttal · Authors · 2023-08-09
>
> We thank Reviewer qywY for the valuable advice and questions. We will address your concerns below.
>
> **1. (1) The significance of the theoretical results? (2) Feasibility of the proposed algorithm.**
>
> **(1)** Our work studies this problem in alignment with the recent progress on developing reinforcement learning algorithms with **provable statistical efficiency**. In particular, we majorly focus on **a)** how to design posterior sampling algorithms, an important research direction in RL, with statistically low sample complexity, **b)** how to incorporate more general function classes that cover a wide range of function approximators. Both questions have attracted a lot of attention in RL theory works (e.g., [1,2,3]. Kindly find more details in Related Work). Concentrating on the competitive RL with full and even the harder partial observability, we successfully identify such function classes named Self-Play and Adversarial GEC and design the first posterior sampling algorithms based on them, which fills a gap in competitive RL theory. More importantly, our algorithm enjoys provable sample efficiency guarantees. The dependence on $O(\sqrt{T})$ in our theorems indicates that our regrets are near-optimal.
>
> **(2)** Our algorithm remains feasible when instantiated for a concrete function class. To exemplify, consider a linear-mixture Markov game (MG) setting, which is subsumed by our framework. The transition model can be represented linearly as $\theta_h^\top \phi(s,a,b,s')$ where $\theta_h$ is the model parameter assumed lying in a finite set. Then, in the posterior sampling step, e.g., Line 3 of Algorithm 1, the function class $\mathcal{F}$ is finite, and we can calculate $p^t(f)$ for each $f\in \mathcal{F}$. The optimism term $V^*_f$ in the distribution can be estimated by many existing iterative methods given any model $f$. This optimism term is unavoidable even in a single-agent setting due to the feel-good Thompson sampling framework [2,3]. In addition, [2] mentioned that posterior sampling is relatively amenable to tractable implementation via ensemble approximations or stochastic gradient Langevin dynamics. In fact, in contrast to sample efficiency, how to design provably computation-efficient RL methods with general function approximation [1] and under partial observability [4,8] remains not well explored in existing works so far. Further designing such algorithms is orthogonal to our contributions.
>
> We deeply appreciate the reviewer making efforts to review our work and raise insightful questions. We humbly hope our work can also be assessed on the basis of its statistical results and theoretical contributions.
>
> **2. Existence of Nash equilibrium (NE)?**
>
> Many existing works, e.g., [5,6,7], show that NE exists for the finite-horizon fully observable MGs. These works show the existence using the finite-horizon property, the notion of best response, the Bellman equation, and the backward induction from the last step $H$ to the initial step $1$. For example, to find such a NE, for any state $s$ at step $H$, there always exists a NE $\big(\pi_H^*(\cdot|s), \nu_H^*(\cdot|s)\big)$ for a matrix game with the payoff matrix $[r\_H(s,a,b)]\_{a\in\mathcal{A},b\in\mathcal{B}}$.
>
> Then, for step $H-1$, given any state $s$, by the Bellman equation, we have a new payoff matrix as $[r\_{H-1}(s,a,b)+\mathbb{E}\_{s'\sim \mathbb{P}\_H(s'|s,a,b),a'\sim \pi\_H^*(\cdot|s'),b'\sim\nu\_H^*(\cdot|s')}(Q_H(s',a',b'))]\_{a\in\mathcal{A},b\in\mathcal{B}}$ where $Q_H(s,a,b) = r_H(s,a,b)$.
>
> Thus, we compute a NE for the above new payoff matrix as $\big(\pi_{H-1}^*(\cdot|s), \nu_{H-1}^*(\cdot|s)\big)$ for any $s$. Iterating from step $H$ to $1$, we obtain the NE for finite-horizon MGs as ${(\pi^*\_h,\nu^*\_h)}\_{h=1}^H$. It indicates that we can find the NE of a finite-horizon MG by decomposition into multiple matrix games whose NE always exists.
> The work [8] studies the finite-horizon tabular partially observable MG, where a similar idea of backward induction can be used to show its existence (consider the history-dependent policies and regard the history as the state in fully observable MGs).
>
> **3. Reward function depends on $o$ instead of $s$.**
>
> Our reward function is modeled following the recent works on the finite-horizon partially observable RL. Many existing works on partially observable MDPs (e.g., [4]) and MGs (e.g., [8]) defined the reward dependent on $o$. Such a definition is also natural since only the observation $o$ can be accessed by learners.
> Moreover, our definition does not conflict with a state-dependent reward function, denoted as $R_h(o,a,b)$.  At state $s$, we view $o$ as a random variable sampled from $\mathbb{O}_h(\cdot|s)$. We always have
> $$
> R\_h(s,a,b) = \sum\_{o\in \mathcal{O}} \mathbb{O}(o|s) r\_h(o,a,b).
> $$
> That is, $R_h(s,a,b)$ is intuitively an expectation of $r_h(o,a,b)$ based on the emission kernel. In this sense, whether the reward function depends on $s$ or $o$ does not introduce extra learning difficulty.
>
>
> **Reference**
>
> [1] Simon Du, et al. Bilinear classes: A structural framework for provable generalization in rl. ICML 2021.
>
> [2] Alekh Agarwal, and Tong Zhang. Model-based rl with optimistic posterior sampling: Structural conditions and sample complexity. NeurIPS 2022.
>
> [3] Tong Zhang. Feel-good thompson sampling for contextual bandits and reinforcement learning. SIAM Journal on Mathematics of Data Science 2022.
>
> [4] Chi Jin, et al. Sample-efficient reinforcement learning of undercomplete pomdps. NeurIPS 2020.
>
> [5] Yu Bai, Chi Jin. Provable Self-Play Algorithms for Competitive Reinforcement Learning. ICML 2020.
>
> [6] Qiwen Cui, and Simon S. Du. When are Offline Two-Player Zero-Sum Markov Games Solvable? NeurIPS 2022.
>
> [7] Qiaomin Xie, et al. Learning zero-sum simultaneous-move markov games using function approximation and correlated equilibrium. COLT 2020.
>
> [8] Qinghua Liu, et al. Sample-efficient reinforcement learning of partially observable markov games. NeurIPS, 2022.

---

> > ### Comment · Reviewer_qywY · 2023-08-14
> > **Comments on the response**
> >
> > Thanks to the authors for answering my questions. Considering them and the opinions of other reviewers, I am ready to raise my rating.

---

> > > ### Author Response · Authors · 2023-08-14
> > > **Thank you for raising the rating**
> > >
> > > Thank you for raising the rating! We greatly appreciate you taking the time to read our rebuttal and reconsider our work. We are happy to answer any further questions to address the remaining concerns regarding our submission.

---

### Official Review · Reviewer_DqQm · 2023-07-06

**Soundness:** 3 good
**Presentation:** 4 excellent
**Contribution:** 3 good
**Rating:** 7
**Confidence:** 4

**Summary:**

This paper investigates posterior sampling algorithms for competitive reinforcement learning (RL) with general function approximations in zero-sum Markov games (MGs). It introduces complexity measures for function approximation and proposes model-based self-play and adversarial posterior sampling methods to learn Nash equilibrium in partially observable states. The algorithms provide low regret bounds and can be applied to various tractable zero-sum MG classes in both fully observable and partially observable settings.

**Strengths:**

I think this work really pushes the Multiagent+PORL community research efforts further by answering:

> can we design a generic posterior sampling algorithm for MGRL in the context of function approximation?

The main contribution of model-based posterior sampling algorithm equipped with rigorous analyses is worthy for publication at NeurIPS.

**Weaknesses:**

I have only minor weaknesses for this work as follows:

1. For the self-play, Algorithm 1 and 2 seems repetitive. One can essentially combine them for presentation.
2. More discussion needs to be added comparing between Self-play and Adversarial results (Thm 1 and 2). For example, the self-play considered here is the zero-sum, hence the player 2 is the worst possible adversary. With this notion, comparing these two results will be helpful for future directions. It will be curious to see if the main player in Self-play (Thm 1 result) can handle the adversary in the Alg 3.

**Questions:**

na

---

> ### Author Rebuttal · Authors · 2023-08-09
>
> We thank Reviewer DqQm for recognizing the contribution of our submission. We will address your concerns below.
>
> **1. Combine the presentation of Algorithm 1 and Algorithm 2.**
>
> We will carefully polish our paper and revise the presentation of the algorithms in our next version.
>
> **2. More discussions on algorithms and the theoretical results for self-play and adversarial settings.**
>
> We politely point out that although Algorithm 1 can also be viewed as a learning algorithm for Player 1 with a worst possible adversary Player 2, it cannot be used to handle the adversarial setting in Algorithm 3 directly. The main reason is that the execution of Algorithm 1 requires both players to be controlled by the learner due to the setting of self-play, while Algorithm 3 cannot control Player 2 and views Player 2 as an arbitrary player. In Line 5 and Line 6 of Algorithm 1, we need to compute a policy $\underline{\nu}^t$ for Player 2, and then the exploration policy $\sigma^t$ is set to be $(\pi^t, \underline{\nu^t})$ for data collection as shown in Line 239 of the main text. Thus, Algorithm 1 forces Player 2 to take a specific policy $\underline{\nu^t}$ instead of an arbitrary uncontrollable one $\nu^t$ as in Algorithm 3. We will further highlight this difference in the revision.
>
> Regarding the theoretical result, the two different settings result in distinct regret metrics. We note that the result in Theorem 1 is inferred from Proposition 1 and Proposition 2, where Proposition 1 shows the regret for Algorithm 1 and Proposition 2 shows the regret for Algorithm 2 under the self-play setting. Then, to compare the results for Algorithm 1 and Algorithm 3, we focus on Proposition 1 and Theorem 2. The regret metric in Proposition 1 is defined as $\mathrm{Reg\_1^\mathrm{sp}}(T):=\sum_{t=1}^T(V_{f^\*}^\* - V\_{f^\*}^{\pi^t,\*}).$  And the regret metric in Theorem 2 is defined as $\mathrm{Reg^{\mathrm{adv}}}(T):=\sum_{t=1}^T(V\_{f^\*}^\* - V\_{f^\*}^{\pi^t,\nu^t}).$ Since $V_{f^*}^{\pi^t,\nu^t}\geq \min_\nu V_{f^*}^{\pi^t,\nu}= V_{f^*}^{\pi^t,*}$ always holds, we have $\mathrm{Reg}^{\mathrm{adv}}(T)\leq \mathrm{Reg}^{\mathrm{sp}}_1(T)$, which indicates that $\mathrm{Reg}^{\mathrm{sp}}_1(T)$ is a tighter regret metric. Moreover, Proposition 1 and Theorem 2 have comparable upper bounds in the same order (ignoring the numerical factors) but under different regret metrics. This indicates that the self-play algorithm for Player 1, i.e., Algorithm 1, can induce a tighter theoretical result, which reflects the power of self-play. We will add this discussion in our revision.

---

> > ### Comment · Reviewer_DqQm · 2023-08-20
> >
> > The rebuttal addressed my concerns. My rating is unchanged considering the rebuttal and other reviewers’ concerns.

---

> > > ### Author Response · Authors · 2023-08-20
> > >
> > > We greatly appreciate you taking the time to read our rebuttal. We are happy to answer any further questions to address the remaining concerns regarding our submission.

---

### Official Review · Reviewer_pkbQ · 2023-07-06

**Soundness:** 3 good
**Presentation:** 3 good
**Contribution:** 3 good
**Rating:** 5
**Confidence:** 1

**Summary:**

This paper focuses on posterior sampling for competitive reinforcement learning, aiming to propose a model-based self-play posterior sampling method to approximate Nash equilibrium in the case of self-play and adversarial learning. The theoretical analysis indicates that the proposed method achieves a low regret bound that can scale sublinearly converge.

**Strengths:**

1. This paper is well-organized and gives a throughout survey of the related work.
2. this paper seems to be a solid theoretical work, though I'm not entirely sure of that.

**Weaknesses:**

1. Too long a supplementary, so that the reviewer may miss some details

**Questions:**

line 56: "... partial observations into the posterior sampling framework under a MARL ...", so the question is what is the difficulty of using posterior sampling in POMDP?

---

> ### Author Rebuttal · Authors · 2023-08-09
>
> We thank Reviewer pkbQ for the valuable advice and questions. We will address your concerns below.
>
> **1. The supplementary material is long so that the reviewer may miss some details.**
>
> In our version, we will carefully revise the whole paper and move additional important details from the supplementary to the main text to highlight our technical contribution.
>
> **2. What is the difficulty of using posterior sampling in partially observable RL?**
>
> Our work studies a more challenging partially observable Markov game (POMG) problem than the single-agent POMDP. When it comes to the multi-agent setting, i.e., POMGs, to the best of our knowledge, there are no prior works studying the posterior sampling methods, particularly when incorporating function approximation. The difficulties of using posterior sampling in POMGs lie in the following aspects:
>
> **(1)** First, as a different learning framework, posterior sampling diverges fundamentally from existing POMG methods in techniques. In posterior sampling methods, we need to propose a proper model sampling distribution incorporating a well-designed loss formulation fitting the POMG models, which does not exist in the non-posterior sampling POMG method. Thus, this leads to different proof methods. Before our work, it was unknown how to design a statistically efficient posterior sampling algorithm for POMGs with a provable guarantee.
>
> **(2)** Second, the posterior sampling POMG method is not a direct extension of existing single-agent POMDP methods. We now have two coupled agents with different policies competing with each other in a min-max way instead of one agent pursuing a single maximization objective. In the POMG setting, we need to consider a more challenging minimax optimization problem. Obtaining such a competitive learning algorithm requires a novel algorithmic design methodology.
>
>
> **(3)** Third, compared to the fully observable Markov games (FOMGs), the intrinsic model structure differences between FOMGs and POMGs bring the challenges of extra emission kernels, unknown underlying states, and history-dependent policies. Thus, POMGs cannot be solved by the existing FOMG posterior sampling approaches.
>
> **(4)** Finally, when considering a more general function approximation setting in POMGs, novel multi-agent general function approximation conditions should be proposed to cover a wide range of known function approximation approaches. The conditions were unclear before our work.
>
> We tackle those challenges by successfully proposing unified statistically efficient posterior sampling algorithms for POMGs and FOMGs incorporating general function approximations. More importantly, our work covers both self-play and adversarial setups. We remark that the adversarial setting is not even studied in the existing posterior sampling method under full observability.

---

> > ### Author Response · Authors · 2023-08-20
> >
> > Dear Reviewer pkbQ,
> >
> > We sincerely appreciate you taking the time to thoroughly evaluate our paper and provide insightful questions.
> >
> > In our rebuttal, we aimed to carefully and comprehensively answer your questions. We sincerely hope our responses and clarifications can adequately alleviate your initial concerns about our work.
> >
> > For your concerns about the long supplementary material, since our work presents a novel RL algorithm with theoretical guarantees, we include rigorous proofs and analysis in the supplementary material to support our results. The detail in the supplementary material is necessary for a technically sound paper.
> >
> > We truly value the discussion period and hope to address any concerns to the best of our ability on the last day of this period. Please do not hesitate to let us know if there are any lingering concerns or unclearness in our rebuttal. We would be more than happy to address them.

---

> > > ### Comment · Reviewer_pkbQ · 2023-08-21
> > > **Thanks for your clarification**
> > >
> > > I thank the authors for their response and clarification, which further helps me understand your contribution. But I'll keep my score as it is hard to check the correctness of such long theoretical parts in such a short period.

---

> > > > ### Author Response · Authors · 2023-08-21
> > > >
> > > > We sincerely appreciate the reviewer taking the time to read through our rebuttal. We are pleased to know that our rebuttal was able to help resolve your questions.
> > > >
> > > > To enable readers to grasp our main proof ideas, we provided a proof sketch at the end of the main text. In our revision, we will ensure to further highlight our technical contributions by extracting additional important details from the supplement and incorporating them into the main text.

---

### Official Review · Reviewer_Gu3i · 2023-07-26

**Soundness:** 3 good
**Presentation:** 3 good
**Contribution:** 3 good
**Rating:** 6
**Confidence:** 3

**Summary:**

This paper investigates posterior sampling algorithms for competitive RL in the context of general function approximations. The authors propose the self-play and adversarial generalized eluder coefficient (GEC) as complexity measures for function approximation, capturing the exploration-exploitation trade-off in MGs. They further provide low regret bounds for proposed algorithms that can scale sublinearly with the proposed GEC and the number of episodes T.

**Strengths:**

1. Two new generalized eluder coefficients are proposed as the complexity measure for the competitive RL with function approximation.
2. The authors also propose a novel model-based posterior sampling algorithm with self-play to learn the Nash equilibrium with provable regret bounds.
3. The technical contribution of the paper looks solid.

**Weaknesses:**

Currently, readers are hard to follow the technical results in the main paper. It will be better if the author could include some explanatory parts for explaining the technical details intuitively, so as to highlight their technical contribution.

**Questions:**

Please refer to the Weaknesses section.

**Limitations:**

No.

---

> ### Author Rebuttal · Authors · 2023-08-09
>
> We thank Reviewer Gu3i for the valuable advice and questions. We will address your concerns below.
>
> **1. Explain the technical results intuitively and the associated contributions.**
>
> We provide the intuitive explanation by elucidating connections between the subsequent four aspects:
>
> **(1) Complexity conditions:** Our paper defines two general complexity conditions, the Self-Play GEC and the Adversarial GEC, for competitive reinforcement learning (RL). Since we consider a Markov game setting, in contrast to the single-agent setting, our definitions of the complexity conditions in Definitions 1 and 3 reflect the competitive nature of the two players via the defined sequence of the policies and fit the two important Markov game scenarios, i.e., the self-play and the adversarial settings. These conditions are not proposed in prior works and cannot be directly generalized from the single-agent setting. The inequalities in these definitions have a particular meaning: the left-hand side of the inequality is the prediction error defined based on the value difference while the right-hand side is the training error with a multiplicative factor $d_\mathrm{GEC}$ defined on a problem-specific loss function plus a small burnt-in error. We use these conditions to characterize the exploration hardness of online competitive RL. Let $\mathcal{F}$ represent such model classes. Intuitively, a sequence of approximation functions in $\mathcal{F}$ satisfies that under the self-play or adversarial scenarios, if they induce small in-sample training error on a collected well-explored dataset, then the out-of-sample prediction error on the next trajectory generated is also small. Interestingly, as we show in Section 5, we discover with detailed proofs that many well-known function classes for function approximation in both fully observable Markov games (FOMGs) and partially observable Markov games (POMGs) can be subsumed in our defined Self-Play GEC and Adversarial GEC function classes with different dimensional factor $d_\mathrm{GEC}$. We can even propose a new function class named Decodable POMG that can be covered by our framework.
>
> **(2) Algorithms:** The definition of Self-Play/Adversarial GEC motivates us to design algorithms based on posterior sampling. In our proposed posterior sampling algorithms, we assign a larger probability to a model $f\in \mathcal{F}$ if its in-sample training error is small. Moreover, according to the characteristics of the self-play and the adversarial scenarios in a competitive setting, we design algorithms from different perspectives. In particular, in Algorithm 1 and Algorithm 2 for the self-play setting, where we can coordinate both players for learning, we have a two-step posterior sampling strategy, where the first sampling step for learning the Nash equilibrium policy and the second sampling step is for constructing the opponent's best response where the opponent assists by exploiting the main player's weakness. On the other hand, in Algorithm 3 for the adversarial setting, since the opponent's policy cannot be controlled by the learner, we do not have the second sampling step, which thus leads to different algorithm designs. We note that our self-play and adversarial learning algorithms are the first unified methods considering both full and partial observability.
> Moreover, in the construction of the posterior sampling distributions in all the algorithms, we add and customize the optimism terms in terms of the different learning scenarios, i.e., $V_f^*$, $V\_f^{\pi^t,\*}$, $V_f^{*,\nu^t}$, for learning efficiency in our proof. Moreover, proving the regret bound itself requires a careful analysis of integrating the Self-Play/Adversarial GEC, the loss functions for FOMGs and POMGs, optimistic posterior sampling, and the self-play and adversarial algorithm frameworks in a unified perspective.
>
> **(3) Main regret results:** For self-play and adversarial settings, under different learning regret metrics, we prove the upper bounds of the proposed posterior sampling algorithms by incorporating the newly proposed Self-Play and Adversarial GEC function classes and the associated complexity conditions. Intuitively, in both Theorem 1 and Theorem 2, we can achieve near-optimal $O(\sqrt{T})$ bounds in terms of the number of episodes $T$, which justifies the statistical efficiency of our algorithms. Moreover, the results have a dependence on another two factors, $d_\mathrm{GEC}$ and $\omega$. The factor $d_\mathrm{GEC}$ as defined in Definitions 1 and 3 represents the complexity of the Self-Play and Adversarial GEC function classes, whose value can be instantiated in concrete example subclasses as elaborated below. The quantity $\omega$ measures how well the initial prior distributions cover the optimal model $f^*$, which thus further reveals the size of the function space $\mathcal{F}$. In fact, we can prove in our supplement that when $\mathcal{F}$ is finite, it equals the value of $\log |\mathcal{F}|$, and when $\mathcal{F}$ is infinite, it is the log covering number of the space.
>
> **(4) Examples:** In Section 5, we present a bunch of examples for approximation function classes in both FOMGs and POMGs. In the supplement, we provide detailed and rigorous proofs to show that all these demonstrated classes are subsumed by our proposed Self-Play and Adversarial GEC classes, indicating the generality of the proposed GEC classes. We additionally provide a new function class named Decodable POMG and show that it can also be covered by our proposed function classes. We eventually calculate the concrete relation between $d_{\mathrm{GEC}}$ and the specific complexity measure in each of these classes. Thus, plugging the calculated value $d_{\mathrm{GEC}}$ into our main theorems leads to the theoretical guarantee for the instantiation of our proposed algorithms on each function class.

---

> > ### Comment · Reviewer_Gu3i · 2023-08-17
> >
> > Thank you very much for your response and clarification.

---

> > > ### Author Response · Authors · 2023-08-19
> > >
> > > We greatly appreciate you taking the time to read our rebuttal. We are happy to answer any further questions to address the remaining concerns regarding our submission.

---

### Decision · Program_Chairs · 2023-09-21

**Decision:**

Accept (poster)

**Comment:**

The paper extends the previous GEC complexity measure for single agent RL to the Markov game settings. The new measures look interesting and capture many existing examples. The proposed posterior sampling algorithms are reasonable and achieve near optimal regret in both self-play and adversarial learning settings.